# Unsupervised Multiple Kernel Learning for Graphs via Ordinality Preservation

**Yan Sun   Stanley Kok**
School of Computing, National University of Singapore, Singapore
`yansun@comp.nus.edu.sg  skok@comp.nus.edu.sg`

## Abstract

Learning effective graph similarities is crucial for tasks like clustering, yet selecting the optimal kernel to evaluate such similarities in unsupervised settings remains a major challenge. Despite the development of various graph kernels, determining the most appropriate one for a specific task is particularly difficult in the absence of labeled data. Existing methods often struggle to handle the complex structure of graph data and rely on heuristic approaches that fail to adequately capture the global relationships between graphs. To overcome these limitations, we propose Unsupervised Multiple Kernel Learning for Graphs (UMKL-G[1]), a model that combines multiple graph kernels without requiring labels or predefined local neighbors. Our approach preserves the topology of the data by maintaining ordinal relationships among graphs through a probability simplex, allowing for a unified and adaptive kernel learning process. We provide theoretical guarantees on the stability, robustness, and generalization of our method. Empirical results demonstrate that UMKL-G outperforms individual kernels and other state-of-the-art methods, offering a robust solution for unsupervised graph analysis.

## 1 Introduction

Graphs are ubiquitously used to represent structured data in diverse domains such as bioinformatics, chemoinformatics, and social networks. Learning a semantically meaningful similarity between graphs is crucial as it captures the essential characteristics and functional properties that distinguish one graph from another. For example, in bioinformatics, the secondary structure of a protein can be regarded as a graph where nodes are atoms and edges are chemical bonds. Learning semantic similarities between such graphs enables more effective graph-level tasks, such as determining enzymatic activity, where subtle structural variations play a key role.

Kernel methods are naturally suited to measuring graph similarity. Graph kernels with the $\mathcal{R}$-convolution framework recursively break down graphs into substructures — like paths (Borgwardt & Kriegel, 2005), graphlets (Shervashidze et al., 2009), walks (Vishwanathan et al., 2010), and subtrees (Shervashidze et al., 2011) — and then compare these substructures between two graphs (Kriege et al., 2018). In addition, there are other types of graph kernels developed from the principles of optimal assignment (Fröhlich et al., 2005; Kriege et al., 2016), optimal transport distance (Togninalli et al., 2019; Chen et al., 2022), and maximum mean discrepancy (Sun & Fan, 2023).

Given the abundance of graph kernels, it is not straightforward to determine which one is the most suitable for a specific task and dataset. Some works try to determine the expressiveness of candidate kernels theoretically (Kriege et al., 2018; Oneto et al., 2017). However, it is still not clear which one would empirically perform the best in a novel setting, where graphs are not labeled and are different from those encountered in previous studies. In reality, the performance of graph kernels varies on a case-by-case basis (Kriege et al., 2020). For instance, the Weisfeiler-Lehman (WL) kernels are theoretically less expressive than the Shortest Path (SP) graph kernels as WL kernels fail to distinguish connectivity (Kriege et al., 2018). However, empirically, WL kernels perform better in terms of classification accuracy in certain chemical compound datasets (Kriege et al., 2020). Similarly, the simple Random Walk (RW) kernels outperform Graphlet kernels (Borgwardt et al.,

---

[1]Our code is available at `https://github.com/yan-sun-x/Ensemble_Kernel/`

2020), even though RW kernels cannot identify triangle freeness (Kriege et al., 2018). There are several possible reasons for this phenomenon ranging from the match between the graph kernels and the graph structure in a particular dataset to the number of isomorphic graphs in the dataset that belong to different classes (Nikolentzos et al., 2021). Given that no single graph kernel dominates all downstream tasks, an intriguing question emerges:

*Is it possible to ensemble individual graph kernels and perform well in an unsupervised setting?*

If so, then we can achieve better performance in graph-level tasks by learning the optimal kernel values in a data-driven manner. To solve this problem, an intuitive solution is to learn an optimal kernel from multiple graph kernels. Previous studies in Multiple Kernel Learning (MKL) provide a *supervised* framework to learn the kernel directly from data (Gönen & Alpaydın, 2011). Specifically, MKL leverages pre-specified weak kernels and labels of data points to obtain the optimal kernel as a weighted combination of the weak ones. It is worthwhile to note that the existing MKL algorithms fail in the *unsupervised* setting (e.g., graph-level clustering, which is common in many real-world graph applications (Ju et al., 2023)). Although there exists an unsupervised algorithm for MKL (Zhuang et al., 2011) based on the locality preserving principle, it is designed for Euclidean data. This limits its applicability for graphs, a more complex, non-Euclidean data type that captures both individual node features and their pairwise relationships. Noticing these limitations, Mariette & Villa-Vialaneix (2018) proposes sparse-UMKL, which aims to preserve local geometry by constructing $k$-nearest neighbors. However, this method falls short in practice, as its heuristic approach to approximating neighbors leads to poor generalization in empirical experiments. In summary, given the current algorithms, we conclude the following for unsupervised MKL: ① *preserving the data topology is essential*, and ② *achieving effective generalization for graph data remains a significant challenge*. These insights motivate us to directly leverage the ordinal relationship between graphs using kernel values, without the need for learning explicit graph representations.

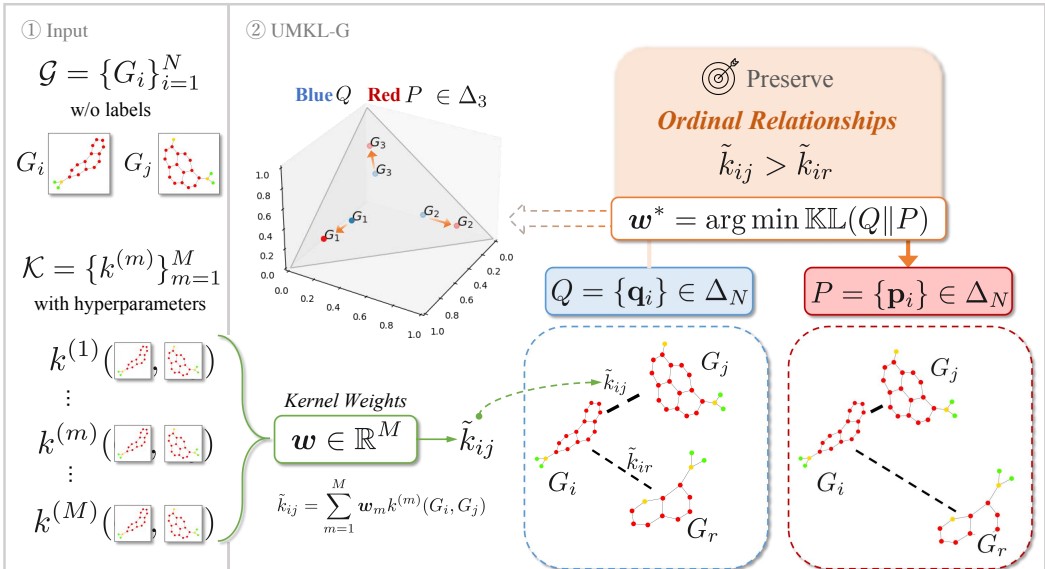

Figure 1: **Overview of the UMKL-G Model.** The model starts with $N$ input graphs, which are processed through $M$ multiple graph kernels with configurable hyperparameters. The learnable kernel weights $\boldsymbol{w}$ combine the kernels into an optimal composite kernel $\tilde{k}$ while preserving the *ordinal relationships* between graphs. Each graph is represented as a point on the simplex, where $Q$ is the initial probability distribution of graph similarities and $P$ is the powered distribution that emphasizes stronger relationships. The goal is to adjust the positions of these points in the simplex to preserve both local and global graph structures inferred from $\tilde{k}$, ensuring effective performance in tasks such as clustering. As an illustration, the probability simplex $\Delta_N$ ($N = 3$) at the center shows the optimization process, indicated by the direction of the arrow.

In this work, we develop a simple yet effective approach to learning an optimal kernel for graphs in an unsupervised manner and name it **U**nsupervised **M**ultiple **K**ernel **L**earning for **G**raphs (UMKL-G; illustrated in Figure 1). Our proposed method automates the procedure to select and combine different types of graph kernels as well as configure their respective hyperparameters. Our empirical results show the composite graph kernel learned by UMKL-G has better performances compared to individual kernels and state-of-the-art baseline algorithms in the graph-level clustering task. The main contributions of this work are summarized as follows:

- We propose UMKL-G, an efficient unsupervised method that combines multiple graph kernels by preserving topological structures through ordinal relationships of the kernel values.

- We provide theoretical guarantees and their empirical validation, such as Lipschitz continuity for smooth optimization, robustness to kernel perturbations, and generalization stability, ensuring that the algorithm effectively adapts to diverse, noisy, and unseen data.

- We empirically evaluate UMKL-G, showing its effectiveness over individual graph kernels and three state-of-the-art baselines on eight benchmark datasets.

## 2 RELATED WORK

**Graph Kernels.** Much existing research on graph kernels can be seen as feature engineering efforts, focusing on identifying the most suitable graph aspects for defining graph similarity (Borgwardt et al., 2020). The initial wave of graph kernels, termed as $\mathcal{R}$-convolution, concentrated on decomposing graphs into smaller, easily comparable substructures. For instance, random walk kernels, introduced by Gärtner et al. (2003), count matching random walks in two graphs; shortest path kernels invented by Borgwardt & Kriegel (2005) count pairs of shortest paths with the same length; graphlet kernels developed by Shervashidze et al. (2009) further contributed by counting occurrences of small subgraphs, efficiently capturing local structures; and Weisfeiler-Lehman (WL) graph kernels proposed by (Shervashidze et al., 2011), iteratively refine node labels to capture larger substructures, enhancing their discriminative power by capturing edge attributes and multi-resolution structures. Additionally, the concept of optimal matching has been explored in various graph kernels, ranging from the optimal assignment kernel (Fröhlich et al., 2005) to the Weisfeiler-Lehman optimal assignment kernel (Kriege et al., 2016). A notable trend in recent research has been the modification of Weisfeiler-Lehman (WL) kernels using Wasserstein distances (Togninalli et al., 2019; Chen et al., 2022). Togninalli et al. (2019) integrate the Wasserstein distance into the WL framework, allowing for a more refined comparison of graphs, particularly those with continuous node attributes. Chen et al. (2022) propose a novel concept of WL distance, a polynomial-time computable metric that is more sensitive to subtle graph differences than traditional WL methods. Deep-learning-inspired graph kernels, such as deep graph kernels by Yanardag & Vishwanathan (2015), combine traditional methods with neural networks to learn latent substructure representations, and GCN-based kernels (Ye et al., 2020) integrate graph convolutional networks to capture complex, high-level features. Sun & Fan (2023) propose a deep graph kernel using maximum mean discrepancy (MMD-GK) that integrates graph kernel learning with graph neural networks, achieving promising performance in graph classification and clustering tasks. In this abundance of graph kernels, each kernel has its strengths and limitations (Borgwardt et al., 2020), both of which are the results of an inherent trade-off between exhaustive feature extraction and computational feasibility. Additionally, many graph kernels are restricted by the attributes of graphs they can handle, e.g., Weisfeiler-Lehman (WL) kernels struggle to distinguish node and edge attributes. In this study, we utilize graph kernels regardless of their time complexity or graph assumptions, provided their kernel matrices can be precomputed. In this study, we concentrate on non-deep kernels, but our method applies to other graph kernels without any loss of generality.

**Multiple Kernel Learning.** Multiple Kernel Learning (MKL) (Lanckriet et al., 2004) aims to learn a linear combination of a set of predefined *weak* kernels to identify a good kernel for a given problem. MKL algorithms have been developed for supervised, semi-supervised and unsupervised learning. In the supervised framework, MKL algorithms are supported by several theoretical results that bound the difference between the true error and the empirical margin error (or, estimation error) (Gönen & Alpaydın, 2011). Beyond the supervised framework, there is a growing interest in exploring MKL approaches for unsupervised scenarios since Zhuang et al. (2011). Most methods seek a kernel that minimizes the distortion between all training data or that minimizes the approximation of the orig-

inal data in the kernel embedding (Lin et al., 2010; Zhuang et al., 2011). Their methods presume that the learned kernel values coincide with the *pseudo-ground truth* of the data's geometry. (In preserving the topological structure in an unsupervised setting, it is intuitive and appealing to consider the ground truth—the true underlying relationships or labels in the data—similar to its role in supervised learning. However, since such ground truth is unavailable in unsupervised settings, we assume a *pseudo-ground truth*, such as estimated similarity measures or inferred geometric structures that approximate the inherent relationships among data points.) In addition, they assume that the data is represented as numerical vectors so that it can be directly computed in the input space using the Euclidean distance. Unfortunately, such unsupervised methods are not applicable when the inputs are graphs, which are not naturally represented as vectors. Furthermore, no explicit kernel embeddings exist for those graph kernels that are not $\mathcal{R}$-convolution. Mariette & Villa-Vialaneix (2018) introduced sparse-UMKL, which preserves local geometry by constructing a k-nearest neighbor graph for each kernel as a *pseudo-ground truth* of the data's underlying structure. However, this method struggles in practice due to its heuristic approximation of local geometry and a rigid objective function, resulting in poor generalization in experiments. Rather than explicitly constructing the local geometry, we tackle the unsupervised multiple kernel learning problem for graphs by learning the complete connectivity on a probability simplex. In this approach, the *pseudo-ground truth* of the topology is implicitly inferred from the data.

## 3 BACKGROUND

Before introducing our method, we first present relevant background and notations.

**Multiple Kernel Learning for Graphs.** Let $\mathcal{G} := \{G_1, \cdots, G_N\}$ denote a set of $N$ graphs and $\mathcal{K} := \{k^{(1)}, \cdots, k^{(M)}\}$ denote a set of $M$ graph kernels, where each graph kernel $k^{(m)}$ is a function $k^{(m)} : \mathcal{G} \times \mathcal{G} \to [0, 1]$. With learnable weights $\boldsymbol{w} = (w_1, \cdots, w_m, \cdots, w_M) \in \mathbb{R}^M$, where $\sum_m w_m = 1$ and $w_m \geq 0$ for all $m$, the pairwise kernel value $\tilde{k}_{ij}$ between any pair of graphs $G_i, G_j \in \mathcal{G}$ over a set of graph kernels $\mathcal{K}$ is defined as a weighted sum of the individual kernel values. Denoting $\boldsymbol{k}_{ij} = (k^{(1)}(G_i, G_j), \cdots, k^{(M)}(G_i, G_j)) \in \mathbb{R}^M$, we have

$$\tilde{k}_{ij} := \tilde{k}_{ij}(\boldsymbol{w}) = \boldsymbol{w}^\top \boldsymbol{k}_{ij} = \sum_{m=1}^{M} w_m \cdot k^{(m)}(G_i, G_j) \tag{1}$$

**Probability Simplex.** Each point on probability simplex $\Delta_N$ represents a probability distribution over a finite number of mutually exclusive events and can be represented by $N$ non-negative numbers that sum to 1. We define an $N$-probability simplex as the collection of points $\Delta_N := \{(x_1, \cdots, x_N) \in \mathbb{R}^N | \sum_{i=1}^{N} x_i = 1, x_i \geq 0 \ \forall i\}$. When $N = 2$, this space is a line, when $N = 3$ it is a filled-in triangle, and when $N = 4$ it is a solid tetrahedron.

## 4 METHOD: UMKL-G

In unsupervised multiple kernel learning (MKL), it is crucial to preserve the topology of data because this maintains the intrinsic structural properties and relationships between data points. This is even more important in our case, where each data point represents the complex structure of a graph. Adhering to this first principle, we treat all kernel values $\{\tilde{k}_{ij}\}$ as the *pseudo-ground truth* structure, without explicitly constructing local geometry for data reconstruction, i.e., graphs in this context. Instead, our method focuses on preserving the data structure through ordinal relationships, through which the *pseudo-ground truth* is implicitly learned. Specifically, we utilize all kernel values $\tilde{k}_{ij}$ to construct a probability simplex space among the graphs. This strategy ensures that the local topology of the data is maintained while learning the weights $\boldsymbol{w}$ within this space.

### 4.1 PRESERVING TOPOLOGY

Inspired by Agarwal et al. (2007) and Vankadara et al. (2023), we preserve the topology between graphs by maintaining the order of the similarity between graphs. Specifically, we focus on the ordinal relationship that, according to the learned composite kernel $\tilde{k}_{ij}$, graph $G_i$ is more similar to $G_j$ than to graph $G_r$ for any triplet $(i, j, r)$.

**Definition 1.** *(Ordinal Relationship) Consider the graph $G_i$ where its similarities to $G_j$ and $G_r$ are respectively given by the learned kernel values $\tilde{k}_{ij}(\boldsymbol{w})$ and $\tilde{k}_{ir}(\boldsymbol{w})$. The ordinal relationship between $G_j$ and $G_r$ with respect to $G_i$ are preserved if, for any weights $\boldsymbol{w}$:*

$$\tilde{k}_{ij}(\boldsymbol{w}) > \tilde{k}_{ir}(\boldsymbol{w})$$

More generally, preserving ordinal relationships for all pairs of graphs ensures that for any graph $G_i$, its most similar graphs remain consistent within the learned kernel space. This approach maintains the local neighborhood structure around each graph, preserving the data's intrinsic topology.

## 4.2 Graphs on a Probability Simplex

To define a feasible probability simplex, we consider a probability space over the set of graphs $\mathcal{G} := \{G_1, \cdots, G_N\}$. The sample space $\Omega$ is the set of all ordered pairs of graphs, $\Omega = \mathcal{G} \times \mathcal{G}$. The $\sigma$-algebra $\mathcal{E}$ represents the events over the graph pairs and the probability measure $P$ assigns probabilities to each pair based on its graph kernel value, which measures the similarity between two graphs. For each fixed graph $G_i$, the event of selecting a similar graph $G_j$ from $\mathcal{G}$ is denoted as $(G_i, G_j)$, where the similarity between $G_i$ and $G_j$ is quantified by their kernel value, i.e.,

$$q_{i_j} := q_{i_j}(\boldsymbol{w}) = \frac{\tilde{k}_{ij}(\boldsymbol{w})}{\sum_{j'=1}^{N} \tilde{k}_{ij'}(\boldsymbol{w})}. \tag{2}$$

where $\sum_j q_{i_j} = 1$ and $q_{i_j} \geq 0$ for all $j$, ensuring the axioms of a probability measure are satisfied. The probabilities $q_{i_j}$ are thus normalized kernel values determined via learned weights $\boldsymbol{w}$, reflecting the relative similarity between $G_i$ and all other graphs in $\mathcal{G}$.

Given the probability measure above, each graph could be represented as a point in the probability simplex $\Delta_N$, where $G_i$ is associated with the point $\boldsymbol{q}_i = (q_{i_1}, \cdots, q_{i_j}, \cdots, q_{i_N}) \in \mathbb{R}^N$. This representation preserves ordinal relationships since $q_{i_j} > q_{i_r}$ if and only if $\tilde{k}_{ij} > \tilde{k}_{ir}$. Overall, we define a set of the probability simplex vectors as $Q = \{\boldsymbol{q}_i\}_{i=1}^{N} \subset \Delta_N$.

## 4.3 Target $P$: Powered Kernels

To effectively preserve the ordinal relationships among graphs, we capture key neighborhood structures by amplifying the stronger similarities between graphs towards a set of target probabilities. Specifically, we define a target probability with powered kernel values to reflect these relationships.

**Definition 2.** *(Powered Kernel) The target probability is transformed through powered kernel values*

$$p_{i_j}^{(o)} = \frac{\tilde{k}_{ij}^o}{\sum_{j'} \tilde{k}_{ij'}^o}, \tag{3}$$

*where the power parameter $o \in \mathbb{N}^+$. On the same probability simplex $\Delta_N$, another set of probability simplex vectors is defined as $P^{(o)} = \{\boldsymbol{p}_i^{(o)}\}_{i=1}^{N} \subset \Delta_N$, where $\boldsymbol{p}_i^{(o)} = (p_{i_1}^{(o)}, \cdots, p_{i_N}^{(o)}) \in \mathbb{R}^N$.*

Note that $Q \triangleq P^{(1)}$ (with $o = 1$) is a special case. For simplicity, we denote $P$ as $P^{(o)}$ for an arbitrary parameter $o$. When the kernel values are raised to a power $o > 1$, for $\tilde{k}_{ij} > \tilde{k}_{ir}$, the powered kernel values will emphasize the stronger similarity by a greater amount than the weaker similarity, i.e., $\tilde{k}_{ij}^o \gg \tilde{k}_{ir}^o$. By emphasizing the larger kernel values, we ensure that the ordinal relationships between graphs are preserved (Theorem 1), while focusing on the more significant relationships between graphs (Theorem 2). Details of these proofs are provided in Appendix B.

**Theorem 1.** *(Ordinality Preservation) Let $\tilde{k}_{ij}$ and $\tilde{k}_{ir}$ represent the kernel values between graph $G_i$ and graphs $G_j$ and $G_r$, respectively. If the ordinal relationship $\tilde{k}_{ij} > \tilde{k}_{ir}$ holds, then for any power $o > 1$, the corresponding probabilities in the powered kernel distribution satisfy $p_{i_j}^{(o)} > p_{i_r}^{(o)}$.*

In the same probability simplex space $\Delta_N$, the target probability $P$ is less uniform compared to the original $Q$, which leads to a concentration of probability mass on fewer components for each vector $\boldsymbol{p}_i$. As such, the concentration effect of the powered kernel is formalized as follows.

**Theorem 2.** *(Concentration Effect) For any graph $G_i$ and any $o > 1$, the entropy of the powered kernel distribution $\boldsymbol{p}_i^{(o)}$ is strictly less than the entropy of the original distribution $\boldsymbol{q}_i$, i.e.,*

$$H(\boldsymbol{p}_i^{(o)}) < H(\boldsymbol{q}_i). \tag{4}$$

The concentration effect suggests that raising the kernel values to a power $o > 1$ enhances the contrast between significant and insignificant relationships. Thus, compared to $Q$, the probability simplex vectors $P$ are more reflective of the underlying structure between graphs. By emphasizing the most meaningful connections, $P$ becomes a more accurate representation of the data's inherent geometry. Consequently, by transforming the original probabilities $Q$ towards the target ones $P$, we implicitly learn a *pseudo-ground truth* structure $\tilde{k}$, which represents the enhanced composite kernel that reveals the more meaningful patterns in the data.

### 4.4 Objective Function

To preserve the topological structure during the learning of the weights $\boldsymbol{w}$, we employ the Kullback-Leibler (KL) divergence. This divergence quantifies the difference between two probability distributions, $P$ and $Q$, enabling us to align the learned distribution with the underlying graph relationships while minimizing the distortion of ordinal relationships. Specially, we use the reverse divergence:

$$\mathcal{L}^{(o)} = \mathbb{KL}(Q\|P) = \sum_{i,j} \left( q_{i_j} \log q_{i_j} - q_{i_j} \log p_{i_j}^{(o)} \right), \tag{5}$$

which measures how much information is lost when $P$ is used to approximate $Q$. This formulation includes a negative entropy term, $q_{i_j} \log q_{i_j}$, which plays a crucial role in regularizing the learned distribution $Q$. When minimizing $\mathcal{L}^{(o)}$, the entropy term penalizes highly concentrated distributions and encourages $Q$ to spread its probability mass more evenly across all graphs. This prevents any single graph from dominating the distribution excessively.

In addition, the reverse divergence tends to be more sensitive to cases where $Q$ has a high probability and $P$ assigns a low probability, thereby encouraging the learned distribution $Q$ to avoid assigning high probabilities to regions not covered by $P$. By minimizing the divergence, we ensure that the learned distribution $Q$ closely approximates the target distribution $P$, thus preserving the topological properties of the original graph data.

### 4.5 Proposed Algorithm

In this section, we summarize the overall algorithm and how these objective functions are integrated into the learning process. The inputs are groups of $N$ graphs $\mathcal{G} := \{G_1, \cdots, G_N\}$, $M$ graph kernels $\mathcal{K} := \{k^{(m)}\}_{m=1}^M$, initial weights $\boldsymbol{w}^{(0)} = (\frac{1}{M}, \cdots, \frac{1}{M}) \in \mathbb{R}^M$, and the power hyperparameter $o$. As outlined in Algorithm 1, our goal is to learn the optimal weights $\boldsymbol{w}$ for combining the graph kernels. Since the composite kernel value is defined in Eq. 1, where the weights must satisfy the condition $\sum_m w_m = 1$ and $w_m \geq 0$ $\forall m$, we incorporate a projection step (see Algorithm 2 in Appendix D) to ensure these conditions are met. Specifically, we project the weights onto the unit simplex, ensuring that the composite kernel retains the properties of the original kernels, such as being positive semi-definite. Although there are other methods to ensure unit simplex and non-negativity of weights, our method is efficient with a computational complexity dominated by the sorting step, which is $O(M \log M)$. In addition, the algorithm provides an optimal projection in terms of the Euclidean distance from the original vector to the simplex. It ensures the smallest adjustment needed to project the vector onto the simplex. We provide a more detailed discussion in Appendix D. Once the optimal weights $\boldsymbol{w}$ are learned, the resulting composite kernel $\tilde{k}$ can be directly applied to various machine learning tasks, both supervised and unsupervised.

### 4.6 Theoretical Analysis

To support the robustness and effectiveness of UMKL-G, we provide a detailed theoretical analysis. Specifically, we establish three key properties: (1) Lipschitz continuity of the gradient of the objective function $\mathcal{L}^{(o)}$ (Theorem 3), ensuring smooth optimization and convergence; (2) robustness to

---

**Algorithm 1:** Unsupervised Multiple Kernel Learning for Graphs (UMKL-G)

---

**Input** : Unlabeled graphs $\mathcal{G} = \{G_1, \cdots, G_N\}$, base kernels $\mathcal{K} = \{k^{(1)}, \cdots, k^{(M)}\}$, initial
        weights $\boldsymbol{w}_0 = (\frac{1}{M}, \cdots, \frac{1}{M})$, power hyperparameter $o$.
**Output:** Kernel weights $\boldsymbol{w}$
Initialize weights $\boldsymbol{w} = \boldsymbol{w}_0$;
**while** *not converged* **do**
    **for** *each pair of graphs* $(G_i, G_j)$ *in* $\mathcal{G}$ **do**
        $\tilde{k}_{ij} = \sum_{m=1}^{M} w_m \cdot k^{(m)}(G_i, G_j)$          // Compute the pairwise kernel value
    **for** *each graph* $G_i$ *in* $\mathcal{G}$ **do**
        $\boldsymbol{q}_i = \left( \frac{\tilde{k}_{i1}}{\sum_{j=1}^{N} \tilde{k}_{ij}}, \cdots, \frac{\tilde{k}_{iN}}{\sum_{j=1}^{N} \tilde{k}_{ij}} \right)$      // Compute the reference $Q$
        $\boldsymbol{p}_i^{(o)} = \left( \frac{\tilde{k}_{i1}^o}{\sum_{j=1}^{N} \tilde{k}_{ij}^o}, \cdots, \frac{\tilde{k}_{iN}^o}{\sum_{j=1}^{N} \tilde{k}_{ij}^o} \right)$      // Compute the target $P$
    Update weights $\boldsymbol{w}$ by minimizing $\mathcal{L}^{(o)}$:
        $\boldsymbol{w} = \boldsymbol{w} + \Delta\boldsymbol{w}$        // $\Delta\boldsymbol{w}$ depends on the optimizer;
    Project $\boldsymbol{w}$ onto the unit simplex using Algorithm 2;
**return** $\boldsymbol{w}$

---

kernel perturbations (Theorem 4), guaranteeing the reliability of the solution even in the presence of noise or slight inaccuracies in kernel computation; and (3) generalization and stability (Theorems 5 and 6), showing the capability of the proposed algorithm to perform well on unseen data. Detailed proofs for these results are provided in Appendix B.

**Theorem 3.** *For the set of graphs* $\mathcal{G}$ *with* $N = |\mathcal{G}|$ *and the graphs* $G_i, G_j \in \mathcal{G}$, *let* $\|\boldsymbol{k}_{ij}\| \leq K_{\max}$
*(* $\boldsymbol{k}_{ij}$ *is defined for Eq. 1),* $0 < \alpha \leq \sum_j \tilde{k}_{ij} \leq \beta$, *and* $0 < \delta \leq q_{i_j} \leq \gamma < 1$. *Denote* $\psi_1$ *as* $\frac{N}{\alpha^2}$,
$\psi_2$ *as* $\frac{\beta+N}{\alpha^3}$, *and* $\psi_3$ *as* $\frac{\gamma}{\delta}$. *The gradient of the objective function* $\mathcal{L}^{(o)}$ *is Lipschitz continuous with a constant* $L$, *such that for any* $\boldsymbol{w}, \boldsymbol{w}' \in \mathbb{R}^M$: $\|\nabla_{\boldsymbol{w}} \mathcal{L}^{(o)}(\boldsymbol{w}) - \nabla_{\boldsymbol{w}} \mathcal{L}^{(o)}(\boldsymbol{w}')\| \leq L\|\boldsymbol{w} - \boldsymbol{w}'\|$ *with*

$$L = C_1 \cdot N^2 (1 + \gamma N) \cdot K_{\max}^2, \tag{6}$$

*where the constant* $C_1 = \left( 1 + (o-1)\log\delta^{-1} + \log(N\delta^{-o}) + \gamma \right) \cdot \psi_1 + (1 + (o-1)\delta^{-1} + (o + (2o-1)\psi_3^o)\psi_3^{o-1}\delta^{-1}) \cdot \psi_2$.

This result ensures that the gradient of the objective function is Lipschitz continuous, which implies that small changes in the weight vector $\boldsymbol{w}$ lead to proportionally small changes in the gradient. This property is crucial for the stability of UMKL-G, allowing for controlled and predictable updates during the optimization process.

Next, we show that UMKL-G is robust to kernel perturbations. Specifically, small variations in the kernel values, whether due to noise, computational inaccuracies, or differences in graph properties, only result in limited changes in the optimal solution.

**Theorem 4.** *Let the perturbed kernel values be* $\boldsymbol{k}'_{ij} = \boldsymbol{k}_{ij} + \Delta k_{ij}$, *where* $\|\Delta k_{ij}\| \leq \eta$ *for any graphs* $G_i$ *and* $G_j$. *Assume* $\sum_{j'} \tilde{k}_{ij'} \geq \alpha$, $\delta \leq q_{i_j} \leq \gamma$ *and* $\|\boldsymbol{w}\| \leq \sigma$. *Denote* $\mathcal{O}(\boldsymbol{w}) = 0$ *as the optimal condition. The magnitude of its change* $\Delta\mathcal{O}$ *due to the kernel perturbations is bounded by*

$$|\Delta\mathcal{O}| \leq C_2 \cdot \eta, \tag{7}$$

*where the constant* $C_2 = \left( (o-1)\delta + o\gamma^{o-1}\delta^o + o \right) \alpha\sigma(1 + \gamma N)$.

This result demonstrates that the optimal solution of UMKL-G is robust to small perturbations in the kernel values, ensuring stability even under minor fluctuations in kernel computation.

Before assessing the generalization of UMKL-G, we first define uniform stability for the algorithm. Let $\mathcal{G} \in \mathcal{X}^N$ be a training set of size $N$. Our algorithm is symmetric with respect to $\mathcal{G}$, meaning it does not depend on the order of elements in $\mathcal{G}$. Thus, the modified training set $\mathcal{G}^{\backslash r}$ is created by removing any $r$-th element from $\mathcal{G}$, where $r \in \{1, \cdots, N\}$. Specifically, we denote $\mathcal{G}^{\backslash r} = \{G_1, \ldots, G_{r-1}, G_{r+1}, \ldots, G_N\}$. Since the loss function $\mathcal{L}^{(o)}$ is Lipschitz continuous, as a consequence of Theorem 3, UMKL-G is also uniformly $\omega$-stable, as shown below:

**Theorem 5.** *Denote $A_\mathcal{G}$ as the output of our unsupervised learning algorithm UMKL-G after training on $\mathcal{G}$. UMKL-G is uniformly $\omega$-stable with respect to the loss function $\mathcal{L}^{(o)}$ if for any $G_i \in \mathcal{X}$, the following holds:*

$$\forall \mathcal{G} \in \mathcal{X}^N, \max_{i=1,\cdots,N} \left| \mathcal{L}^{(o)}(G_i, A_\mathcal{G}) - \mathcal{L}^{(o)}(G_i, A_{\mathcal{G}\backslash r}) \right| \leq \omega. \tag{8}$$

Using UMKL-G's uniform $\omega$-stability and applying Corollary 1 from Abou-Moustafa & Schuurmans (2015), we derive the following generalization bounds for our algorithm:

**Theorem 6.** *Denote $A$ as the algorithm UMKL-G, which is uniformly $\omega$-stable, $\forall G \in \mathcal{X}$, and $\forall \mathcal{G} \in \mathcal{G}^N$. Then, for any $N \geq 1$, and any $\delta \in (0,1)$, the following bounds hold with probability at least $1 - \delta$ over any $\mathcal{G}$,*

$$(i) \ R(A_\mathcal{G}) \leq \hat{R}_{EMP}(A_\mathcal{G}) + 2\omega + (4N\omega + c)\sqrt{\frac{\log(1/\delta)}{2N}}, \tag{9}$$

$$(ii) \ R(A_\mathcal{G}) \leq \hat{R}_{LOO}(A_\mathcal{G}) + \omega + (4N\omega + c)\sqrt{\frac{\log(1/\delta)}{2N}}, \tag{10}$$

*where $\hat{R}_{LOO}(A_\mathcal{G}) = \frac{1}{N}\sum_{i=1}^N \mathcal{L}^{(o)}(G_i, A_{\mathcal{G}\backslash i}(G_i))$, is the leave-one-out (LOO) error estimate.*

These bounds provide a guarantee that UMKL-G generalizes well from the training set to unseen data, ensuring consistent performance even in the absence of labeled data. With the aforementioned theoretical foundations of UMKL-G established, it is also important to understand how our method compares to existing approaches in the field of unsupervised multiple kernel learning.

### 4.7 CONNECTION TO BASELINES (PARTIAL)

Our proposed method shares a foundational goal with previous methods, such as UMKL (Zhuang et al., 2011) and sparse-UMKL (Mariette & Villa-Vialaneix, 2018), which is to preserve the local geometry of the data. These methods have laid significant groundwork in unsupervised multiple kernel learning. In UMKL, the authors propose two main principles: first, for each data point $\boldsymbol{x}_i$, the optimal kernel should minimize the approximation error $\|\boldsymbol{x}_i - \sum_j k_{ij}\boldsymbol{x}_j\|^2$; second, the method should minimize the distortion over all training data, $\sum_{ij} k_{ij}\|\boldsymbol{x}_i - \mathbf{x}_j\|^2$, where $k_{ij} = k(\boldsymbol{x}_i, \boldsymbol{x}_j)$. Similarly, sparse-UMKL aims to approximately preserve the local geometry of the data by building $k$-nearest neighbor graphs for each kernel and defining a weight matrix based on these graphs. Despite their innovative approaches, both UMKL and sparse-UMKL have limitations in handling graph data and achieving satisfactory empirical performances (see Section 5 for details).

| Feature | UMKL | sparse-UMKL | UMKL-G (Ours) |
|---|---|---|---|
| Objective Function | $\min_{\mu,D} \frac{1}{2}\|X(I - K \circ D)\|_F^2$ $+\gamma_1 \mathrm{tr}(K \circ D \circ M) + \gamma_2\|D\|_{1,1}$ | $\min_{\mathbf{b}} \mathrm{tr}(\mathbf{W}K) + \lambda\|\mathbf{b}\|_1,$ $K = \sum_{m=1}^M b_m K_m$ | $\min_{\mathbf{w}} L^{(o)} = \mathrm{KL}(Q\|P),$ $Q_{ij} = \frac{\bar{k}_{ij}}{\sum_{j'} \bar{k}_{ij'}}, \quad P_{ij} = \frac{\bar{k}_{ij}^o}{\sum_{j'} \bar{k}_{ij'}^o}$ |
| Beyond Euclidean | ✗ | ✓ | ✓ |
| Global Topology | ✗ | ✗ | ✓ |
| Theoretical Guarantees | ✓ | ✗ | ✓ |
| Topology Preservation | Local reconstruction ($D$) | k-NN graph heuristics ($\mathbf{W}$) | Ordinal relationships |
| Algorithm | Alternating minimization | Quadratic programming solver | KL divergence |
| Complexity | $O(I \cdot (MN^2 + N^3))$ | $O(I \cdot (MN^2 \log N + M^3))$ | $O(I \cdot (MN^2 + M \log M))$ |

Table 1: Comparison of UMKL, sparse-UMKL, and UMKL-G.

In Table 1, we highlight the key features of UMKL-G in comparison to the baselines. A detailed analysis can be found in Appendix E.

## 5 EXPERIMENTS

In this study, we evaluate UMKL-G in a common unsupervised task—graph-level clustering.

## 5.1 DATASETS AND SETUPS

**Datasets.** We include eight benchmark datasets in our experiments, encompassing diverse types of graph data (Kersting et al., 2016). These datasets are BZR, COX2, DD, DHFR, ENZYMES, IMDB-BINARY, MUTAG, and PTC_FM, as described in Table 5 (Appendix F). Each dataset presents unique characteristics, suitable for testing the robustness and generalizability of our approach.

**Configurations.** We evaluate our algorithm exhaustively on six state-of-the-art graph kernels with various hyperparameters, as shown in Table 6 (Appendix F). For each dataset, the number of base kernels is 30, i.e., $M = 30$. Besides, we set the power hyperparameters $o$ to $\{2, 3, 4\}$ to inspect its concentration effect. As for the initial weights, we set them uniformly as $1/M$ by default. For the baseline methods, UMKL and sparse-UMKL, we trained a Graph Convolutional Network (GCN) (Kipf & Welling, 2016) with 10 layers to represent the graphs in vector form. To learn the composite kernel, we explored two approaches: (1) pre-training the GCN to produce fixed graph representations, followed by freezing the GCN parameters during kernel weight updates, and (2) co-training the GCN and kernel weights simultaneously. In the following sections, we report the superior performance obtained from our two approaches. We also consider a set of neighborhood hyperparameters $k \in \{10, 50, 100\}$ for sparse-UMKL. In all experiments, we use the Adam optimizer (Kingma & Ba, 2015) at 1e-3 initial learning rate and 1e-4 weight decay. The total epochs is set to 500 across all methods. We also include a simple baseline, AverageMKL, where each weight is set to $1/M$.

**Evaluation Metrics.** We evaluate the empirical performance of our method with three commonly used metrics: clustering accuracy (ACC), Normalized Mutual Information (NMI), and Adjusted Rand Index (ARI), as defined in Appendix F. These metrics provide a comprehensive evaluation of the clustering performance and the preservation of the topology.

## 5.2 RESULTS

Overall, we conducted two experiments, comparing (a) the proposed UMKL-G algorithm with baselines, (b) the proposed UMKL-G algorithm and base graph kernels. All of the findings support the superiority of our method, as highlighted below. Please see the supplements in Appendix G.

**UMKL-G consistently outperforms the baseline methods across all datasets.** Overall, our proposed method demonstrates remarkable improvements in clustering performance. The performance comparison presented in Table 2 underscores the effectiveness of UMKL-G across all metrics. This is evident across various domains, including chemical compounds, biological data, social networks, and protein structures. By focusing on ordinal relationships among graphs, UMKL-G more effectively preserves the intrinsic relationships within the data. This approach maintains the relative ordering of similarities, capturing both local and global structural features more accurately than the baseline methods.

Table 2: Comparison with Baseline Methods. *The best score is in bold. The second best is underlined.*

| Method | BZR | | | COX2 | | | DD | | | DHFR | | |
|---|---|---|---|---|---|---|---|---|---|---|---|---|
| | ACC | NMI | ARI | ACC | NMI | ARI | ACC | NMI | ARI | ACC | NMI | ARI |
| AverageMKL | 0.7341 | 0.0041 | 0.0307 | 0.6167 | 0.0000 | -0.0016 | 0.5764 | 0.0060 | 0.0172 | 0.6495 | 0.0000 | -0.0021 |
| UMKL | 0.7341 | 0.0041 | 0.0307 | 0.6167 | 0.0000 | -0.0016 | 0.5764 | 0.0060 | 0.0172 | 0.6495 | 0.0000 | -0.0021 |
| sparse-UMKL ($k = 10$) | 0.7400 | 0.0040 | 0.0299 | 0.6200 | 0.0001 | -0.0010 | 0.5750 | 0.0059 | 0.0170 | 0.6480 | 0.0001 | -0.0020 |
| sparse-UMKL ($k = 50$) | 0.7415 | 0.0042 | 0.0305 | 0.6180 | 0.0000 | -0.0015 | 0.5770 | 0.0061 | 0.0175 | 0.6498 | 0.0000 | -0.0022 |
| sparse-UMKL ($k = 100$) | 0.7420 | 0.0041 | 0.0306 | 0.6175 | 0.0000 | -0.0016 | 0.5768 | 0.0060 | 0.0172 | 0.6592 | 0.0000 | -0.0021 |
| UMKL-G | **0.9432** | **0.0279** | **0.0812** | **0.8009** | **0.0045** | **0.0247** | **0.5815** | **0.0098** | **0.0224** | **0.6984** | **0.0111** | **0.0180** |

| Method | ENZYMES | | | IMDB-BINARY | | | MUTAG | | | PTC_FM | | |
|---|---|---|---|---|---|---|---|---|---|---|---|---|
| | ACC | NMI | ARI | ACC | NMI | ARI | ACC | NMI | ARI | ACC | NMI | ARI |
| AverageMKL | 0.2617 | 0.0539 | 0.0220 | 0.5470 | 0.0152 | 0.0083 | 0.5585 | 0.1468 | 0.1946 | 0.8825 | 0.0208 | 0.0343 |
| UMKL | 0.2567 | 0.0517 | 0.0199 | 0.5470 | 0.0152 | 0.0083 | 0.5585 | 0.1469 | 0.1947 | 0.8729 | 0.0208 | 0.0343 |
| sparse-UMKL ($k = 10$) | 0.2570 | 0.0520 | 0.0201 | 0.5485 | 0.0153 | 0.0084 | 0.5590 | 0.1475 | 0.1950 | 0.8320 | 0.0210 | 0.0345 |
| sparse-UMKL ($k = 50$) | 0.2580 | 0.0518 | 0.0200 | 0.5475 | 0.0154 | 0.0085 | 0.5595 | 0.1470 | 0.1948 | 0.8373 | 0.0211 | 0.0344 |
| sparse-UMKL ($k = 100$) | 0.2575 | 0.0521 | 0.0198 | 0.5480 | 0.0151 | 0.0082 | 0.5588 | 0.1468 | 0.1946 | 0.8528 | 0.0209 | 0.0342 |
| UMKL-G | **0.2983** | **0.0648** | **0.0399** | **0.5590** | **0.0159** | **0.0132** | **0.8455** | **0.2950** | **0.3389** | **0.8825** | **0.0394** | **0.0637** |

**UMKL-G can beat the base graph kernels across all metrics.** As demonstrated in Figure 2a, UMKL-G, indicated by the dashed grey lines, consistently surpasses the base graph kernels on the

DHFR dataset. The base kernels, such as VH, SP, and various iterations of WLOA and WL, exhibit varying degrees of performance. While some kernels achieve relatively high accuracy, they fall short in other metrics such as NMI and ARI, indicating an imbalance in capturing the overall data structure. This trend is consistent across other benchmark datasets, as Figures 3-8 show.

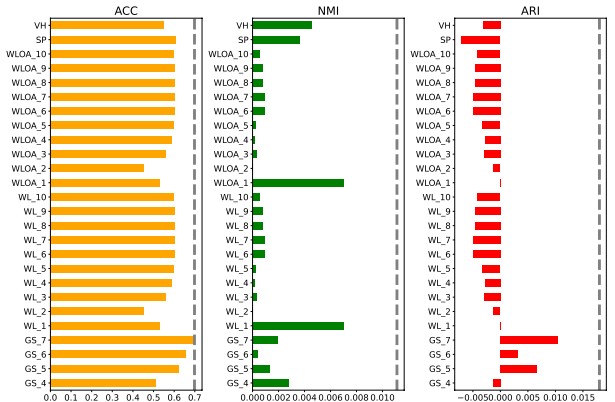
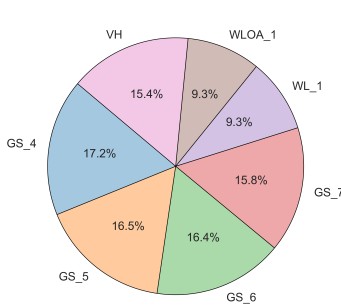

(a) Comparison with Base Graph Kernels. *The bar plots represent the performance metrics for different kernels. The dashed grey lines indicate the performances of UMKL-G.*

(b) Learned Kernel Weights of UMKL-G.

Figure 2: Performance on the DHFR dataset. *Kernel names are shown with hyperparameters.*

**UMKL-G can automatically select graph kernels and their hyperparameters.** This capability is evident from the learned weights, which indicate the relative importance of each kernel in the clustering task. As shown in Figure 2b, by assigning higher weights to more relevant kernels, UMKL-G effectively prioritizes the kernels that best capture the underlying structure of the data. This automatic weighting and selection process not only streamlines the model tuning but also enhances the performance by leveraging the strengths of multiple kernels as shown in the previous finding.

**UMKL-G performances are insensitive to the hyperparameter ($o = 2$ is enough).** We provide full experimental results in Figure 10 in Appendix G, where UMKL-G reaches a stable performance across all power $o$. The concentration effect is insensitive to the hyperparameter setting.

## 6 CONCLUSION AND FUTURE WORK

In this paper, we propose UMKL-G, an unsupervised algorithm for combining multiple graph kernels by focusing on ordinal relationships to preserve the topological structure between graphs. Without the need to learn a separate graph representation, UMKL-G leverages the concentration effect of powered kernels to maintain the relative ordering of similarities among graphs, effectively capturing both local and global structural features. Moreover, we provide theoretical analysis guarantees to ensure the convergence, robustness, and generalization of our method, reinforcing its reliability in practical applications. Empirically, UMKL-G significantly outperforms two baseline algorithms adapted from GCNs, highlighting its robustness and efficacy in handling graph data where the baselines fall short.

In the future, our work can be extended to other domains by developing UMKL-X, where "X" represents various types of unstructured data. This extension would involve integrating different types of similarity measures suitable for diverse data structures, thereby broadening the applicability and impact of our approach. Additionally, exploring the scalability and efficiency of UMKL-G on larger datasets and in real-world applications could further validate and enhance its utility.

ACKNOWLEDGEMENTS

We thank all anonymous reviewers for their valuable feedback and constructive suggestions. This work was partially supported by the Google PhD Fellowship awarded to Yan Sun.

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

# Appendices

## A  MOTIVATION AND INTUITION BEHIND $P$

To address the intuition behind why the target $P$ provides a more accurate representation of the data's inherent geometry, we include the following example. As explained in Theorem 2, the concentration effect ensures that $P$ has lower entropy compared to $Q$, as illustrated in Figure 1, where $P$ (red points) spreads further outside $Q$ (blue points). This implies that $P$ focuses more on the most meaningful connections and identifies nearest neighbors in a soft manner. By raising kernel values to a power $o > 1$, $P$ magnifies the differences between highly similar and less similar graphs, which is crucial for capturing the essential structure of the data.

To make this intuition clear, consider an example with $N = 5$ graphs and their pairwise kernel similarities computed using a graph kernel (e.g., Weisfeiler-Lehman kernel). The symmetric kernel matrix $\tilde{K}$ is defined as:

$$\tilde{K} = \begin{pmatrix} 1.0 & 0.8 & 0.3 & 0.2 & 0.1 \\ 0.8 & 1.0 & 0.4 & 0.3 & 0.2 \\ 0.3 & 0.4 & 1.0 & 0.7 & 0.6 \\ 0.2 & 0.3 & 0.7 & 1.0 & 0.9 \\ 0.1 & 0.2 & 0.6 & 0.9 & 1.0 \end{pmatrix} \tag{11}$$

where each element $\tilde{k}_{ij}$ represents the similarity between graphs $G_i$ and $G_j$.

To emphasize differences in similarities, we raise the kernel values to a power $o = 5$. For $G_1$, the original probability distribution $\mathbf{q}_1 = (q_{1_1}, q_{1_2}, q_{1_3}, q_{1_4}, q_{1_5})$ is computed as $\mathbf{q}_1 = \frac{\tilde{k}_{1j}}{\sum_{j=1}^{5} \tilde{k}_{1j}} = (0.4167, 0.3333, 0.1250, 0.0833, 0.0417)$. After raising the kernel values to the power $o = 5$, the new powered distribution $\mathbf{p}_1^{(5)} = (p_{1_1}^{(5)}, p_{1_2}^{(5)}, p_{1_3}^{(5)}, p_{1_4}^{(5)}, p_{1_5}^{(5)})$ becomes $\mathbf{p}_1^{(5)} = \frac{\tilde{k}_{1j}^o}{\sum_{j=1}^{5} \tilde{k}_{1j}^o} = (0.7516, 0.2463, 0.0018, 0.0002, 0.0000)$.

In $\mathbf{q}_1$, the probabilities are distributed more evenly among the graphs, whereas in $\mathbf{p}_1^{(5)}$, the probability is heavily concentrated on $p_{1_1}^{(5)}$ and $p_{1_2}^{(5)}$. This highlights how $P$ helps $G_1$ find its nearest neighbor $G_2$ by reducing the influence of less similar graphs $(G_3, G_4, G_5)$.

In analogy, $P$ creates a "soft cut" of the fully connected network among all graphs, accurately representing the data's inherent geometry. Importantly, the relative ordering of similarities remains unchanged: $p_{1_1}^{(5)} > p_{1_2}^{(5)} > p_{1_3}^{(5)} > p_{1_4}^{(5)} > p_{1_5}^{(5)}$, $q_{1_1} > q_{1_2} > q_{1_3} > q_{1_4} > q_{1_5}$, and $\tilde{k}_{11} > \tilde{k}_{12} > \tilde{k}_{13} > \tilde{k}_{14} > \tilde{k}_{15}$.

## B  PROOF OF THEOREMS

**Proof of Theorem 1**

*Proof.* For a given $i$, if $\tilde{k}_{ij} \in [0,1]$ represents the similarity between graphs $G_i$ and $G_j$, then for $q_{i_j}$, the distribution is $q_{i_j} = \frac{\tilde{k}_{ij}}{\sum_{j'} \tilde{k}_{ij'}}$, and for $p_{i_j}^{(o)}$ with $o > 1$, the distribution is: $p_{i_j}^{(o)} = \frac{\tilde{k}_{ij}^o}{\sum_{j'} \tilde{k}_{ij'}^o}$. For any ordinal relationship $\tilde{k}_{ij} > \tilde{k}_{ir}$ and any power $o > 1$, we have $\tilde{k}_{ij}^o > \tilde{k}_{ir}^o$. By normalizing,

$$\frac{\tilde{k}_{ij}^o}{\sum_{j'} \tilde{k}_{ij'}^o} > \frac{\tilde{k}_{ir}^o}{\sum_{j'} \tilde{k}_{ij'}^o},$$

which implies that $p_{i_j}^{(o)} > p_{i_r}^{(o)}$. Hence the relationship between $(G_i, G_j)$ and $(G_i, G_r)$ as reflected in $\tilde{k}_{ij} > \tilde{k}_{ir}$ is preserved by the probabilities $p_{i_j}^{(o)} > p_{i_r}^{(o)}$.  □

**Proof of Theorem 2**

*Proof.* The entropy of the distribution $\boldsymbol{q}_i$ is given by

$$H(\boldsymbol{q}_i) = -\sum_{j=1}^N q_{i_j} \log q_{i_j}. \tag{12}$$

Note that $q_{i_j} = \frac{\tilde{k}_{ij}}{\sum_{j'} \tilde{k}_{ij'}}$ and $p_{i_j}^{(o)} = \frac{\tilde{k}_{ij}^o}{\sum_{j'} \tilde{k}_{ij'}^o}$ for some integer $o \in \mathbb{N}^+$. From the definition of $q_{i_j}$, we can write $\tilde{k}_{ij} = q_{i_j} \sum_{j'} \tilde{k}_{ij'}$ and thus

$$\begin{aligned}
p_{i_j}^{(o)} &= \frac{(q_{i_j} \sum_{j'} \tilde{k}_{ij'})^o}{\sum_{j'} (q_{i_{j'}} \sum_{j''} \tilde{k}_{ij''})^o} \\
&= \frac{q_{i_j}^o (\sum_{j'} \tilde{k}_{ij'})^o}{\sum_{j'} q_{i_{j'}}^o (\sum_{j''} \tilde{k}_{ij''})^o} \\
&= \frac{q_{i_j}^o}{\sum_{j'} q_{i_{j'}}^o}.
\end{aligned} \tag{13}$$

Thus, the entropy of $\boldsymbol{p}_i^{(o)}$ can be expressed as

$$\begin{aligned}
H(\boldsymbol{p}_i^{(o)}) &= -\sum_{j=1}^N \frac{q_{i_j}^o}{\sum_{j'} q_{i_{j'}}^o} \log \left( \frac{q_{i_j}^o}{\sum_{j'} q_{i_{j'}}^o} \right) \\
&= \left( -o \sum_{j=1}^N \frac{q_{i_j}^o}{S_o} \log q_{i_j} \right) + \log S_o
\end{aligned} \tag{14}$$

where $S_o = \sum_{j=1}^N q_{i_j}^o$. To analyze the behavior of the entropy for $o \geq 1$, we will compute the first-order derivative of $H(\boldsymbol{p}_i^{(o)})$ with respect to $o$:

$$\frac{\partial}{\partial o} H(\boldsymbol{p}_i^{(o)}) = \frac{\partial}{\partial o} \left( -o \sum_{j=1}^{N} \frac{q_{i_j}^o}{S_o} \log q_{i_j} \right) + \frac{\partial}{\partial o} \log S_o$$

$$= -\sum_{j=1}^{N} \frac{q_{i_j}^o}{S_o} \log q_{i_j} - o \sum_{j=1}^{N} \left( \frac{q_{i_j}^o \log q_{i_j}}{S_o} - \frac{q_{i_j}^o \sum_{j'=1}^{N} q_{i_{j'}}^o \log q_{i_{j'}}}{S_o^2} \right) \log q_{i_j} + \frac{1}{S_o} \sum_{j=1}^{N} q_{i_j}^o \log q_{i_j}$$

$$= -o \sum_{j=1}^{N} \left( \frac{q_{i_j}^o \log q_{i_j}}{S_o} - \frac{q_{i_j}^o \sum_{j'=1}^{N} q_{i_{j'}}^o \log q_{i_{j'}}}{S_o^2} \right) \log q_{i_j}$$

$$= -\frac{o}{S_o^2} \left( \sum_{j'=1}^{N} q_{i_{j'}}^o \sum_{j=1}^{N} q_{i_j}^o \left( \log q_{i_j} \right)^2 - \sum_{j=1}^{N} q_{i_j}^o \log q_{i_j} \sum_{j'=1}^{N} q_{i_{j'}}^o \log q_{i_{j'}} \right).$$

$$(15)$$

By the Cauchy-Schwarz inequality, we have

$$\left( \sum_{j=1}^{N} q_{i_j}^o \log q_{i_j} \right)^2 < \left( \sum_{j'=1}^{N} q_{i_{j'}}^o \right) \left( \sum_{j=1}^{N} q_{i_j}^o \left( \log q_{i_j} \right)^2 \right). \tag{16}$$

Thus, for $o > 1$, the derivative

$$\frac{\partial}{\partial o} H(\boldsymbol{p}_i^{(o)}) < 0, \tag{17}$$

meaning that the entropy decreases as $o$ increases. Since $H(\boldsymbol{p}_i^{(o)}) = H(\boldsymbol{q}_i)$ when $o = 1$, the following inequality holds for any graph $G_i$ for $o > 1$,

$$H(\boldsymbol{p}_i^{(o)}) < H(\boldsymbol{q}_i). \tag{18}$$

$$\square$$

**Proof of Theorem 3**

*Proof.* As $p_{i_j}^{(o)} = \frac{q_{i_j}^o}{\sum_{j'} q_{i_{j'}}^o}$ (see Eq. 13), the loss function $\mathcal{L}^{(o)}(\boldsymbol{w})$ can be represented as

$$\mathcal{L}^{(o)}(\boldsymbol{w}) = \sum_{i,j} \left( q_{i_j} \log q_{i_j} - q_{i_j} \log \left( \frac{q_{i_j}^o}{\sum_{j'} q_{i_{j'}}^o} \right) \right)$$

$$= \sum_{i,j} \left( -q_{i_j} \log q_{i_j}^{o-1} + q_{i_j} \log \sum_{j'} q_{i_{j'}}^o \right). \tag{19}$$

Its gradient with respect to the learnable parameters $\boldsymbol{w}$ is

$$\nabla_{\boldsymbol{w}} \mathcal{L}^{(o)} = \sum_{i,j} \left( -\nabla_{\boldsymbol{w}} \left( q_{i_j} \log q_{i_j}^{o-1} \right) + \nabla_{\boldsymbol{w}} \left( q_{i_j} \log \sum_{j'} q_{i_{j'}}^o \right) \right)$$

$$= \sum_{i,j} \left( -(o-1)(\log q_{i_j} + 1) + \log \sum_{j'} q_{i_{j'}}^o + q_{i_j} \frac{1}{\sum_{j'} q_{i_{j'}}^o} \sum_{j'} o q_{i_{j'}}^{o-1} \right) \nabla_{\boldsymbol{w}} q_{i_j} \quad (20)$$

$$= \sum_{i,j} \left( -(o-1)(\log q_{i_j} + 1) + \log \sum_{j'} q_{i_{j'}}^o + o \cdot q_{i_j} \frac{\sum_{j'} q_{i_{j'}}^{o-1}}{\sum_{j'} q_{i_{j'}}^o} \right) \nabla_{\boldsymbol{w}} q_{i_j}.$$

Given $\tilde{k}_{ij} = \boldsymbol{w}^\top \boldsymbol{k}_{ij}$, we denote $q_{i_j} := q_{i_j}(\boldsymbol{w}) = \frac{\boldsymbol{w}^\top \boldsymbol{k}_{ij}}{Z_i(\boldsymbol{w})}$, where $Z_i(\boldsymbol{w}) = \sum_{j'} \tilde{k}_{ij'} = \sum_{j'} \boldsymbol{w}^\top \boldsymbol{k}_{ij'}$. The gradient of $q_{i_j}$ with respect to $\boldsymbol{w}$ is

$$
\begin{aligned}
\nabla_{\boldsymbol{w}} q_{i_j} &= \frac{\boldsymbol{k}_{ij} Z_i(\boldsymbol{w}) - \boldsymbol{w}^\top \boldsymbol{k}_{ij} \sum_{j'} \boldsymbol{k}_{ij'}}{[Z_i(\boldsymbol{w})]^2} \\
&= \frac{\boldsymbol{k}_{ij}}{Z_i(\boldsymbol{w})} - q_{i_j} \frac{\sum_{j'} \boldsymbol{k}_{ij'}}{Z_i(\boldsymbol{w})} \\
&= Z_i(\boldsymbol{w})^{-1} \left( \boldsymbol{k}_{ij} - q_{i_j} \sum_{j'} \boldsymbol{k}_{ij'} \right).
\end{aligned}
\tag{21}
$$

Since we are interested in bounding the difference between the gradient of $\mathcal{L}^{(o)}$ at $\boldsymbol{w}$ and $\boldsymbol{w}'$, i.e., $\|\nabla_{\boldsymbol{w}} \mathcal{L}^{(o)}(\boldsymbol{w}) - \nabla_{\boldsymbol{w}} \mathcal{L}^{(o)}(\boldsymbol{w}')\|$, in which each term involves a sum of the form $\nabla_{\boldsymbol{w}} \mathcal{L}^{(o)}(\boldsymbol{w}) = \sum_{i,j} T_{i,j}(\boldsymbol{w})$, where

$$
T_{i,j}(\boldsymbol{w}) = \left( -(o-1) \left( \frac{\log q_{i_j}(\boldsymbol{w})}{Z_i(\boldsymbol{w})} + \frac{1}{Z_i(\boldsymbol{w})} \right) + \frac{\log \sum_{j'} q_{i_{j'}}^o(\boldsymbol{w})}{Z_i(\boldsymbol{w})} + o \cdot \frac{q_{i_j}(\boldsymbol{w})}{Z_i(\boldsymbol{w})} \frac{\sum_{j'} q_{i_{j'}}^{o-1}(\boldsymbol{w})}{\sum_{j'} q_{i_{j'}}^o(\boldsymbol{w})} \right) \left( \boldsymbol{k}_{ij} - q_{i_j}(\boldsymbol{w}) \sum_{j'} \boldsymbol{k}_{ij'} \right).
\tag{22}
$$

① **Let's focus on bounding the difference in the term** $\frac{\log q_{i_j}(\boldsymbol{w})}{Z_i(\boldsymbol{w})}$, which is

$$
\begin{aligned}
\frac{\log q_{i_j}(\boldsymbol{w})}{Z_i(\boldsymbol{w})} - \frac{\log q_{i_j}(\boldsymbol{w}')}{Z_i(\boldsymbol{w}')} &= \left( \frac{\log q_{i_j}(\boldsymbol{w})}{Z_i(\boldsymbol{w})} - \frac{\log q_{i_j}(\boldsymbol{w})}{Z_i(\boldsymbol{w}')} \right) + \left( \frac{\log q_{i_j}(\boldsymbol{w})}{Z_i(\boldsymbol{w}')} - \frac{\log q_{i_j}(\boldsymbol{w}')}{Z_i(\boldsymbol{w}')} \right) \\
&= \log q_{i_j}(\boldsymbol{w}) \left( \frac{1}{Z_i(\boldsymbol{w})} - \frac{1}{Z_i(\boldsymbol{w}')} \right) + \frac{1}{Z_i(\boldsymbol{w}')} \left( \log q_{i_j}(\boldsymbol{w}) - \log q_{i_j}(\boldsymbol{w}') \right) \\
&= \log q_{i_j}(\boldsymbol{w}) \frac{Z_i(\boldsymbol{w}') - Z_i(\boldsymbol{w})}{Z_i(\boldsymbol{w}) Z_i(\boldsymbol{w}')} + \frac{1}{Z_i(\boldsymbol{w}')} \left( \log q_{i_j}(\boldsymbol{w}) - \log q_{i_j}(\boldsymbol{w}') \right).
\end{aligned}
\tag{23}
$$

Since $Z_i(\boldsymbol{w}') - Z_i(\boldsymbol{w}) = \sum_{j'} \left( \boldsymbol{w}'^\top \boldsymbol{k}_{ij'} - \boldsymbol{w}^\top \boldsymbol{k}_{ij'} \right)$, we have

$$
|Z_i(\boldsymbol{w}') - Z_i(\boldsymbol{w})| \leq \|\boldsymbol{w} - \boldsymbol{w}'\| \sum_{j'} \|\boldsymbol{k}_{ij'}\| \leq \|\boldsymbol{w} - \boldsymbol{w}'\| N K_{\max},
\tag{24}
$$

where $N$ is the number of graphs and $K_{\max}$ is the maximum norm of $\|\boldsymbol{k}_{ij}\|$ among all $i, j$. Therefore, we have

$$
\left| \log q_{i_j}(\boldsymbol{w}) \frac{Z_i(\boldsymbol{w}') - Z_i(\boldsymbol{w})}{Z_i(\boldsymbol{w}) Z_i(\boldsymbol{w}')} \right| \leq \frac{(\log \delta^{-1}) N K_{\max}}{\alpha^2} \|\boldsymbol{w} - \boldsymbol{w}'\|,
\tag{25}
$$

where $\alpha > 0$ is the lower bound for all $Z_i(\boldsymbol{w})$ and $|\log q_{i_j}| \leq \log \delta^{-1}$, where $\delta > 0$ is the lower bound for all $q_{i_j}(\boldsymbol{w})$. According to the Mean Value Theorem, we have

$$
\left| \log q_{i_j}(\boldsymbol{w}) - \log q_{i_j}(\boldsymbol{w}') \right| \leq \frac{\left| q_{i_j}(\boldsymbol{w}) - q_{i_j}(\boldsymbol{w}') \right|}{\delta}.
\tag{26}
$$

Recall that $q_{i_j}(\boldsymbol{w}) = \frac{\boldsymbol{w}^\top \boldsymbol{k}_{ij}}{Z_i(\boldsymbol{w})}$. The difference between $q_{i_j}(\boldsymbol{w})$ and $q_{i_j}(\boldsymbol{w}')$ is

$$
q_{i_j}(\boldsymbol{w}) - q_{i_j}(\boldsymbol{w}') = \frac{\boldsymbol{w}^\top \boldsymbol{k}_{ij} Z_i(\boldsymbol{w}') - \boldsymbol{w}'^\top \boldsymbol{k}_{ij} Z_i(\boldsymbol{w})}{Z_i(\boldsymbol{w}) Z_i(\boldsymbol{w}')}
\tag{27}
$$

For the numerator, since $\boldsymbol{w}'^\top \boldsymbol{k}_{ij} = \tilde{k}_{ij} \leq 1$, we have

$$
\begin{aligned}
\left| \boldsymbol{w}^\top \boldsymbol{k}_{ij} Z_i(\boldsymbol{w}') - \boldsymbol{w}'^\top \boldsymbol{k}_{ij} Z_i(\boldsymbol{w}) \right| &\leq \left| \boldsymbol{w}^\top \boldsymbol{k}_{ij} - \boldsymbol{w}'^\top \boldsymbol{k}_{ij} \right| Z_i(\boldsymbol{w}') + \boldsymbol{w}'^\top \boldsymbol{k}_{ij} |Z_i(\boldsymbol{w}') - Z_i(\boldsymbol{w})| \\
&\leq K_{\max} \|\boldsymbol{w} - \boldsymbol{w}'\| \beta + \|\boldsymbol{w} - \boldsymbol{w}'\| N K_{\max},
\end{aligned}
\tag{28}
$$

where $\beta$ is the upper bound for all $Z_i(\boldsymbol{w})$.

Thus, we have

$$\left|q_{i_j}(\boldsymbol{w}) - q_{i_j}(\boldsymbol{w}')\right| \leq \frac{\beta + N}{\alpha^2} K_{\max} \|\boldsymbol{w} - \boldsymbol{w}'\|. \tag{29}$$

The upper bound of the full second term is

$$\left|\frac{1}{Z_i(\boldsymbol{w}')} \left(\log q_{i_j}(\boldsymbol{w}) - \log q_{i_j}(\boldsymbol{w}')\right)\right| \leq \frac{\beta + N}{\alpha^3 \delta} K_{\max} \|\boldsymbol{w} - \boldsymbol{w}'\|. \tag{30}$$

To sum up ①, we denote $\alpha^{-2}N$ as $\psi_1$ and denote $\frac{\beta+N}{\alpha^3}$ as $\psi_2$. We have

$$\left|\frac{\log q_{i_j}(\boldsymbol{w})}{Z_i(\boldsymbol{w})} - \frac{\log q_{i_j}(\boldsymbol{w}')}{Z_i(\boldsymbol{w}')}\right| \leq \left(\psi_1 \left(\log \delta^{-1}\right) + \psi_2 \delta^{-1}\right) K_{\max} \|\boldsymbol{w} - \boldsymbol{w}'\|. \tag{31}$$

**② We have the bound for the difference in the term $\frac{1}{Z_i(\boldsymbol{w})}$ as below.**

$$\begin{aligned}
\left(\frac{1}{Z_i(\boldsymbol{w})} - \frac{1}{Z_i(\boldsymbol{w}')}\right) &= \frac{Z_i(\boldsymbol{w}') - Z_i(\boldsymbol{w})}{Z_i(\boldsymbol{w})Z_i(\boldsymbol{w}')} \\
&\leq \alpha^{-2} \|\boldsymbol{w} - \boldsymbol{w}'\| N K_{\max} \\
&= \psi_1 K_{\max} \|\boldsymbol{w} - \boldsymbol{w}'\|.
\end{aligned} \tag{32}$$

**③ Next, we focus on bounding the difference in the term $\frac{\log \sum_{j'} q_{i_{j'}}^o(\boldsymbol{w})}{Z_i(\boldsymbol{w})}$, which is**

$$\frac{\log \sum_{j'} q_{i_{j'}}^o(\boldsymbol{w})}{Z_i(\boldsymbol{w})} - \frac{\log \sum_{j'} q_{i_{j'}}^o(\boldsymbol{w}')}{Z_i(\boldsymbol{w}')} = \left(\frac{\log \sum_{j'} q_{i_{j'}}^o(\boldsymbol{w})}{Z_i(\boldsymbol{w})} - \frac{\log \sum_{j'} q_{i_{j'}}^o(\boldsymbol{w})}{Z_i(\boldsymbol{w}')}\right) + \left(\frac{\log \sum_{j'} q_{i_{j'}}^o(\boldsymbol{w})}{Z_i(\boldsymbol{w}')} - \frac{\log \sum_{j'} q_{i_{j'}}^o(\boldsymbol{w}')}{Z_i(\boldsymbol{w}')}\right) \tag{33}$$

As $\left|\log \sum_{j'} q_{i_{j'}}^o\right| \leq \log(N\delta^{-o})$, the first term is bounded as

$$\left|\frac{\log \sum_{j'} q_{i_{j'}}^o(\boldsymbol{w})}{Z_i(\boldsymbol{w})} - \frac{\log \sum_{j'} q_{i_{j'}}^o(\boldsymbol{w})}{Z_i(\boldsymbol{w}')}\right| \leq \frac{\log(N\delta^{-o}) N K_{\max}}{\alpha^2} \|\boldsymbol{w} - \boldsymbol{w}'\| \tag{34}$$

For the second term, we use the Mean Value Theorem to get

$$\begin{aligned}
\left|\frac{\log \sum_{j'} q_{i_{j'}}^o(\boldsymbol{w})}{Z_i(\boldsymbol{w}')} - \frac{\log \sum_{j'} q_{i_{j'}}^o(\boldsymbol{w}')}{Z_i(\boldsymbol{w}')}\right| &\leq \frac{o \left|\sum_{j'} q_{i_{j'}}^{o-1}\right| \left|q_{i_j}(\boldsymbol{w}) - q_{i_j}(\boldsymbol{w}')\right|}{\alpha N \delta^o} \\
&\leq \frac{o \gamma^{o-1} (\beta + N) K_{\max}}{\alpha^3 \delta^o} \|\boldsymbol{w} - \boldsymbol{w}'\|,
\end{aligned} \tag{35}$$

where $\gamma < 1$ is the upper bound of all $q_{i_j}$.

To sum up ③, we denote $\frac{\gamma}{\delta}$ as $\psi_3$. We have

$$\left|\frac{\log \sum_{j'} q_{i_{j'}}^o(\boldsymbol{w})}{Z_i(\boldsymbol{w})} - \frac{\log \sum_{j'} q_{i_{j'}}^o(\boldsymbol{w}')}{Z_i(\boldsymbol{w}')}\right| \leq \left(\psi_1 \log(N\delta^{-o}) + o\psi_2 \psi_3^{o-1} \delta^{-1}\right) K_{\max} \|\boldsymbol{w} - \boldsymbol{w}'\| \tag{36}$$

**④ Then, let's focus on bounding the difference in the term $\frac{q_{i_j}(\boldsymbol{w})}{Z_i(\boldsymbol{w})} \frac{\sum_{j'} q_{i_{j'}}^{o-1}(\boldsymbol{w})}{\sum_{j'} q_{i_{j'}}^o(\boldsymbol{w})}$, which is**

$$\begin{aligned}
&\frac{q_{i_j}(\boldsymbol{w})}{Z_i(\boldsymbol{w})} \frac{\sum_{j'} q_{i_{j'}}^{o-1}(\boldsymbol{w})}{\sum_{j'} q_{i_{j'}}^o(\boldsymbol{w})} - \frac{q_{i_j}(\boldsymbol{w}')}{Z_i(\boldsymbol{w}')} \frac{\sum_{j'} q_{i_{j'}}^{o-1}(\boldsymbol{w}')}{\sum_{j'} q_{i_{j'}}^o(\boldsymbol{w}')} \\
&= \left(\frac{q_{i_j}(\boldsymbol{w})}{Z_i(\boldsymbol{w})} - \frac{q_{i_j}(\boldsymbol{w}')}{Z_i(\boldsymbol{w}')}\right) \frac{\sum_{j'} q_{i_{j'}}^{o-1}(\boldsymbol{w})}{\sum_{j'} q_{i_{j'}}^o(\boldsymbol{w})} + \frac{q_{i_j}(\boldsymbol{w}')}{Z_i(\boldsymbol{w}')} \left(\frac{\sum_{j'} q_{i_{j'}}^{o-1}(\boldsymbol{w})}{\sum_{j'} q_{i_{j'}}^o(\boldsymbol{w})} - \frac{\sum_{j'} q_{i_{j'}}^{o-1}(\boldsymbol{w}')}{\sum_{j'} q_{i_{j'}}^o(\boldsymbol{w}')}\right).
\end{aligned} \tag{37}$$

Eq.37 can be split into two terms

$$
\begin{cases}
A(\boldsymbol{w}) = \left( \dfrac{q_{i_j}(\boldsymbol{w})}{Z_i(\boldsymbol{w})} - \dfrac{q_{i_j}(\boldsymbol{w}')}{Z_i(\boldsymbol{w}')} \right) \dfrac{\sum_{j'} q_{i_{j'}}^{o-1}(\boldsymbol{w})}{\sum_{j'} q_{i_{j'}}^{o}(\boldsymbol{w})} \\[4mm]
B(\boldsymbol{w}) = \dfrac{q_{i_j}(\boldsymbol{w}')}{Z_i(\boldsymbol{w}')} \left( \dfrac{\sum_{j'} q_{i_{j'}}^{o-1}(\boldsymbol{w})}{\sum_{j'} q_{i_{j'}}^{o}(\boldsymbol{w})} - \dfrac{\sum_{j'} q_{i_{j'}}^{o-1}(\boldsymbol{w}')}{\sum_{j'} q_{i_{j'}}^{o}(\boldsymbol{w}')} \right)
\end{cases}
\tag{38}
$$

By using the bounds in Eq. 24 and 29, we have

$$
\begin{aligned}
\frac{q_{i_j}(\boldsymbol{w})}{Z_i(\boldsymbol{w})} - \frac{q_{i_j}(\boldsymbol{w}')}{Z_i(\boldsymbol{w}')} &= \left( \frac{q_{i_j}(\boldsymbol{w})}{Z_i(\boldsymbol{w})} - \frac{q_{i_j}(\boldsymbol{w})}{Z_i(\boldsymbol{w}')} \right) + \left( \frac{q_{i_j}(\boldsymbol{w})}{Z_i(\boldsymbol{w}')} - \frac{q_{i_j}(\boldsymbol{w}')}{Z_i(\boldsymbol{w}')} \right) \\
&= q_{i_j}(\boldsymbol{w}) \left( \frac{1}{Z_i(\boldsymbol{w})} - \frac{1}{Z_i(\boldsymbol{w}')} \right) + \frac{1}{Z_i(\boldsymbol{w}')} \left( q_{i_j}(\boldsymbol{w}) - q_{i_j}(\boldsymbol{w}') \right) \\
&= q_{i_j}(\boldsymbol{w}) \frac{Z_i(\boldsymbol{w}') - Z_i(\boldsymbol{w})}{Z_i(\boldsymbol{w}) Z_i(\boldsymbol{w}')} + \frac{1}{Z_i(\boldsymbol{w}')} \left( q_{i_j}(\boldsymbol{w}) - q_{i_j}(\boldsymbol{w}') \right) . \\
&\leq (\psi_1 \gamma + \psi_2) K_{\max} \| \boldsymbol{w} - \boldsymbol{w}' \|.
\end{aligned}
\tag{39}
$$

For any $o \in \mathbb{N}^+$, $\dfrac{\sum_{j'} q_{i_{j'}}^{o-1}(\boldsymbol{w})}{\sum_{j'} q_{i_{j'}}^{o}(\boldsymbol{w})} \leq 1$. Thus, we bound the term $A(\boldsymbol{w})$ as

$$
|A(\boldsymbol{w})| \leq (\psi_1 \gamma + \psi_2) K_{\max} \| \boldsymbol{w} - \boldsymbol{w}' \|.
\tag{40}
$$

For the term $B(\boldsymbol{w})$, we formulate it as

$$
\frac{q_{i_j}(\boldsymbol{w}')}{Z_i(\boldsymbol{w}')} \left( \frac{C(\boldsymbol{w})}{D(\boldsymbol{w})} - \frac{C(\boldsymbol{w}')}{D(\boldsymbol{w}')} \right) = \frac{q_{i_j}(\boldsymbol{w}')}{Z_i(\boldsymbol{w}')} \cdot \frac{C(\boldsymbol{w}) D(\boldsymbol{w}') - C(\boldsymbol{w}') D(\boldsymbol{w})}{D(\boldsymbol{w}) D(\boldsymbol{w}')},
\tag{41}
$$

where $C(\boldsymbol{w}) = \sum_{j'} q_{i_{j'}}^{o-1}(\boldsymbol{w})$, $D(\boldsymbol{w}) = \sum_{j'} q_{i_{j'}}^{o}(\boldsymbol{w})$. It is obvious to observe that the denominator is bounded by

$$
D(\boldsymbol{w}) D(\boldsymbol{w}') \geq N^2 \delta^{2o}.
\tag{42}
$$

And the numerator can be expressed as follows

$$
\begin{aligned}
&C(\boldsymbol{w}) D(\boldsymbol{w}') - C(\boldsymbol{w}') D(\boldsymbol{w}) \\
&= \sum_{j'} \sum_{k'} \left( q_{i_{j'}}^{o-1}(\boldsymbol{w}) q_{i_{k'}}^{o}(\boldsymbol{w}') - q_{i_{j'}}^{o-1}(\boldsymbol{w}') q_{i_{k'}}^{o}(\boldsymbol{w}) \right) \\
&= \sum_{j'} \sum_{k'} \left( \left( q_{i_{j'}}^{o-1}(\boldsymbol{w}) - q_{i_{j'}}^{o-1}(\boldsymbol{w}') \right) q_{i_{k'}}^{o}(\boldsymbol{w}') + q_{i_{j'}}^{o-1}(\boldsymbol{w}') \left( q_{i_{k'}}^{o}(\boldsymbol{w}') - q_{i_{k'}}^{o}(\boldsymbol{w}) \right) \right),
\end{aligned}
\tag{43}
$$

which would be bounded using

$$
\begin{cases}
\left| q_{i_{j'}}^{o-1}(\boldsymbol{w}) - q_{i_{j'}}^{o-1}(\boldsymbol{w}') \right| \leq (o-1) \gamma^{o-2} \cdot \alpha \psi_2 K_{\max} \| \boldsymbol{w} - \boldsymbol{w}' \|, \\
\left| q_{i_{k'}}^{o}(\boldsymbol{w}') - q_{i_{k'}}^{o}(\boldsymbol{w}) \right| \leq o \gamma^{o-1} \cdot \alpha \psi_2 K_{\max} \| \boldsymbol{w} - \boldsymbol{w}' \|
\end{cases}
\tag{44}
$$

Thus, the numerator can be bounded as follows

$$
|C(\boldsymbol{w}) D(\boldsymbol{w}') - C(\boldsymbol{w}') D(\boldsymbol{w})| \leq N^2 (2o-1) \gamma^{2o-2} \alpha \psi_2 K_{\max} \| \boldsymbol{w} - \boldsymbol{w}' \|,
\tag{45}
$$

which allows us to bound the entire term $B(\boldsymbol{w})$ by

$$
|B(\boldsymbol{w})| \leq \frac{(2o-1) \gamma^{2o-1} \psi_2 K_{\max}}{\delta^{2o}} \| \boldsymbol{w} - \boldsymbol{w}' \|.
\tag{46}
$$

To sum up ④, we have

$$
\left| \frac{q_{i_j}(\boldsymbol{w})}{Z_i(\boldsymbol{w})} \frac{\sum_{j'} q_{i_{j'}}^{o-1}(\boldsymbol{w})}{\sum_{j'} q_{i_{j'}}^{o}(\boldsymbol{w})} - \frac{q_{i_j}(\boldsymbol{w}')}{Z_i(\boldsymbol{w}')} \frac{\sum_{j'} q_{i_{j'}}^{o-1}(\boldsymbol{w}')}{\sum_{j'} q_{i_{j'}}^{o}(\boldsymbol{w}')} \right| \leq \left( \psi_1 \gamma + \psi_2 + (2o-1) \psi_2 \psi_3^{2o-1} \delta^{-1} \right) K_{\max} \| \boldsymbol{w} - \boldsymbol{w}' \|.
\tag{47}
$$

⑤ **Lastly, we obtain an upper bound for the term** $k_{ij} - q_{i_j}(w) \sum_{j'} k_{ij'}$

Since $q_{i_j}(w) \leq \gamma$ for any $w$, we have

$$\left| k_{ij} - q_{i_j}(w) \sum_{j'} k_{ij'} \right| \leq |k_{ij}| + \left| \max(q_{i_j}(w)) \sum_{j'} k_{ij'} \right| \leq (1 + \gamma N) K_{\max}, \quad (48)$$

which is irrelevant to $w$ and could be treated as a constant.

**Combing the steps ① ② ③ ④ ⑤**, we obtain the upper bound of $\|\nabla_w \mathcal{L}^{(o)}(w) - \nabla_w \mathcal{L}^{(o)}(w')\|$ as

$$\left| \sum_{i,j} (T_{i,j}(w) - T_{i,j}(w')) \right| \leq C \cdot N^2 (1 + \gamma N) \cdot K_{\max}^2 \|w - w'\|, \quad (49)$$

where we denote the constant $C = \left( 1 + (o-1) \log \delta^{-1} + \log(N\delta^{-o}) + \gamma \right) \cdot \psi_1 + (1 + (o-1)\delta^{-1} + (o + (2o-1)\psi_3^o)\psi_3^{o-1}\delta^{-1}) \cdot \psi_2$. Hence, the Lipschitz constant $L$ for the gradient of $w$ is

$$L = C \cdot N^2 (1 + \gamma N) \cdot K_{\max}^2. \quad (50)$$

$\square$

**Proof of Theorem 4**

*Proof.* Recall that the probability $q_{i_j}$ is defined as $q_{i_j} = \frac{\tilde{k}_{ij}(w)}{Z_i(w)} = \frac{w^\top k_{ij}}{\sum_{j'} w^\top k_{ij'}}$, where $Z_i(w) = \sum_{j'} w^\top k_{ij'}$. Given a small perturbation $\Delta k_{ij} \in \mathbb{R}^M$ with its magnitude $\|\Delta k_{ij}\| \leq \eta$, the new kernel becomes $k'_{ij} = k_{ij} + \Delta k_{ij}$, and the perturbed probability $q'_{i_j}$ becomes

$$\begin{aligned} q'_{i_j} &= \frac{w^\top (k_{ij} + \Delta k_{ij})}{\sum_{j'} w^\top (k_{ij'} + \Delta k_{ij'})} \\ &= \frac{w^\top k_{ij} + w^\top \Delta k_{ij}}{\sum_{j'} (w^\top k_{ij'} + w^\top \Delta k_{ij'})} \end{aligned} \quad (51)$$

Then, the change in the probability $q_{i_j}$ due to the perturbation is given by $\Delta q_{i_j} = q'_{i_j} - q_{i_j}$:

$$\begin{aligned} \Delta q_{i_j} &= \frac{w^\top k_{ij} + w^\top \Delta k_{ij}}{\sum_{j'} (w^\top k_{ij'} + w^\top \Delta k_{ij'})} - \frac{w^\top k_{ij}}{\sum_{j'} w^\top k_{ij'}} \\ &= \frac{w^\top \Delta k_{ij}}{\sum_{j'} (w^\top k_{ij'} + w^\top \Delta k_{ij'})} + \frac{w^\top k_{ij}}{\sum_{j'} (w^\top k_{ij'} + w^\top \Delta k_{ij'})} - \frac{w^\top k_{ij}}{\sum_{j'} w^\top k_{ij'}} \\ &= \frac{w^\top \Delta k_{ij}}{\sum_{j'} (w^\top k_{ij'} + w^\top \Delta k_{ij'})} - w^\top k_{ij} \frac{\sum_{j'} w^\top \Delta k_{ij'}}{\sum_{j'} (w^\top k_{ij'} + w^\top \Delta k_{ij'}) \sum_{j'} (w^\top k_{ij'})} \\ &= \frac{w^\top \Delta k_{ij}}{Z_i(w) + \sum_{j'} w^\top \Delta k_{ij'}} - q_{i_j} \cdot \frac{\sum_{j'} w^\top \Delta k_{ij'}}{Z_i(w) + \sum_{j'} w^\top \Delta k_{ij'}} \end{aligned} \quad (52)$$

Since $\sum_{j'} w^\top \Delta k_{ij'} \geq 0$, the magnitude of the change $\Delta q_{i_j}$ satisfies that

$$|\Delta q_{i_j}| \leq \left| \frac{w^\top \Delta k_{ij}}{Z_i(w)} \right| + \left| q_{i_j} \cdot \frac{\sum_{j'} w^\top \Delta k_{ij'}}{Z_i(w)} \right|. \quad (53)$$

Given that $Z_i(w) \geq \alpha$, $\|w\| \leq \sigma$, and $0 < \delta \leq q_{i_j} \leq \gamma < 1$, we can bound $|\Delta q_{i_j}|$ as

$$\begin{aligned} |\Delta q_{i_j}| &\leq \alpha\sigma(\|\Delta k_{ij}\| + \gamma \sum_{j'} \|\Delta k_{ij'}\|) \\ &\leq \alpha\sigma(1 + \gamma N) \cdot \eta \end{aligned} \quad (54)$$

Based on the gradient $\nabla_{\boldsymbol{w}}\mathcal{L}$ in Equation 22, the optimal condition $\mathcal{O}(\boldsymbol{w})$ is expressed as

$$\mathcal{O}(\boldsymbol{w}) = -(o-1)(\log q_{i_j}(\boldsymbol{w}) + 1) + \log \sum_{j'} q^o_{i_{j'}}(\boldsymbol{w}) + oq_{i_j}(\boldsymbol{w})\frac{\sum_{j'} q^{o-1}_{i_{j'}}(\boldsymbol{w})}{\sum_{j'} q^o_{i_{j'}}(\boldsymbol{w})} = 0 \qquad (55)$$

To analyze how this condition is affected by the perturbation, we bound the change in $\mathcal{O}(\boldsymbol{w})$ due to the change in $q_{i_j}(\boldsymbol{w})$ as

$$
\begin{aligned}
|\Delta\mathcal{O}| \leq & (o-1)\left|\Delta\log q_{i_j}(\boldsymbol{w})\right| + \left|\Delta\log\sum_{j'} q^o_{i_{j'}}(\boldsymbol{w})\right| + \left|\Delta\left(oq_{i_j}(\boldsymbol{w})\frac{\sum_{j'} q^{o-1}_{i_{j'}}(\boldsymbol{w})}{\sum_{j'} q^o_{i_{j'}}(\boldsymbol{w})}\right)\right| \\
\leq & (o-1)\left|\frac{\Delta q_{i_j}}{q_{i_j}}\right| + \left|\frac{\sum_{j'} o\cdot q^{o-1}_{i_{j'}}\cdot\Delta q_{i_{j'}}}{\sum_{j'} q_{i_{j'}}}\right| + \left|o\cdot\Delta q_{i_j}\right| \\
\leq & (o-1)\delta|\Delta q_{i_j}| + o\gamma^{o-1}\delta^o|\Delta q_{i_j}| + o|\Delta q_{i_j}| \\
\leq & \left((o-1)\delta + o\gamma^{o-1}\delta^o + o\right)\alpha\sigma(1+\gamma N)\cdot\eta \\
= & C\cdot\eta,
\end{aligned}
\qquad (56)
$$

where $C = \left((o-1)\delta + o\gamma^{o-1}\delta^o + o\right)\alpha\sigma(1+\gamma N)$. □

**Proof of Theorem 5**

*Proof.* With the algorithm $A_{\mathcal{G}\setminus r}$, each $q^{\setminus r}_{i_j} = \frac{\tilde{k}_{ij}}{\sum_{j'\neq r}\tilde{k}_{ij'}}$ is deviated from $q_{i_j} = \frac{\tilde{k}_{ij}}{\sum_{j'}\tilde{k}_{ij'}}$ in the original algorithm $A_{\mathcal{G}}$. Based on Eq.19, we have the following losses on graph $G_i$ when processed by the algorithm $A_{\mathcal{G}}$ and $A_{\mathcal{G}\setminus r}$, respectively:

$$\mathcal{L}^{(o)}(G_i, A_{\mathcal{G}}) = \sum_j\left(-q_{i_j}\log q^{o-1}_{i_j} + q_{i_j}\log\sum_{j'} q^o_{i_{j'}}\right) \qquad (57)$$

$$\mathcal{L}^{(o)}(G_i, A_{\mathcal{G}\setminus r}) = \sum_{j\neq r}\left(-q^{\setminus r}_{i_j}\log(q^{\setminus r}_{i_j})^{o-1} + q^{\setminus r}_{i_j}\log\sum_{j'\neq r}(q^{\setminus r}_{i_{j'}})^o\right) \qquad (58)$$

Taking the difference between the equations above, we obtain its magnitude as

$$
\begin{aligned}
& \left|\mathcal{L}^{(o)}(G_i, A_{\mathcal{G}}) - \mathcal{L}^{(o)}(G_i, A_{\mathcal{G}\setminus r})\right| \\
= & \left| -q_{i_r}\log q^{o-1}_{i_r} + q_{i_r}\log\sum_{j'} q^o_{i_{j'}} + \sum_{j\neq r}\left(-q_{i_j}\log q^{o-1}_{i_j} + q_{i_j}\log\sum_{j'} q^o_{i_{j'}} + q^{\setminus r}_{i_j}\log(q^{\setminus r}_{i_j})^{o-1} - q^{\setminus r}_{i_j}\log\sum_{j'\neq r}(q^{\setminus r}_{i_{j'}})^o\right)\right| \\
= & \left| \underbrace{-q_{i_r}\log q^{o-1}_{i_r} + q_{i_r}\log\sum_{j'} q^o_{i_{j'}}}_{T_1(i)} + \underbrace{\sum_{j\neq r}\left(q^{\setminus r}_{i_j}\log(q^{\setminus r}_{i_j})^{o-1} - q^{\setminus r}_{i_j}\log q^{o-1}_{i_j}\right)}_{T_2(i)} + \underbrace{\sum_{j\neq r}\left(q^{\setminus r}_{i_j}\log q^{o-1}_{i_j} - q_{i_j}\log q^{o-1}_{i_j}\right)}_{T_3(i)} \right. \\
& \left. + \underbrace{\sum_{j\neq r}\left(q_{i_j}\log\sum_{j'} q^o_{i_{j'}} - q^{\setminus r}_{i_j}\log\sum_{j'} q^o_{i_{j'}}\right)}_{T_4(i)} + \underbrace{\sum_{j\neq r}\left(q^{\setminus r}_{i_j}\log\sum_{j'} q^o_{i_{j'}} - q^{\setminus r}_{i_j}\log\sum_{j'\neq r}(q^{\setminus r}_{i_{j'}})^o\right)}_{T_5(i)} \right| \\
\triangleq & |T_1(i) + T_2(i) + T_3(i) + T_4(i) + T_5(i)|
\end{aligned}
\qquad (59)
$$

For simplification, we define $T_1(i), T_2(i), T_3(i), T_4(i)$, and break down each term as follows.

$$T_1(i) \triangleq -q_{i_r} \log q_{i_r}^{o-1} + q_{i_r} \log \sum_{j'} q_{i_{j'}}^o = -(o-1)q_{i_r} \log q_{i_r} + q_{i_r} \log \sum_{j'} q_{i_{j'}}^o$$

$$= -(o-1)\frac{\tilde{k}_{ir}}{\sum_{j'} \tilde{k}_{ij'}} \log\left(\frac{\tilde{k}_{ir}}{\sum_{j'} \tilde{k}_{ij'}}\right) + \frac{\tilde{k}_{ir}}{\sum_{j'} \tilde{k}_{ij'}} \log \sum_{j'} \left(\frac{\tilde{k}_{ij'}}{\sum_{j''} \tilde{k}_{ij''}}\right)^o$$

$$= -(o-1)\frac{\tilde{k}_{ir}}{\sum_{j'} \tilde{k}_{ij'}} \log\left(\frac{\tilde{k}_{ir}}{\sum_{j'} \tilde{k}_{ij'}}\right) + \frac{\tilde{k}_{ir}}{\sum_{j'} \tilde{k}_{ij'}} \log \left(\frac{\sum_{j'} \tilde{k}_{ij'}^o}{(\sum_{j'} \tilde{k}_{ij'})^o}\right)$$

$$(60)$$

$$T_2(i) \triangleq \sum_{j \neq r} \left(q_{i_j}^{\backslash r} \log(q_{i_j}^{\backslash r})^{o-1} - q_{i_j}^{\backslash r} \log q_{i_j}^{o-1}\right) = (o-1)\sum_{j \neq r} q_{i_j}^{\backslash r}\left(\log(q_{i_j}^{\backslash r}) - \log q_{i_j}\right)$$

$$= (o-1)\sum_{j \neq r} q_{i_j}^{\backslash r} \cdot \log\left(\frac{q_{i_j}^{\backslash r}}{q_{i_j}}\right)$$

$$= (o-1)\frac{\sum_{j \neq r} \tilde{k}_{ij}}{\sum_{j' \neq r} \tilde{k}_{ij'}} \cdot \log\left(1 + \frac{\tilde{k}_{ir}}{\sum_{j' \neq r} \tilde{k}_{ij'}}\right)$$

$$= (o-1)\log\left(1 + \frac{\tilde{k}_{ir}}{\sum_{j' \neq r} \tilde{k}_{ij'}}\right)$$

$$(61)$$

$$T_3(i) \triangleq \sum_{j \neq r} \left(q_{i_j}^{\backslash r} \log q_{i_j}^{o-1} - q_{i_j} \log q_{i_j}^{o-1}\right) = \sum_{j \neq r} \left(q_{i_j}^{\backslash r} - q_{i_j}\right) \log q_{i_j}^{o-1}$$

$$= (o-1)\sum_{j \neq r} \tilde{k}_{ij}\left(\frac{1}{\sum_{j' \neq r} \tilde{k}_{ij'}} - \frac{1}{\sum_j \tilde{k}_{ij'}}\right) \log\left(\frac{\tilde{k}_{ij}}{\sum_{j' \neq r} \tilde{k}_{ij'}}\right)$$

$$= (o-1)\frac{\tilde{k}_{ir}}{\left(\sum_{j' \neq r} \tilde{k}_{ij'}\right)\left(\sum_j \tilde{k}_{ij'}\right)} \log\left(\sum_{j' \neq r} \tilde{k}_{ij'}\right) \sum_{j \neq r}\left(-\tilde{k}_{ij} \log \tilde{k}_{ij}\right)$$

$$(62)$$

$$T_4(i) \triangleq \sum_{j \neq r} \left(q_{i_j} \log \sum_{j'} q_{i_{j'}}^o - q_{i_j}^{\backslash r} \log \sum_{j'} q_{i_{j'}}^o\right) = \sum_{j \neq r} \left(q_{i_j} - q_{i_j}^{\backslash r}\right) \log \sum_{j'} q_{i_{j'}}^o$$

$$= \sum_{j \neq r} \tilde{k}_{ij}\left(\frac{1}{\sum_{j' \neq r} \tilde{k}_{ij'}} - \frac{1}{\sum_j \tilde{k}_{ij'}}\right) \log\left(\frac{(\sum_{j'} \tilde{k}_{ij'})^o}{\sum_{j'} \tilde{k}_{ij'}^o}\right)$$

$$= \frac{\left(\sum_{j \neq r} \tilde{k}_{ij}\right)\tilde{k}_{ir}}{\left(\sum_{j' \neq r} \tilde{k}_{ij'}\right)\left(\sum_j \tilde{k}_{ij'}\right)} \log\left(\frac{(\sum_{j'} \tilde{k}_{ij'})^o}{\sum_{j'} \tilde{k}_{ij'}^o}\right)$$

$$= \frac{\tilde{k}_{ir}}{\sum_j \tilde{k}_{ij'}} \log\left(\frac{(\sum_{j'} \tilde{k}_{ij'})^o}{\sum_{j'} \tilde{k}_{ij'}^o}\right)$$

$$(63)$$

$$T_5(i) \triangleq \sum_{j \neq r} \left( q_{i_j}^{\backslash r} \log \sum_{j'} q_{i_{j'}}^o - q_{i_j}^{\backslash r} \log \sum_{j' \neq r} (q_{i_{j'}}^{\backslash r})^o \right) = \sum_{j \neq r} q_{i_j}^{\backslash r} \left( \log \sum_{j'} q_{i_{j'}}^o - \log \sum_{j' \neq r} (q_{i_{j'}}^{\backslash r})^o \right)$$

$$= \sum_{j \neq r} q_{i_j}^{\backslash r} \log \frac{\sum_{j'} q_{i_{j'}}^o}{\sum_{j' \neq r} (q_{i_{j'}}^{\backslash r})^o}$$

$$= \log \frac{\sum_{j'} \tilde{k}_{ij'}^o / (\sum_{j''} \tilde{k}_{ij''})^o}{\sum_{j' \neq r} \tilde{k}_{ij'}^o / (\sum_{j'' \neq r} \tilde{k}_{ij''})^o}$$

$$= \log \frac{\sum_{j'} \tilde{k}_{ij'}^o}{\sum_{j' \neq r} \tilde{k}_{ij'}^o} - \log \frac{(\sum_{j'} \tilde{k}_{ij'})^o}{(\sum_{j' \neq r} \tilde{k}_{ij'})^o}$$

$$= \log \left( 1 + \frac{\tilde{k}_{ir}^o}{\sum_{j' \neq r} \tilde{k}_{ij'}^o} \right) - o \cdot \log \left( 1 + \frac{\tilde{k}_{ir}}{\sum_{j' \neq r} \tilde{k}_{ij'}} \right)$$

$$\text{(64)}$$

Therefore, the maximal magnitude of the difference among all graphs $\{G_1, \cdots, G_N\}$ is

$$\max_{i=1,\cdots,N} \left| \mathcal{L}^{(o)}(G_i, A_{\mathcal{G}}) - \mathcal{L}^{(o)}(G_i, A_{\mathcal{G} \backslash r}) \right|$$
$$= \max_i T_1(i) + \max_i T_2(i) + \max_i T_3(i) + \max_i T_4(i) - \min_i T_5(i) \tag{65}$$

Denote $\rho_u$ as the upper bound and $\rho_l$ as lower bound of $\tilde{k}_{ij}$ for all $i, j \in \{i, \cdots, N\}$, i.e., $0 \leq \rho_l \leq \tilde{k}_{ij} \leq \rho_u \leq 1$. Therefore, each maximum or minimum term above has the following bound. (Note: We use $-x \log x \leq e^{-1}$ for $T_1(i)$ and $T_3(i)$).

$$\begin{cases} \max_i T_1(i) \leq (o-1)e^{-1} + N^{-1}\rho_u\rho_l^{-1} \left( o \log \rho_u - (o-1) \log N - o \log \rho_l^{-1} \right) \\[2mm] \max_i T_2(i) \leq (o-1) \log(1 + (N-1)^{-1}\rho_u\rho_l^{-1}) \\[2mm] \max_i T_3(i) \leq (o-1)(N-1)^{-1}N^{-1}\rho_u\rho_l^{-2} \log \left( (N-1)\rho_u \right) (N-1)e^{-1} \\[2mm] \max_i T_4(i) \leq N^{-1}\rho_u\rho_l \left( o \log \rho_u + (o-1) \log N - o \log \rho_l^{-1} \right) \\[2mm] \min_i T_5(i) \geq 2o \log \frac{N}{N-1} + 2o \log \rho_l^{-1} - 2o \log \rho_u \end{cases} \tag{66}$$

Hence, we can define the $\omega$-stability as below:

$$\begin{aligned} \omega := {} & (o-1)e^{-1} + N^{-1}\rho_u\rho_l^{-1} \left( o \log \rho_u - (o-1) \log N - o \log \rho_l^{-1} \right) \\ & + (o-1) \log(1 + (N-1)^{-1}\rho_u\rho_l^{-1}) \\ & + (o-1)(N-1)^{-1}N^{-1}\rho_u\rho_l^{-2} \log \left( (N-1)\rho_u \right) (N-1)e^{-1} \\ & + N^{-1}\rho_u\rho_l \left( o \log \rho_u + (o-1) \log N - o \log \rho_l^{-1} \right) \\ & + 2o \log \frac{N}{N-1} + 2o \log \rho_l^{-1} - 2o \log \rho_u. \end{aligned} \tag{67}$$

Also, the algorithm UMKL-G is stable as $\omega_N \propto \frac{1}{N}$. $\qquad\qquad\qquad\qquad\qquad\qquad\square$

## C  ALGORITHM 2

---

**Algorithm 2:** Projecting Weights onto the Unit Simplex

---

**Input**  : Vector $\boldsymbol{w}$
**Output:** Projected vector $\boldsymbol{w}$
Sort $\boldsymbol{w}$ in descending order: $\boldsymbol{w}_{\text{sorted}}$;
Initialize cumulative sum: $S = 0$;
**for** $i = 1$ **to** $M$ **do**
    Update cumulative sum: $S = S + \boldsymbol{w}_{\text{sorted}}[i]$;
    Compute $\lambda[i] = \frac{S-1}{i}$             // Adjustment of each element to a unit simplex
**for** $i = 1$ **to** $M - 1$ **do**
    **if** $\lambda[i] \geq \boldsymbol{w}_{sorted}[i+1]$ **then**
        $\alpha = \lambda[i]$     // Assign the threshold to the first $\lambda$ that satisfies the condition
        **break**
**if** $i = M$ **then**
    $\alpha = \lambda[M-1]$               // If no break occurs, use the last $\lambda$
$\boldsymbol{w} = \max(\boldsymbol{w} - \alpha, 0)$;
**return** $\boldsymbol{w}$

---

## D  THEOREM AND PROOF FOR ALGORITHM 2

**Theorem 7.** *The Algorithm 2 projects any vector $\boldsymbol{x} \in \mathbb{R}^M$ onto the unit simplex $\Delta^{M-1}$, ensuring that the projected vector $\boldsymbol{w}$ satisfies: ① $\sum_{i=1}^{M} w_i = 1$, ② $w_i \geq 0$ for all $i$, ③ smallest adjustment in terms of the Euclidean distance needed to project the vector onto the simplex.*

*Proof.* The unit simplex in $\mathbb{R}^M$ is defined as:

$$\Delta^{M-1} = \{\boldsymbol{w} \in \mathbb{R}^M \mid \sum_{i=1}^{M} w_i = 1 \text{ and } w_i \geq 0 \text{ for all } i\}.$$

Let $\boldsymbol{x} \in \mathbb{R}^M$ be the original vector, and let $\boldsymbol{w} \in \mathbb{R}^M$ be the projected vector. The Algorithm 2 involves the following steps:

① Sort $\boldsymbol{x}$ in descending order to get $\boldsymbol{x}_{\text{sorted}}$.

② Compute the cumulative sum and the normalized difference to 1

$$\lambda_a[i] = \frac{\sum_{j=1}^{i} x_{\text{sorted}}[j] - 1}{i}.$$

③ Find the largest $i$ such that $\lambda_a[i] \geq x_{\text{sorted}}[i+1]$ and set $\alpha = \lambda_a[i]$.

④ Project $\boldsymbol{x}$ onto the simplex by setting

$$w_i = \max(x_i - \alpha, 0).$$

We show that the resulting $\boldsymbol{w}$ satisfies the properties of the unit simplex as follows.

**Sum of Projected Elements.**

The sum of the projected elements is

$$\sum_{i=1}^{M} w_i = \sum_{i=1}^{k} (x_i - \alpha) + \sum_{i=k+1}^{M} 0 = \sum_{i=1}^{k} x_i - k\alpha.$$

Since $\alpha = \lambda_a[k] = \frac{\sum_{j=1}^{k} x_j - 1}{k}$, we have

$$k\alpha = \sum_{j=1}^{k} x_j - 1.$$

Therefore,

$$\sum_{i=1}^{M} w_i = \sum_{i=1}^{k} x_i - (\sum_{j=1}^{k} x_j - 1) = 1.$$

**Non-negativity.**

By definition of the projection step is

$$w_i = \max(x_i - \alpha, 0),$$

which ensures that all $w_i \geq 0$.

**Optimality in Euclidean Distance.**

The optimality of the projection in terms of Euclidean distance can be shown by solving the following optimization problem:

$$\min_{\boldsymbol{w}} \|\boldsymbol{w} - \boldsymbol{x}\|_2^2 \quad \text{subject to} \quad \sum_{i=1}^{M} w_i = 1 \text{ and } w_i \geq 0 \text{ for all } i.$$

This is a constrained quadratic optimization problem, which can be solved using the method of Lagrange multipliers and the optimizality condition are give by the KKT conditions: $w_i = \max(x_i - \lambda, 0)$. With $\lambda = \alpha$ in the Algorithm 2, it provides the optimal projection in terms of the Euclidean distance from the original vector to the simplex. This ensures the smallest adjustment needed to project the vector onto the simplex.

$\square$

# E    CONNECTION TO BASELINES (FULL)

Our proposed method shares a foundational goal with previous methods, such as UMKL (Zhuang et al., 2011) and sparse-UMKL (Mariette & Villa-Vialaneix, 2018), which is to preserve the local geometry of the data. These methods have laid significant groundwork in unsupervised multiple kernel learning. In UMKL, the authors propose two main principles: first, for each data point $\boldsymbol{x}_i$, the optimal kernel should minimize the approximation error $\|\boldsymbol{x}_i - \sum_j k_{ij}\boldsymbol{x}_j\|^2$; second, the method should minimize the distortion over all training data, $\sum_{ij} k_{ij}\|\boldsymbol{x}_i - \mathbf{x}_j\|^2$, where $k_{ij} = k(\boldsymbol{x}_i, \boldsymbol{x}_j)$. Similarly, sparse-UMKL aims to approximately preserve the local geometry of the data by building $k$-nearest neighbor graphs for each kernel and defining a weight matrix based on these graphs. Despite their innovative approaches, both UMKL and sparse-UMKL have limitations in handling graph data and achieving satisfactory empirical performances (see Section 5 for details).

| Feature | UMKL | sparse-UMKL | UMKL-G (Ours) |
|---|---|---|---|
| Objective Function | $\min_{\mu,D} \frac{1}{2}\|X(I - K \circ D)\|_F^2$ $+\gamma_1 \operatorname{tr}(K \circ D \circ M) + \gamma_2\|D\|_{1,1}$ | $\min_{\mathbf{b}} \operatorname{tr}(\mathbf{W}K) + \lambda\|\mathbf{b}\|_1,$ $K = \sum_{m=1}^M b_m K_m$ | $\min_{\mathbf{w}} L^{(o)} = \mathrm{KL}(Q\|P),$ $Q_{ij} = \frac{\bar{k}_{ij}}{\sum_{j'} \bar{k}_{ij'}}, \quad P_{ij} = \frac{\bar{k}_{ij}^o}{\sum_{j'} \bar{k}_{ij'}^o}$ |
| Beyond Euclidean Global Topology Theoretical Guarantees | ✗ ✗ ✓ | ✓ ✗ ✗ | ✓ ✓ ✓ |
| Topology Preservation Algorithm Complexity | Local reconstruction ($D$) Alternating minimization $O(I \cdot (MN^2 + N^3))$ | k-NN graph heuristics ($\mathbf{W}$) Quadratic programming solver $O(I \cdot (MN^2 \log N + M^3))$ | Ordinal relationships KL divergence $O(I \cdot (MN^2 + M \log M))$ |

Table 3: Comparison of UMKL, sparse-UMKL, and UMKL-G.

While our method aligns with these foundational principles, it also introduces a novel perspective by focusing on ordinal relationships to preserve the topology of the data. Unlike UMKL, which primarily focuses on minimizing approximation errors, our approach leverages the concentration effect of powered kernels to maintain the local and global topology more effectively. Moreover, compared to sparse-UMKL, which builds $k$-nearest neighbor graphs for each kernel, our method avoids explicitly constructing local graphs, thereby reducing the computational complexity and potential inaccuracies introduced by thresholding kernel values to obtain the nearest neighbors. Our method also differs in its handling of unsupervised learning for graph data. Whereas traditional UMKL and sparse-UMKL are designed primarily for vector data and rely on approximating local geometry — which can be sensitive to noise, struggle with high-dimensional data, require careful tuning of parameters like $k$, and often fail to capture global structural patterns — our method operates directly on probability simplices. By focusing on ordinal relationships, we avoid the limitations of local approximations and achieve a more robust and holistic preservation of both local and global topological features.

| Dataset | $N$ | UMKL-G (seconds) | UMKL (seconds) | sparse-UMKL (seconds) |
|---|---|---|---|---|
| MUTAG | 188 | 15.9384 | 30.9085 | 21.3190 |
| PTC_FM | 344 | 18.8914 | 39.5487 | 23.4447 |
| BZR | 405 | 23.5574 | 45.8796 | 29.4764 |
| COX2/DHFR | 467 | 28.9875 | 71.0475 | 33.3794 |
| ENZYMES | 600 | 30.2123 | 93.4008 | 39.9868 |
| IMDB-BINARY | 1000 | 43.4140 | 199.1917 | 48.4064 |
| DD | 1113 | 43.5285 | 819.8227 | 51.8620 |

Table 4: Runtime comparison of UMKL-G and baselines across benchmark datasets.

## F DESCRIPTION OF BENCHMARK DATASETS

### F.1 DATASETS

1. *BZR*: A dataset consisting of graphs representing chemical compounds classified based on their bioactivity against a certain protein.

2. *COX2*: Another chemical compound dataset, where the task is to predict the inhibition of the COX2 enzyme.

3. *DD*: A large dataset of protein structures, where the goal is to classify proteins into enzymes and non-enzymes.

4. *DHFR*: This dataset includes graphs representing compounds tested for their ability to inhibit dihydrofolate reductase.

5. *ENZYMES*: Contains graphs of protein tertiary structures categorized into six enzyme classes.

6. *IMDB-BINARY*: A social network dataset where the task is to classify movies into two genres based on their actor/actress co-appearance networks.

7. *MUTAG*: A dataset of chemical compounds labeled according to their mutagenic effect on a specific bacterium.

8. *PTC_FM*: Contains chemical compounds labeled based on their carcinogenicity in male rats.

Table 5: Description of the Benchmark Datasets.

| Dataset | BZR | COX2 | DD | DHFR | ENZYMES | IMDB-BINARY | MUTAG | PTC_FM |
|---|---|---|---|---|---|---|---|---|
| Num. of graphs | 405 | 467 | 1178 | 467 | 600 | 1000 | 188 | 349 |
| Num. of graph labels | 2 | 2 | 2 | 2 | 6 | 2 | 2 | 2 |
| Dim. of node attributes | 3 | 3 | / | 3 | 18 | / | / | / |
| Avg. number of nodes | 35.75 | 41.22 | 284.32 | 42.43 | 32.63 | 19.77 | 17.93 | 14.11 |
| Avg. number of edges | 38.36 | 43.45 | 715.66 | 44.54 | 62.14 | 96.53 | 19.79 | 14.48 |
| Label Proportion | 319/86 | 365/102 | 691/487 | 461/295 | 1/1/1/1/1/1 | 500/500 | 125/63 | 206/143 |

### F.2 BASE KERNELS

Table 6: Description of the Base Graph Kernels.

| Graph Kernel | Abbr. | Kernel Function | Hyperparameter | Values |
|---|---|---|---|---|
| Random Walk (Gärtner et al., 2003) | RW | $k_{RW}(G_i, G_j) = \sum_{p,q=1}^{|V_\times|} [\sum_{l=0}^{\infty} \lambda^l A_\times^l]_{pq}$ | $\lambda$ | $0.1, 0.5, 0.8$ |
| Shortest Path (Borgwardt & Kriegel, 2005) | SP | $k_{SP}(S_i, S_j) = \sum_{e_i \in E_i} \sum_{e_j \in E_j} k_{\text{walk}^{(1)}}(e_i, e_j)$ | – | – |
| Graphlet Sampling (Pržulj, 2007) | GS | $k_{WLOA}(G_i, G_j) = f_{G_i}^\top f_{G_j}; f_{G,i} = \#(graphlet_i \sqsubseteq G)$ | $k$ | $4, \cdots, 8$ |
| Vertex Histogram (Sugiyama & Borgwardt, 2015) | VH | $k_{VH}(G_i, G_j) = \langle \boldsymbol{f}_i, \boldsymbol{f}_j \rangle$ | – | – |
| Weisfeiler-Lehman Subtree (Kriege et al., 2016) | WL | $k_{WL}(G_i, G_j) = \langle \phi(G_i), \phi(G_j) \rangle; \phi(G) = (c_0(G, \sigma_{01}), \cdots, c_h(G, \sigma_{h|\Sigma_h|}))$ | $h$ | $1, \cdots, 10$ |
| Weisfeiler-Lehman Optimal Assignment (Kriege et al., 2016) | WLOA | $k_{WLOA}(G_i, G_j) = k_{\mathfrak{B}}^k(V_i, V_j); k(v, v') = \sum_{i=0}^{h} \delta(\tau_i(v), \tau_i(v'))$ | $h$ | $1, \cdots, 10$ |

### F.3 EVALUATION METRICS

**Clustering Accuracy (ACC)** measures the maximum one-to-one correspondence between true labels and predicted cluster labels. The Hungarian algorithm is used to determine the optimal mapping.

$$\text{ACC} = \frac{1}{n} \sum_{i=1}^{n} \mathbb{I}\{y_{\text{true},i} = \text{map}(y_{\text{pred},i})\}, \tag{68}$$

**Normalized Mutual Information (NMI)** quantifies the similarity between the ground truth and predicted cluster assignments based on mutual information and entropy.

$$\text{NMI}(Y_{\text{true}}, Y_{\text{pred}}) = \frac{2 \cdot I(Y_{\text{true}}; Y_{\text{pred}})}{H(Y_{\text{true}}) + H(Y_{\text{pred}})}, \tag{69}$$

where $I(\cdot)$ denotes the mutual information and $H(\cdot)$ represents the entropy.

**Adjusted Rand Index (ARI)** measures the similarity between the clustering results and ground truth labels, adjusting for chance. It is computed as:

$$\text{ARI} = \frac{\sum_{ij} \binom{n_{ij}}{2} - \left[ \sum_i \binom{a_i}{2} \sum_j \binom{b_j}{2} \right] / \binom{n}{2}}{\frac{1}{2} \left[ \sum_i \binom{a_i}{2} + \sum_j \binom{b_j}{2} \right] - \left[ \sum_i \binom{a_i}{2} \sum_j \binom{b_j}{2} \right] / \binom{n}{2}}, \tag{70}$$

where $n_{ij}$ is number of samples in both ground truth cluster $i$ and predicted cluster $j$, $a_i$ is number of samples in ground truth cluster $i$, $b_j$ is number of samples in predicted cluster $j$, and $n$ is total number of samples.

# G EMPIRICAL RESULTS

## G.1 UMKL-G V.S. BASE KERNELS

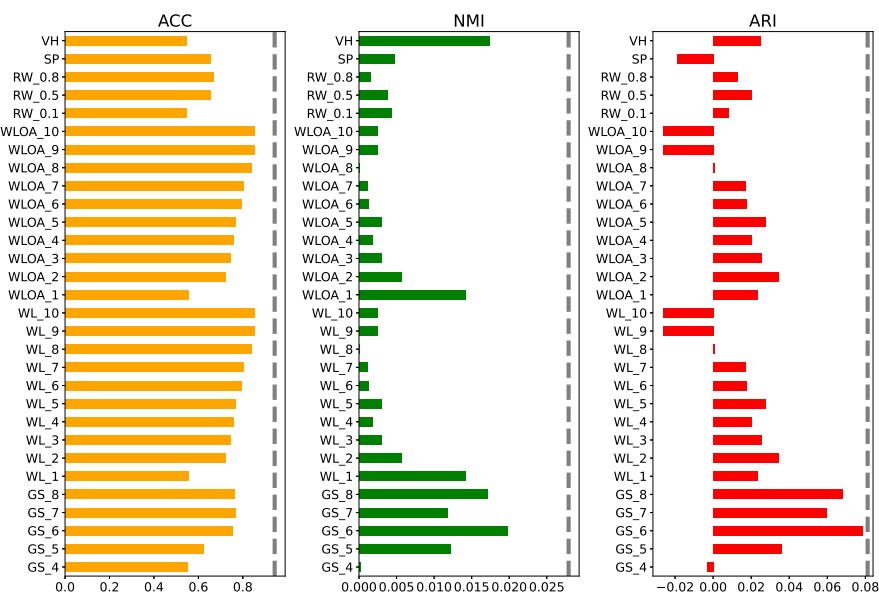

Figure 3: Comparison with Individual Base Kernels on the BZR dataset. *The bar plots represent the performance metrics for different kernels. The dashed grey lines indicate the performances of UMKL-G. Kernel names are shown with their respective hyperparameters.*

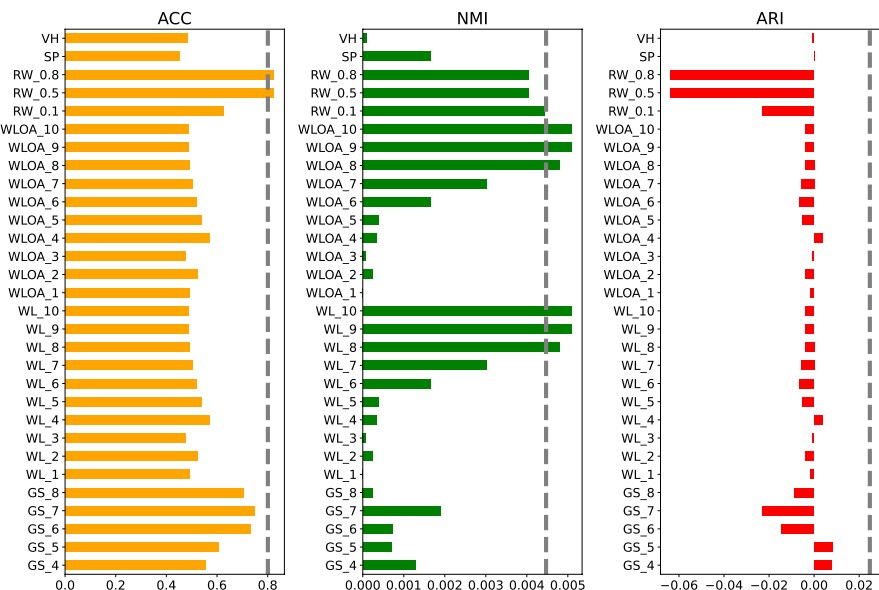

Figure 4: Comparison with Individual Base Kernels on the COX2 dataset. *The bar plots represent the performance metrics for different kernels. The dashed grey lines indicate the performances of UMKL-G. Kernel names are shown with their respective hyperparameters.*

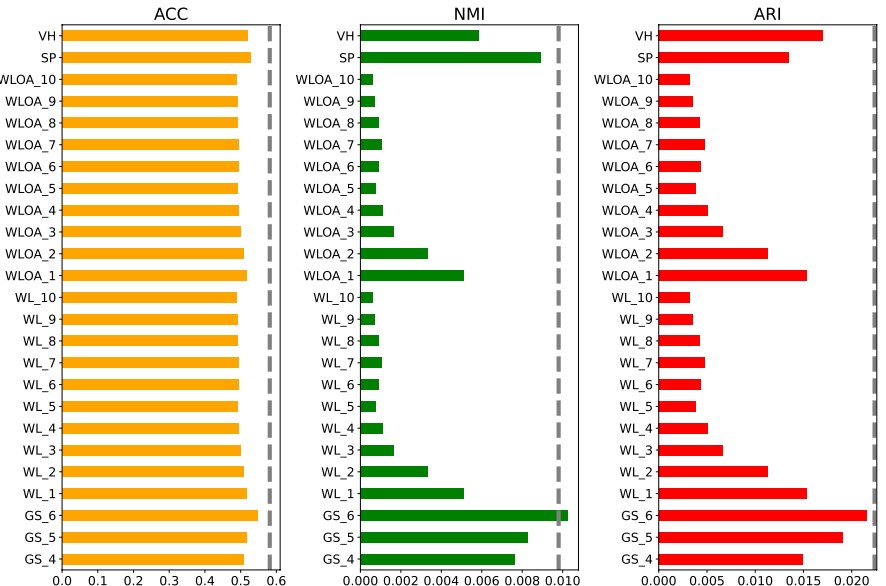

Figure 5: Comparison with Individual Base Kernels on the DD dataset. *The bar plots represent the performance metrics for different kernels. The dashed grey lines indicate the performances of UMKL-G. Kernel names are shown with their respective hyperparameters.*

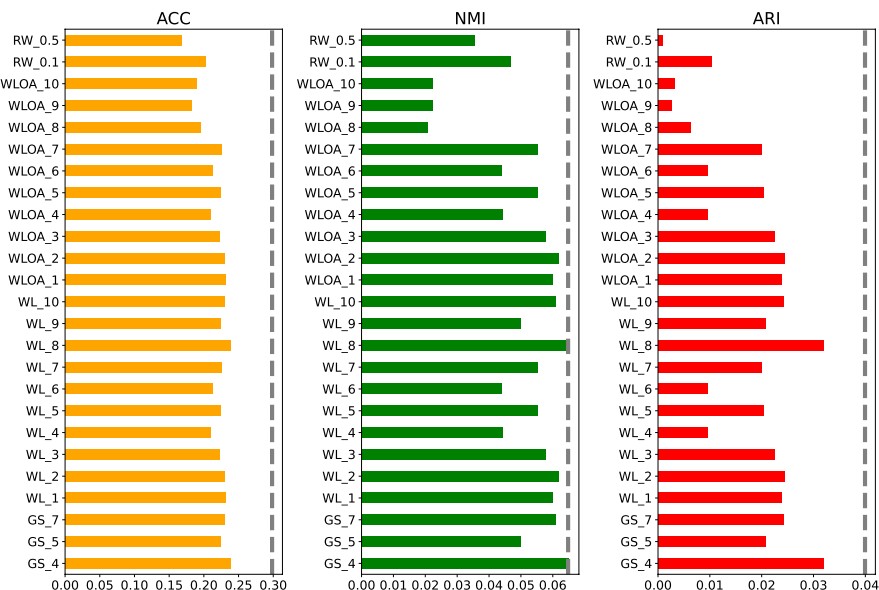

Figure 6: Comparison with Individual Base Kernels on the ENZYMES dataset. *The bar plots represent the performance metrics for different kernels. The dashed grey lines indicate the performances of UMKL-G. Kernel names are shown with their respective hyperparameters.*

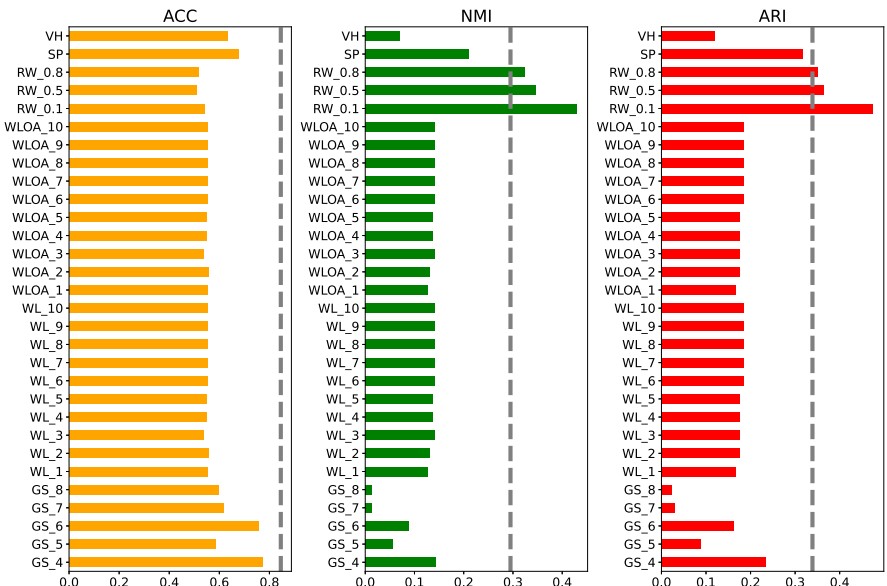

Figure 7: Comparison with Individual Base Kernels on the MUTAG dataset. *The bar plots represent the performance metrics for different kernels. The dashed grey lines indicate the performances of UMKL-G. Kernel names are shown with their respective hyperparameters.*

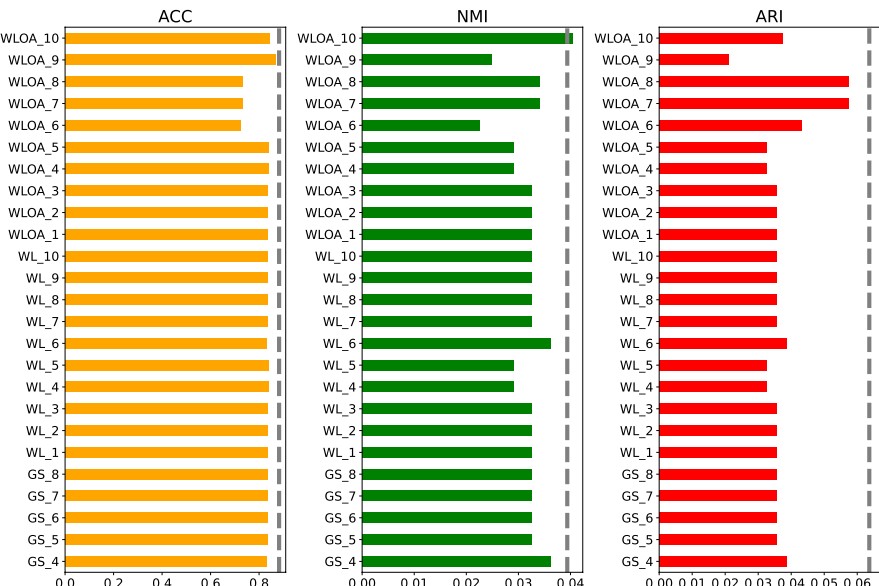

Figure 8: Comparison with Individual Base Kernels on the PTC_FM dataset. *The bar plots represent the performance metrics for different kernels. The dashed grey lines indicate the performances of UMKL-G. Kernel names are shown with their respective hyperparameters.*

## G.2 LEARNED WEIGHTS FROM UMKL-G

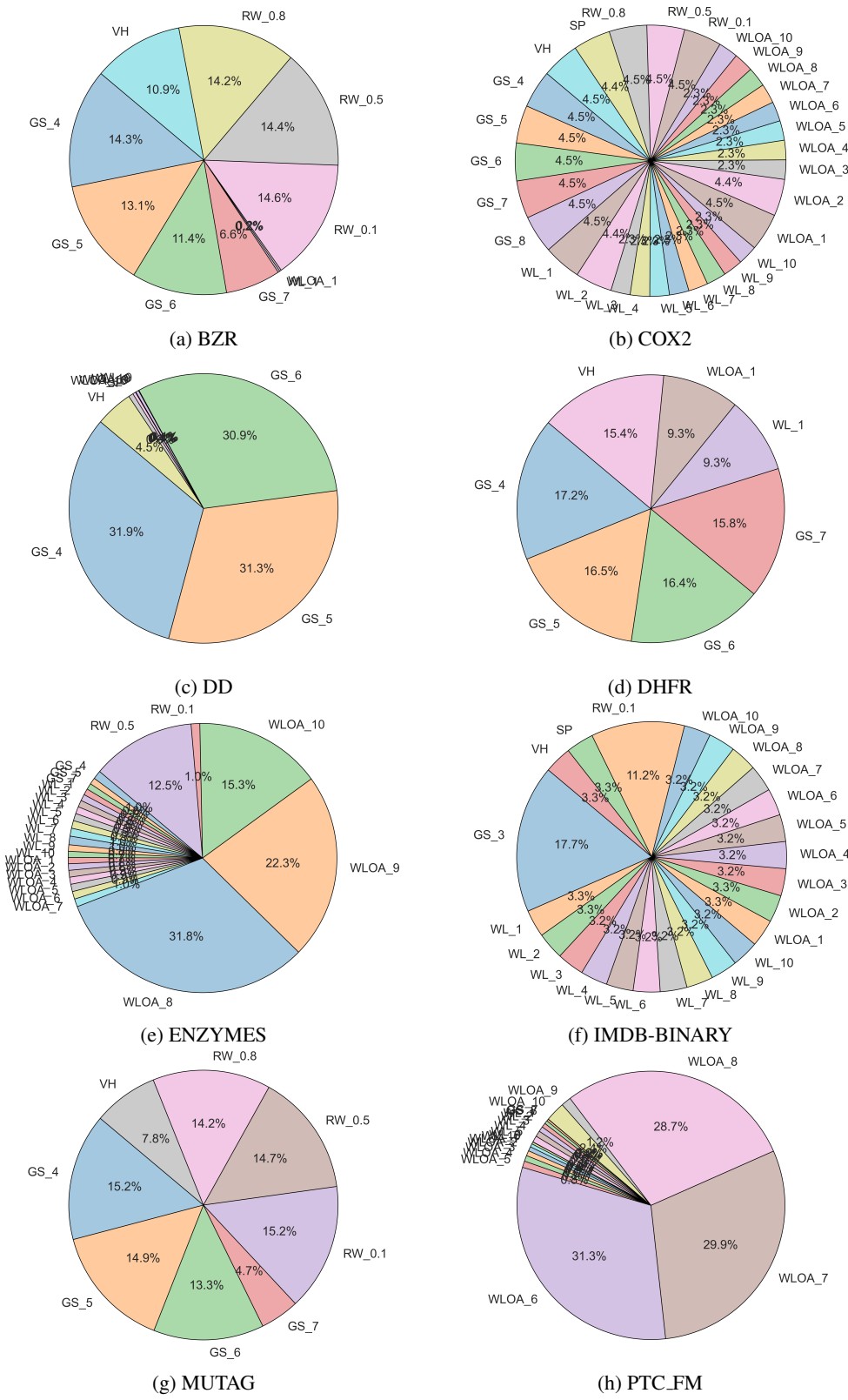

Figure 9: Learned Kernel Weights of UMKL-G on All Benchmark Datasets. *Kernel names are shown with their respective hyperparameters.*

## G.3 SENSITIVITY ANALYSIS

In this section, we provide the full sensitivity analysis of hyperparameter $o$ and initial weight $w$ in the following tables, where we initialize the weights using four different methods:

1. Each weight is set to $1/M$ (default).

2. $1 - \lambda/\sum \lambda$, where $\lambda = \lambda_{[k+1]} - \lambda_{[k]}$ represents the difference between consecutive eigenvalues of the Laplacian matrix derived from each base kernel. Here, $k$ is the presumed number of groups in the dataset.

3. $\lambda/\sum \lambda$, where $\lambda$ is defined as above.

4. Weights are drawn randomly from a Dirichlet distribution.

Table 7: Sensitivity Analysis on BZR Dataset.

| $o$ | Initial $w$ | ACC | NMI | ARI |
|---|---|---|---|---|
| 2 | $1/M$ | 0.9432 | 0.0279 | 0.0811 |
| 2 | $1 - \lambda/\sum \lambda$ | 0.9432 | 0.0279 | 0.0811 |
| 2 | $\lambda/\sum \lambda$ | 0.9580 | 0.0260 | 0.0787 |
| 2 | Random | 0.9418 | 0.0309 | 0.0859 |
| 3 | $1/M$ | 0.9432 | 0.0279 | 0.0811 |
| 3 | $1 - \lambda/\sum \lambda$ | 0.9432 | 0.0279 | 0.0811 |
| 3 | $\lambda/\sum \lambda$ | 0.9580 | 0.0260 | 0.0787 |
| 3 | Random | 0.9418 | 0.0309 | 0.0859 |
| 4 | $1/M$ | 0.9432 | 0.0279 | 0.0812 |
| 4 | $1 - \lambda/\sum \lambda$ | 0.9432 | 0.0279 | 0.0811 |
| 4 | $\lambda/\sum \lambda$ | 0.9580 | 0.0260 | 0.0787 |
| 4 | Random | 0.9418 | 0.0309 | 0.0859 |

Table 8: Sensitivity Analysis on COX2 Dataset.

| $o$ | Initial $w$ | ACC | NMI | ARI |
|---|---|---|---|---|
| 2 | $1/M$ | 0.8009 | 0.0045 | 0.0247 |
| 2 | $1 - \lambda/\sum \lambda$ | 0.8009 | 0.0045 | 0.0247 |
| 2 | $\lambda/\sum \lambda$ | 0.7580 | 0.0046 | 0.0247 |
| 2 | Random | 0.8009 | 0.0045 | 0.0247 |
| 3 | $1/M$ | 0.8009 | 0.0045 | 0.0247 |
| 3 | $1 - \lambda/\sum \lambda$ | 0.8009 | 0.0045 | 0.0247 |
| 3 | $\lambda/\sum \lambda$ | 0.7580 | 0.0046 | 0.0247 |
| 3 | Random | 0.8009 | 0.0045 | 0.0247 |
| 4 | $1/M$ | 0.8009 | 0.0045 | 0.0247 |
| 4 | $1 - \lambda/\sum \lambda$ | 0.8009 | 0.0045 | 0.0247 |
| 4 | $\lambda/\sum \lambda$ | 0.7580 | 0.0046 | 0.0247 |
| 4 | Random | 0.8009 | 0.0045 | 0.0247 |

Table 9: Sensitivity Analysis on DHFR Dataset.

| $o$ | Initial $w$ | ACC | NMI | ARI |
|---|---|---|---|---|
| 2 | $1/M$ | 0.6984 | 0.0111 | 0.0180 |
| 2 | $1 - \lambda/\sum\lambda$ | 0.6984 | 0.0111 | 0.0180 |
| 2 | $\lambda/\sum\lambda$ | 0.6653 | 0.0111 | 0.0180 |
| 2 | Random | 0.6865 | 0.0115 | 0.0187 |
| 3 | $1/M$ | 0.6984 | 0.0111 | 0.0180 |
| 3 | $1 - \lambda/\sum\lambda$ | 0.6984 | 0.0111 | 0.0180 |
| 3 | $\lambda/\sum\lambda$ | 0.6653 | 0.0111 | 0.0180 |
| 3 | Random | 0.6865 | 0.0115 | 0.0187 |
| 4 | $1/M$ | 0.6984 | 0.0111 | 0.0180 |
| 4 | $1 - \lambda/\sum\lambda$ | 0.6984 | 0.0111 | 0.0180 |
| 4 | $\lambda/\sum\lambda$ | 0.6653 | 0.0111 | 0.0180 |
| 4 | Random | 0.6865 | 0.0115 | 0.0187 |

Table 10: Sensitivity Analysis on ENZYMES Dataset.

| $o$ | Initial $w$ | ACC | NMI | ARI |
|---|---|---|---|---|
| 2 | $1/M$ | 0.2983 | 0.0645 | 0.0396 |
| 2 | $1 - \lambda/\sum\lambda$ | 0.2983 | 0.0646 | 0.0393 |
| 2 | $\lambda/\sum\lambda$ | 0.2833 | 0.0662 | 0.0352 |
| 2 | Random | 0.3050 | 0.0670 | 0.0400 |
| 3 | $1/M$ | 0.2983 | 0.0645 | 0.0396 |
| 3 | $1 - \lambda/\sum\lambda$ | 0.2983 | 0.0641 | 0.0393 |
| 3 | $\lambda/\sum\lambda$ | 0.2833 | 0.0662 | 0.0338 |
| 3 | Random | 0.3050 | 0.0669 | 0.0398 |
| 4 | $1/M$ | 0.2983 | 0.0648 | 0.0399 |
| 4 | $1 - \lambda/\sum\lambda$ | 0.2983 | 0.0650 | 0.0393 |
| 4 | $\lambda/\sum\lambda$ | 0.2833 | 0.0662 | 0.0346 |
| 4 | Random | 0.3050 | 0.0669 | 0.0398 |

Table 11: Sensitivity Analysis on IMDB-BINARY Dataset.

| $o$ | Initial $w$ | ACC | NMI | ARI |
|---|---|---|---|---|
| 2 | $1/M$ | 0.5590 | 0.0159 | 0.0132 |
| 2 | $1 - \lambda/\sum\lambda$ | 0.5590 | 0.0239 | 0.0132 |
| 2 | $\lambda/\sum\lambda$ | 0.5620 | 0.0174 | 0.0147 |
| 2 | Random | 0.5600 | 0.0239 | 0.0137 |
| 3 | $1/M$ | 0.5590 | 0.0159 | 0.0132 |
| 3 | $1 - \lambda/\sum\lambda$ | 0.5590 | 0.0239 | 0.0132 |
| 3 | $\lambda/\sum\lambda$ | 0.5620 | 0.0174 | 0.0147 |
| 3 | Random | 0.5600 | 0.0239 | 0.0137 |
| 4 | $1/M$ | 0.5590 | 0.0159 | 0.0132 |
| 4 | $1 - \lambda/\sum\lambda$ | 0.5580 | 0.0239 | 0.0128 |
| 4 | $\lambda/\sum\lambda$ | 0.5620 | 0.0174 | 0.0147 |
| 4 | Random | 0.5600 | 0.0239 | 0.0137 |

Table 13: Sensitivity Analysis on PTC_FM Dataset.

| $o$ | Initial $w$ | ACC | NMI | ARI |
|---|---|---|---|---|
| 2 | $1/M$ | 0.8825 | 0.0394 | 0.0637 |
| 2 | $1 - \lambda/\sum \lambda$ | 0.8825 | 0.0394 | 0.0637 |
| 2 | $\lambda/\sum \lambda$ | 0.9112 | 0.0396 | 0.0637 |
| 2 | Random | 0.8711 | 0.0394 | 0.0637 |
| 3 | $1/M$ | 0.8825 | 0.0394 | 0.0637 |
| 3 | $1 - \lambda/\sum \lambda$ | 0.8825 | 0.0394 | 0.0637 |
| 3 | $\lambda/\sum \lambda$ | 0.9112 | 0.0396 | 0.0637 |
| 3 | Random | 0.8711 | 0.0394 | 0.0637 |
| 4 | $1/M$ | 0.8825 | 0.0394 | 0.0637 |
| 4 | $1 - \lambda/\sum \lambda$ | 0.8797 | 0.0394 | 0.0637 |
| 4 | $\lambda/\sum \lambda$ | 0.9112 | 0.0396 | 0.0637 |
| 4 | Random | 0.8711 | 0.0394 | 0.0637 |

Table 12: Sensitivity Analysis on MUTAG Dataset.

| $o$ | Initial $w$ | ACC | NMI | ARI |
|---|---|---|---|---|
| 2 | $1/M$ | 0.8455 | 0.2950 | 0.3389 |
| 2 | $1 - \lambda/\sum \lambda$ | 0.8455 | 0.2950 | 0.3389 |
| 2 | $\lambda/\sum \lambda$ | 0.8239 | 0.1743 | 0.2533 |
| 2 | Random | 0.8340 | 0.2469 | 0.3020 |
| 3 | $1/M$ | 0.8455 | 0.2950 | 0.3389 |
| 3 | $1 - \lambda/\sum \lambda$ | 0.8455 | 0.2950 | 0.3389 |
| 3 | $\lambda/\sum \lambda$ | 0.8239 | 0.1743 | 0.2533 |
| 3 | Random | 0.8340 | 0.2469 | 0.3020 |
| 4 | $1/M$ | 0.8455 | 0.2950 | 0.3389 |
| 4 | $1 - \lambda/\sum \lambda$ | 0.8455 | 0.2950 | 0.3389 |
| 4 | $\lambda/\sum \lambda$ | 0.8239 | 0.1743 | 0.2533 |
| 4 | Random | 0.8340 | 0.2469 | 0.3020 |

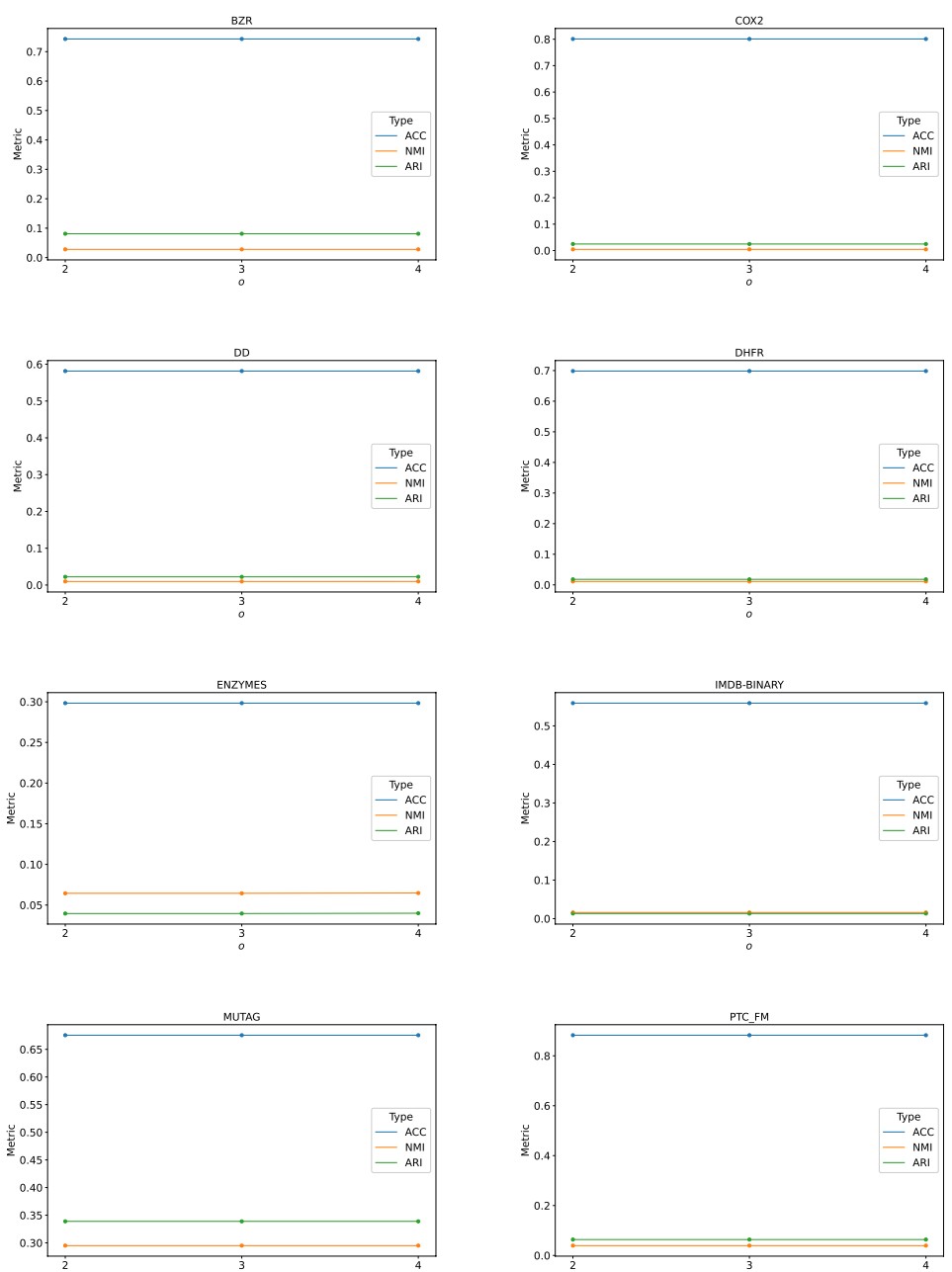

Figure 10: Sensitivity Analysis of Parameter $o$ Across All Benchmark Datasets.

### G.4 CONVERGENCE ANALYSIS

In this section, we demonstrate the smooth convergence plots, which validate Theorem 3.

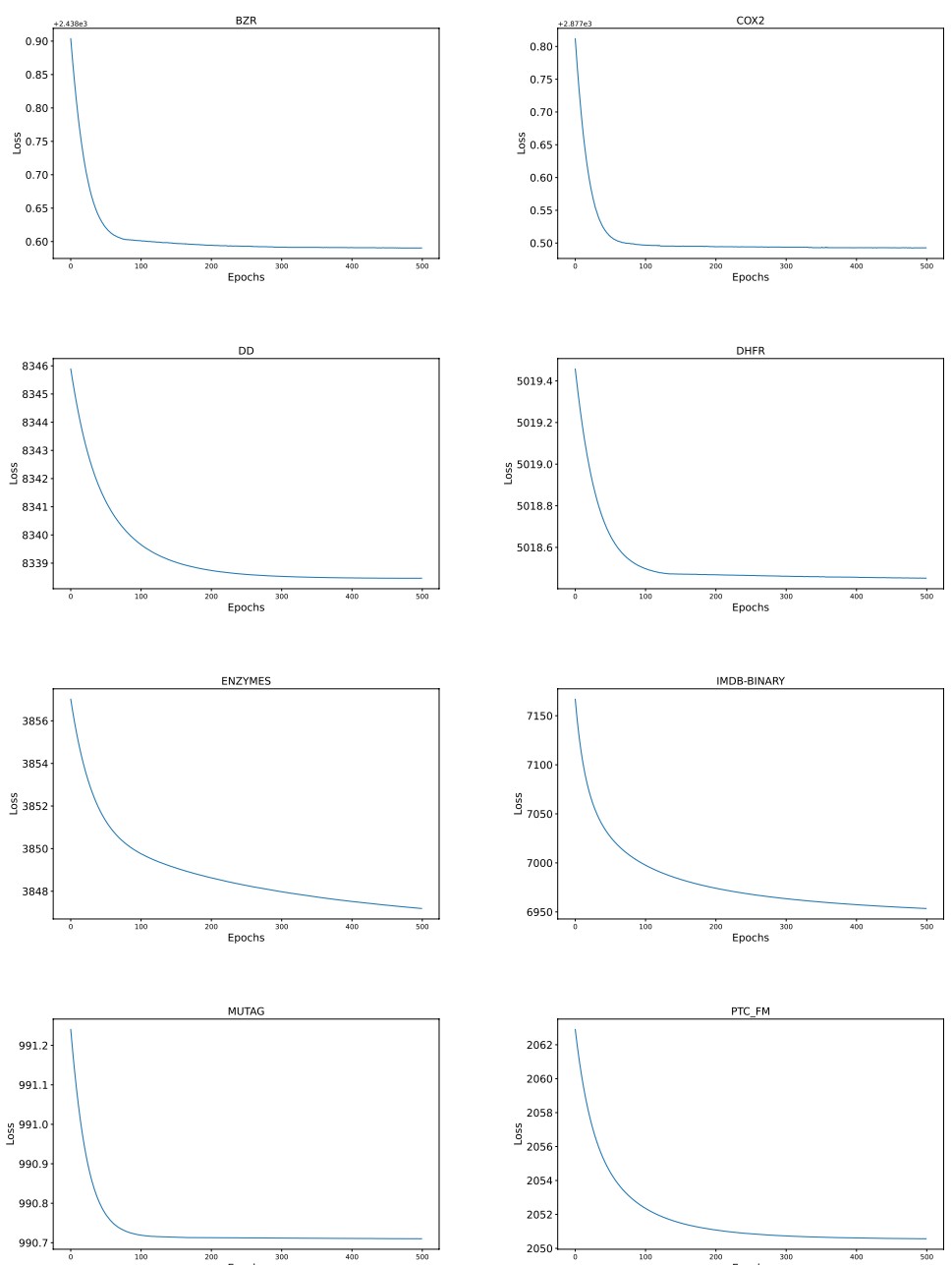

Figure 11: Training Losses Across All Benchmark Datasets.

## G.5 ROBUSTNESS TO PERTURBATION

In this section, we provide the robustness analysis across all datasets, where we perturb the base kernels by adding Gaussian noise $\mathcal{N}(0, \sigma^2)$. As the tables show below, noise perturbations have negligible effects on performance across datasets, which validate Theorem 4.

Table 14: Evaluation of Perturbation $\mathcal{N}(0, \sigma^2)$ in Base Kernels on BZR Dataset.

| $\sigma$ | ACC | NMI | ARI |
|---|---|---|---|
| 0.01 | 0.9432 | 0.0279 | 0.0811 |
| 0.001 | 0.9432 | 0.0279 | 0.0811 |
| – | 0.9432 | 0.0279 | 0.0812 |

Table 15: Evaluation of Perturbation $\mathcal{N}(0, \sigma^2)$ in Base Kernels on COX2 Dataset.

| $\sigma$ | ACC | NMI | ARI |
|---|---|---|---|
| 0.01 | 0.8030 | 0.0045 | 0.0247 |
| 0.001 | 0.8030 | 0.0045 | 0.0247 |
| – | 0.8009 | 0.0045 | 0.0247 |

Table 16: Evaluation of Perturbation $\mathcal{N}(0, \sigma^2)$ in Base Kernels on DD Dataset.

| $\sigma$ | ACC | NMI | ARI |
|---|---|---|---|
| 0.01 | 0.5823 | 0.0100 | 0.0215 |
| 0.001 | 0.5815 | 0.0099 | 0.0224 |
| – | 0.5815 | 0.0098 | 0.0224 |

Table 17: Evaluation of Perturbation $\mathcal{N}(0, \sigma^2)$ in Base Kernels on DHFR Dataset.

| $\sigma$ | ACC | NMI | ARI |
|---|---|---|---|
| 0.01 | 0.7037 | 0.0109 | 0.0173 |
| 0.001 | 0.6997 | 0.0111 | 0.0180 |
| – | 0.6984 | 0.0111 | 0.0180 |

Table 18: Evaluation of Perturbation $\mathcal{N}(0, \sigma^2)$ in Base Kernels on ENZYMES Dataset.

| $\sigma$ | ACC | NMI | ARI |
|---|---|---|---|
| 0.01 | 0.2967 | 0.0620 | 0.0373 |
| 0.001 | 0.2983 | 0.0650 | 0.0399 |
| – | 0.2983 | 0.0648 | 0.0399 |

Table 19: Evaluation of Perturbation $\mathcal{N}(0, \sigma^2)$ in Base Kernels on IMDB-BINARY Dataset.

| $\sigma$ | ACC | NMI | ARI |
|---|---|---|---|
| 0.01 | 0.5590 | 0.0159 | 0.0132 |
| 0.001 | 0.5590 | 0.0159 | 0.0132 |
| – | 0.5590 | 0.0159 | 0.0132 |

Table 20: Evaluation of Perturbation $\mathcal{N}(0, \sigma^2)$ in Base Kernels on MUTAG Dataset.

| $\sigma$ | ACC | NMI | ARI |
|---|---|---|---|
| 0.01 | 0.8455 | 0.3028 | 0.3514 |
| 0.001 | 0.8455 | 0.2950 | 0.3389 |
| – | 0.8455 | 0.2950 | 0.3389 |

Table 21: Evaluation of Perturbation $\mathcal{N}(0, \sigma^2)$ in Base Kernels on PTC_FM Dataset.

| $\sigma$ | ACC | NMI | ARI |
|---|---|---|---|
| 0.01 | 0.8825 | 0.0394 | 0.0637 |
| 0.001 | 0.8825 | 0.0394 | 0.0637 |
| – | 0.8825 | 0.0394 | 0.0637 |

## G.6 GENERALIZATION

In this section, we demonstrate the generalizability of our methods on those unseen data, by splitting out 20% test data. As the tables show below, test performances are closely aligned with those on the full dataset, empirically validating the theoretical bounds in Theorem 6.

Table 22: Generalization Evaluation on BZR Dataset.

| Dataset | ACC | NMI | ARI |
|---|---|---|---|
| Test | 0.9407 | 0.0329 | 0.0886 |
| All | 0.9432 | 0.0279 | 0.0812 |

Table 23: Generalization Evaluation on DD Dataset.

| Dataset | ACC | NMI | ARI |
|---|---|---|---|
| Test | 0.5658 | 0.0076 | 0.0148 |
| All | 0.5815 | 0.0098 | 0.0224 |

Table 24: Generalization Evaluation on COX2 Dataset.

| Dataset | ACC | NMI | ARI |
|---|---|---|---|
| Test | 0.8043 | 0.0048 | 0.0258 |
| All | 0.8009 | 0.0045 | 0.0247 |

Table 25: Generalization Evaluation on DHFR Dataset.

| Dataset | ACC | NMI | ARI |
|---|---|---|---|
| Test | 0.7053 | 0.0125 | 0.0193 |
| All | 0.6984 | 0.0111 | 0.0180 |

Table 26: Generalization Evaluation on ENZYMES Dataset.

| Dataset | ACC | NMI | ARI |
|---------|--------|--------|--------|
| Test | 0.3063 | 0.0785 | 0.0422 |
| All | 0.2983 | 0.0648 | 0.0399 |

Table 27: Generalization Evaluation on IMDB-BINARY Dataset.

| Dataset | ACC | NMI | ARI |
|---------|--------|--------|--------|
| Test | 0.5550 | 0.0152 | 0.0112 |
| All | 0.5590 | 0.0159 | 0.0132 |

Table 28: Generalization Evaluation on MUTAG Dataset.

| Dataset | ACC | NMI | ARI |
|---------|--------|--------|--------|
| Test | 0.8392 | 0.1289 | 0.1756 |
| All | 0.8455 | 0.2950 | 0.3389 |

Table 29: Generalization Evaluation on PTC_FM Dataset.

| Dataset | ACC | NMI | ARI |
|---------|--------|--------|--------|
| Test | 0.8853 | 0.0568 | 0.0747 |
| All | 0.8825 | 0.0394 | 0.0637 |

## G.7 VISUALIZATION OF LEARNING TRAJECTORY

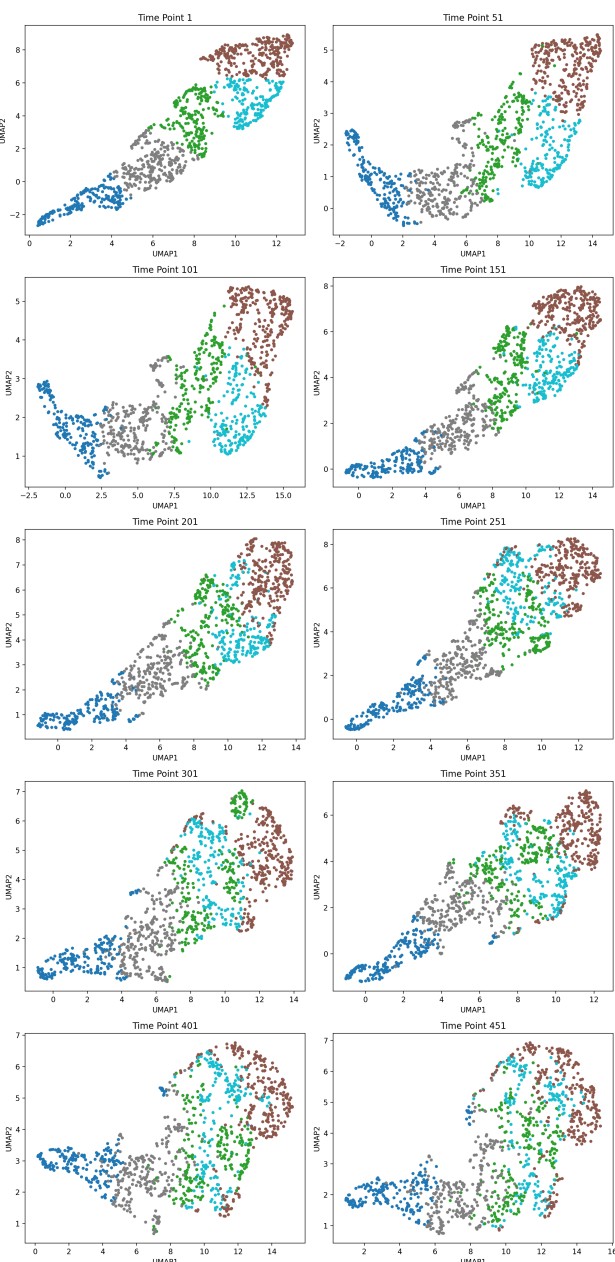

Figure 12: UMAP (McInnes et al., 2018) Visualization of the Learning Trajectory of $Q$ on the DD Dataset with target $P^{(o)}$. *Each point represents a graph with colors indicating the local connectivity at the initial stage (i.e., epoch = 0). Time points are illustrated at various epochs to depict the progression over time.*

