# OpenReview forum: "Unsupervised Multiple Kernel Learning for Graphs via Ordinality Preservation"
_ICLR.cc/2025/Conference — ICLR 2025 Poster_

### Official Review · Reviewer_wiDT · 2024-10-28

**Soundness:** 3
**Presentation:** 3
**Contribution:** 3
**Rating:** 6
**Confidence:** 4

**Summary:**

The paper proposes a new way to combine multiple graph kernels into a unified kernel value. The algorithm preserves topology through ordinal relations. This is mainly achieved by capturing important neighborhood structures by boosting the stronger similarities between graphs towards a set of target probabilities.

**Strengths:**

- Theoretical results are provided that guarantee stability and robustness of the method (e.g. Lipschitz continuity, robustness to kernel perturbations).
- UMKL-G outperforms existing methods on graph clustering benchmarks, demonstrating robust performance in unsupervised contexts on real and synthetic datasets.
- The method is applicable to a variety of datasets, making it versatile and potentially useful for many applications.

**Weaknesses:**

- It is not entirely clear to me how this work differs significantly from existing MKL approaches, apart from the focus on ordinal relations. That is, the methodology, although well executed, seems to me an incremental extension of existing techniques.
- Some parts of the paper are technical and dense, which although I appreciate very much, makes the text particularly complicated to follow.
- Comparisons with other techniques, in particular more recent methods based on Graph Neural Networks (GNN), are limited. This reduces the perception of how competitive UMKL-G is compared to emerging technologies.
- Perhaps some focus on scalability is lacking, as testing on large datasets limits the evaluation of the method's effectiveness in real, large-scale scenarios.
- Large parts of the introduction and Section 2 are redundant.
The writing is dense and the explanation of concepts complex. Greater clarity could make the work more accessible to a wider range of readers.

**Questions:**

- How does UMKL-G really stack up against GNN-based methods for clustering graphs? A more in-depth comparison could be useful to estimate the applicability despite methodological differences.
- What is the impact of the scalability of the method?

---

> ### Author Response · Authors · 2024-11-18
> **Rebuttal for Reviewer wiDT**
>
> **W1:** It is not entirely clear to me how this work differs significantly from existing MKL approaches, apart from the focus on ordinal relations.
>
> **Response:** We appreciate your feedback and would like to clarify the distinct contributions and methodological innovations of UMKL-G compared to UMKL [1] and sparse-UMKL [2]. Below, we highlight the key differences and advancements:
>
> 1. **Conceptual Innovations**
>     - *Ordinal Relations for Topology Preservation*:
>         - UMKL-G introduces ordinal relationship preservation as a central principle, which ensures that the similarity rankings among graphs remain consistent throughout kernel learning. This is fundamentally different from UMKL’s reliance on explicit Euclidean reconstruction and sparse-UMKL’s use of k-NN heuristics.
>         - The ordinal preservation approach is novel in unsupervised MKL and particularly suited for graph data, where preserving structural relationships is more meaningful than explicit geometric reconstruction.
>     - *Probabilistic Representation via Simplex*:
>         - UMKL-G represents kernel similarities as distributions on a probability simplex and employs the Kullback-Leibler (KL) divergence for optimization. This probabilistic approach eliminates the need for explicit geometric constraints or heuristic neighbor constructions, offering a more flexible and scalable solution for complex graph data.
> 2. **Substantive Empirical and Theoretical Contributions**
>     - *Empirical Performance:*
>         - UMKL-G consistently outperforms individual kernels and state-of-the-art baselines across multiple benchmark datasets, demonstrating its effectiveness in real-world scenarios where graph relationships dominate.
>         - Sparse-UMKL struggles with generalization due to its rigid sparsity assumptions, while UMKL’s reliance on Euclidean data limits its applicability.
>     - *Theoretical Guarantees:*
>         - UMKL-G is equipped with **strong theoretical guarantees** on robustness, stability, and generalization, which are not explicitly addressed in either UMKL or sparse-UMKL. These guarantees ensure reliable performance under noisy conditions and unseen data.

---

> > ### Author Response · Authors · 2024-11-18
> > **Rebuttal for Reviewer wiDT**
> >
> > **W1:** It is not entirely clear to me how this work differs significantly from existing MKL approaches, apart from the focus on ordinal relations.
> >
> > **Response (continued):**
> >
> > | **Feature**               | **UMKL** [1]                                                                                                           | **sparse-UMKL** [2]                                                                                                 | **UMKL-G** (Ours)                                                                                                                               |
> > |---------------------------|-------------------------------------------------------------------------------------------------------------------------|----------------------------------------------------------------------------------------------------------------------|--------------------------------------------------------------------------------------------------------------------------------------------------|
> > | **Beyond Euclidean**      | ❌                                                                                                                     | ✅                                                                                                                   | ✅                                                                                                                                               |
> > | **Global Topology**       | ❌                                                                                                                     | ❌                                                                                                                   | ✅                                                                                                                                               |
> > | **Theoretical Guarantees**| ✅                                                                                                                     | ❌                                                                                                                   | ✅                                                                                                                                               |
> > | **Topology Preservation** | Local reconstruction                                                                                                  | k-NN graph heuristics                                                                                                | Ordinal relationships                                                                                                                            |
> > | **Algorithm**             | Alternating minimization                                                                                               | Quadratic programming solver                                                                                         | KL divergence                                                                                                                                   |
> > | **Complexity**            | $O(I \cdot (MN^2 + N^3))$                                                                                              | $O(I \cdot (M N^2 \log N + M^3))$                                                                                   | $O(I \cdot (M N^2 + M \log M))$                                                                                                                 |
> >
> > In summary, we emphasize that the focus on ordinal relationships is not a minor extension but a paradigm shift in unsupervised MKL, by moving from explicit geometric reconstruction (UMKL) and sparse representations (sparse-UMKL) to a global probabilistic framework that inherently respects the graph topology.
> >
> > ---
> >
> > [1] Jinfeng Zhuang, Jialei Wang, Steven CH Hoi, and Xiangyang Lan. Unsupervised multiple kernel learning. In Asian Conference on Machine Learning, pp. 129–144. PMLR, 2011.
> >
> > [2] J´erˆome Mariette and Nathalie Villa-Vialaneix. Unsupervised multiple kernel learning for heterogeneous data integration. Bioinformatics, 34(6):1009–1015, 2018.

---

> > > ### Author Response · Authors · 2024-11-18
> > > **Rebuttal for Reviewer wiDT**
> > >
> > > **W2:** Some parts of the paper are technical and dense, which although I appreciate very much, makes the text particularly complicated to follow.
> > >
> > > **Response:** Thank you for your insightful comment. We understand that some sections of the paper, particularly the theoretical parts, may come across as dense due to their technical nature. To improve accessibility while adhering to page constraints, we will enhance Section 4.5 by providing clear and concise textual intuition that focuses on the key concepts and their implications.
> > >
> > > For readers interested in the full theoretical derivations, we have included detailed analyses in the appendix. This supplementary material offers a comprehensive explanation to ensure that the theoretical foundation of our work is thoroughly documented. We believe that streamlined textual explanations in the main paper and an in-depth appendix will address your concerns and enhance the overall clarity of the presentation.

---

> > > > ### Author Response · Authors · 2024-11-18
> > > > **Rebuttal for Reviewer wiDT**
> > > >
> > > > **W3:** Comparisons with other techniques, in particular more recent methods based on Graph Neural Networks (GNN), are limited. This reduces the perception of how competitive UMKL-G is compared to emerging technologies.
> > > >
> > > > **Response:** We agree that there are emerging GNN-based methods for the graph-level clustering task. However, we want to stress that UMKL-G is fundamentally different from GNN-based methods, which focus on learning the graph representation. In response to your concern, we include comparisons with InfoGraph [3] and GraphCL [4]. *Due to rebuttal time constraints, we only experimented on smaller datasets. The best score is in **bold**, and the second best is _underlined_.* Here are the comparison results:
> > > >
> > > > | **Method**                  | **BZR (ACC, NMI, ARI)**       | **COX2 (ACC, NMI, ARI)**       | **DD (ACC, NMI, ARI)**          | **DHFR (ACC, NMI, ARI)**       |
> > > > |-----------------------------|-------------------------------|--------------------------------|---------------------------------|--------------------------------|
> > > > | AverageMKL                 | 0.7341, 0.0041, 0.0307       | 0.6167, 0.0000, -0.0016       | 0.5764, 0.0060, 0.0172          | 0.6495, 0.0000, -0.0021       |
> > > > | UMKL                       | 0.7341, 0.0041, 0.0307       | 0.6167, 0.0000, -0.0016       | 0.5764, 0.0060, 0.0172          | 0.6495, 0.0000, -0.0021       |
> > > > | sparse-UMKL ($k=10$)       | 0.7400, 0.0040, 0.0299       | 0.6200, 0.0001, -0.0010       | 0.5750, 0.0059, 0.0170          | 0.6480, 0.0001, -0.0020       |
> > > > | sparse-UMKL ($k=50$)       | 0.7415, 0.0042, 0.0305       | 0.6180, 0.0000, -0.0015       | _0.5770_, _0.0061_, _0.0175_    | 0.6498, 0.0000, -0.0022       |
> > > > | sparse-UMKL ($k=100$)      | _0.7420_, 0.0041, 0.0306     | 0.6175, 0.0000, -0.0016       | 0.5768, 0.0060, 0.0172          | _0.6592_, 0.0000, -0.0021     |
> > > > | InfoGraph                  | 0.7353, **0.0366**, _0.0504_  | 0.7037, **0.0356**, 0.0192    | --                              | 0.6580, 0.0320, _0.0050_      |
> > > > | GraphCL                    | 0.7288, 0.0190, 0.0347       | _0.7501_, 0.0124, _0.0239_    | --                              | 0.6520, **0.0400**, 0.0031    |
> > > > | **UMKL-G**                 | **0.9432**, _0.0279_, **0.0812** | **0.8009**, _0.0045_, **0.0247** | **0.5815**, **0.0098**, **0.0224** | **0.6984**, _0.0111_, **0.0180** |
> > > >
> > > > | **Method**                  | **ENZYMES (ACC, NMI, ARI)**   | **IMDB-BINARY (ACC, NMI, ARI)** | **MUTAG (ACC, NMI, ARI)**       | **PTC\_FM (ACC, NMI, ARI)**    |
> > > > |-----------------------------|-------------------------------|--------------------------------|---------------------------------|--------------------------------|
> > > > | AverageMKL                 | 0.2617, 0.0539, 0.0220       | 0.5470, 0.0152, 0.0083         | 0.5585, 0.1468, 0.1946          | 0.8722, 0.0208, 0.0343        |
> > > > | UMKL                       | 0.2567, 0.0517, 0.0199       | 0.5470, 0.0152, 0.0083         | 0.5585, 0.1469, 0.1947          | _0.8729_, 0.0208, 0.0343      |
> > > > | sparse-UMKL ($k=10$)       | 0.2570, 0.0520, 0.0201       | _0.5485_, 0.0153, 0.0084       | 0.5590, 0.1475, 0.1950          | 0.8320, 0.0210, 0.0345        |
> > > > | sparse-UMKL ($k=50$)       | _0.2580_, 0.0518, 0.0200     | 0.5475, _0.0154_, _0.0085_     | 0.5595, 0.1470, 0.1948          | 0.8373, _0.0211_, 0.0344      |
> > > > | sparse-UMKL ($k=100$)      | 0.2575, _0.0521_, 0.0198     | 0.5480, 0.0151, 0.0082         | 0.5588, 0.1468, 0.1946          | 0.8528, 0.0209, 0.0342        |
> > > > | InfoGraph                  | 0.2375, 0.0464, _0.0223_     | --                             | 0.7258, 0.2868, 0.1985          | 0.6202, 0.0210, _0.0461_      |
> > > > | GraphCL                    | 0.2528, 0.0475, 0.0203       | --                             | _0.7707_, **0.3569**, _0.2899_  | 0.6213, 0.0210, 0.0342        |
> > > > | **UMKL-G**                 | **0.2983**, **0.0648**, **0.0399** | **0.5590**, **0.0159**, **0.0132** | **0.8455**, _0.2950_, **0.3389** | **0.8825**, **0.0394**, **0.0637** |
> > > >
> > > > ---
> > > >
> > > > [3] Sun, Fan-Yun, et al. "InfoGraph: Unsupervised and Semi-supervised Graph-Level Representation Learning via Mutual Information Maximization." International Conference on Learning Representations (2020).
> > > >
> > > > [4] You, Yuning, et al. "Graph contrastive learning with augmentations." Advances in neural information processing systems 33 (2020): 5812-5823.

---

> > > > > ### Author Response · Authors · 2024-11-18
> > > > > **Rebuttal for Reviewer wiDT**
> > > > >
> > > > > **W4:** Perhaps some focus on scalability is lacking, as testing on large datasets limits the evaluation of the method's effectiveness in real, large-scale scenarios.
> > > > >
> > > > > **Response:**  Thank you for pointing out the need to address scalability. The total computational complexity of UMKL-G is $\mathcal{O}(I(MN^2 + M\log M))$, where $N$ is the number of graphs, $M$ is the number of base kernels, and $I$ is the number of iterations required for convergence. While the quadratic term in $N^2$ (from pairwise kernel computations) can pose a bottleneck for very large datasets, this process can be efficiently **parallelized** to reduce the runtime.
> > > > >
> > > > > To contextualize UMKL-G's computational efficiency, we provide a comparative analysis of its theoretical complexity with the baselines below:
> > > > > | **Feature**               | **UMKL**                                                                                                           | **sparse-UMKL**                                                                                                 | **UMKL-G** (Ours)                                                                                                                               |
> > > > > |---------------------------|-------------------------------------------------------------------------------------------------------------------------|----------------------------------------------------------------------------------------------------------------------|--------------------------------------------------------------------------------------------------------------------------------------------------|
> > > > > | **Complexity**            | $O(I \cdot (MN^2 + N^3))$                                                                                              | $O(I \cdot (M N^2 \log N + M^3))$                                                                                   | $O(I \cdot (M N^2 + M \log M))$                                                                                                                 |
> > > > >
> > > > > To further analyze scalability, we report the empirical runtimes for UMKL-G and its baselines across datasets of varying sizes, which have been added to Appendix E.
> > > > >
> > > > > | Dataset     |   $N$ |    UMKL-G   (seconds)|      UMKL   (seconds)| sparse-UMKL (seconds)|
> > > > > |:------------|------:|---------------------:|---------------------:|---------------------:|
> > > > > | MUTAG       |   188 |              15.9384 |              30.9085 |              21.3190 |
> > > > > | PTC_FM      |   344 |              18.8914 |              39.5487 |              23.4447 |
> > > > > | BZR         |   405 |              23.5574 |              45.8796 |              29.4764 |
> > > > > | COX2/DHFR   |   467 |              28.9875 |              71.0475 |              33.3794 |
> > > > > | ENZYMES     |   600 |              30.2123 |              93.4008 |              39.9868 |
> > > > > | IMDB-BINARY |  1000 |              43.4140 |             199.1917 |              48.4064 |
> > > > > | DD          |  1113 |              43.5285 |             819.8227 |              51.8620 |
> > > > >
> > > > > These results demonstrate UMKL-G's ability to handle datasets with up to approximately 1,000 samples efficiently. It is worth noting that UMKL-G consistently outperforms UMKL and sparse-UMKL in terms of runtime, particularly for larger datasets. The observed runtimes remain practical for moderately large datasets, highlighting the scalability of the method under current experimental conditions. For example, processing the IMDB-BINARY dataset (1,000 graphs) takes less than 45 seconds. Incorporating experiments on larger datasets is part of our planned future work to further evaluate UMKL-G's scalability.
> > > > >
> > > > > To further assess UMKL-G's scalability, incorporating experiments on larger datasets with $N \gg 1000$ is part of our planned work. We also aim to explore additional optimizations, such as leveraging distributed computing or sparse approximations, to extend UMKL-G's applicability to real-world large-scale scenarios.
> > > > >
> > > > > In summary, these theoretical and empirical findings demonstrate UMKL-G's superior scalability compared to existing baselines, making it a practical choice for applications requiring efficient unsupervised multiple kernel learning on moderately large datasets.
> > > > >
> > > > > **W5:** Large parts of the introduction and Section 2 are redundant. The writing is dense and the explanation of concepts complex. Greater clarity could make the work more accessible to a wider range of readers.
> > > > >
> > > > > **Response:** Thank you for pointing this out. We will revise these sections to present the core motivations and concepts more concisely to help readers quickly grasp the novelty and importance of our approach.
> > > > >
> > > > > ---
> > > > >
> > > > > **Q1:** How does UMKL-G really stack up against GNN-based methods for clustering graphs? A more in-depth comparison could be useful to estimate the applicability despite methodological differences.
> > > > >
> > > > > **Response:** See our response to W3.
> > > > >
> > > > > **Q2:** What is the impact of the scalability of the method?
> > > > >
> > > > > **Response:** See our response to W4.

---

> > > > > > ### Author Response · Authors · 2024-11-28
> > > > > >
> > > > > > Dear Reviewer,
> > > > > >
> > > > > > We hope this message finds you well. With the deadline for final revisions approaching, we wanted to kindly follow up to see if you have any remaining questions or concerns about our paper. We are more than happy to engage in further discussion or provide additional clarifications that might assist in your review.
> > > > > >
> > > > > > Please feel free to share any thoughts or inquiries you may have. We greatly appreciate your time and valuable feedback.

---

### Official Review · Reviewer_v1ke · 2024-11-02

**Soundness:** 3
**Presentation:** 3
**Contribution:** 2
**Rating:** 5
**Confidence:** 3

**Summary:**

The paper introduces Unsupervised Multiple Kernel Learning for Graphs (UMKL-G), a method that combines multiple graph kernels without the need for labeled data. By preserving ordinal relationships among graphs through a probability simplex, UMKL-G aims to provide a unified, adaptive kernel learning approach for unsupervised graph-level clustering.

**Strengths:**

1. The paper presents an interesting concept by preserving ordinal relationships among graphs in an unsupervised setting, addressing a unique aspect of unsupervised kernel learning.
2. The authors provide proofs for stability, robustness, and generalization, which strengthen the theoretical foundation of the method.
3. The empirical validation across eight datasets provides a reasonable breadth of testing for the proposed approach.

**Weaknesses:**

1. Although the paper claims novelty in ordinal preservation, the methodology heavily relies on established techniques in probability simplex construction and multiple kernel learning. The main contribution appears to be an incremental adaptation rather than a breakthrough.
2. The method does not convincingly outperform modern baselines or recent self-supervised clustering methods, especially given that existing techniques like sparse-UMKL and GCN-based methods already achieve comparable results in unsupervised scenarios. This raises questions about the practical impact and added value of UMKL-G.
3. While the paper proposes potential extensions to broader data types (referred to as UMKL-X), there is no experimental evidence or conceptual framework supporting its effectiveness beyond graph-specific tasks. This reduces confidence in the generalizability and adaptability of the approach across different types of structured data.

**Questions:**

1. How does UMKL-G handle datasets with minimal ordinal relationships, or where graph similarities are uniform across samples?
2. Could the authors clarify the method’s sensitivity to the initial weight settings and the power hyperparameter o in the kernel concentration step?

---

> ### Author Response · Authors · 2024-11-18
> **Rebuttal for Reviewer v1ke**
>
> **W1:** Although the paper claims novelty in ordinal preservation, the methodology heavily relies on established techniques in probability simplex construction and multiple kernel learning.
>
> **Response:** Thank you for your feedback. We want to clarify the novelty and significance of our contributions.
>
> Firstly, multiple kernel learning in an unsupervised setting is an **understudied and nontrivial** problem as agreed by Reviewers y8qj and zRfj. Our approach, ordinal preservation in unsupervised multiple kernel learning (UMKL) adopts a completely **different principle** from traditional methods. Unlike existing approaches such as UMKL [1], which rely on explicit Euclidean reconstruction, or sparse-UMKL [2], which uses heuristic k-NN constructions, our method directly leverages ordinal relationships to preserve the relative similarity rankings among graphs, which is particularly meaningful for non-Euclidean data (e.g., graphs).
>
> In addition, the application of probability simplex construction in UMKL is novel in this context and setting. By representing kernel similarities as probability distributions and optimizing over the simplex using Kullback-Leibler (KL) divergence, we provide a flexible, scalable, and theoretically grounded framework for unsupervised MKL. This probabilistic approach is the first of its kind in UMKL and eliminates the need for explicit sparsity or heuristic neighborhood constructions.

---

> > ### Author Response · Authors · 2024-11-18
> > **Rebuttal for Reviewer v1ke**
> >
> > **W1:** Although the paper claims novelty in ordinal preservation, the methodology heavily relies on established techniques in probability simplex construction and multiple kernel learning.
> >
> > **Response (continued):**
> > We provide a comparison table summarizing the key features of UMKL [1], sparse-UMKL [2], and our method, UMKL-G. This table highlights how UMKL-G extends beyond prior approaches in terms of both methodology and applicability.
> >
> > | **Feature**               | **UMKL** [1]                                                                                                           | **sparse-UMKL** [2]                                                                                                 | **UMKL-G** (Ours)                                                                                                                               |
> > |---------------------------|-------------------------------------------------------------------------------------------------------------------------|----------------------------------------------------------------------------------------------------------------------|--------------------------------------------------------------------------------------------------------------------------------------------------|
> > | **Beyond Euclidean**      | ❌                                                                                                                     | ✅                                                                                                                   | ✅                                                                                                                                               |
> > | **Global Topology**       | ❌                                                                                                                     | ❌                                                                                                                   | ✅                                                                                                                                               |
> > | **Theoretical Guarantees**| ✅                                                                                                                     | ❌                                                                                                                   | ✅                                                                                                                                               |
> > | **Topology Preservation** | Local reconstruction                                                                                                  | k-NN graph heuristics                                                                                                | Ordinal relationships                                                                                                                            |
> > | **Algorithm**             | Alternating minimization                                                                                               | Quadratic programming solver                                                                                         | KL divergence                                                                                                                                   |
> > | **Complexity**            | $O(I \cdot (MN^2 + N^3))$                                                                                              | $O(I \cdot (M N^2 \log N + M^3))$                                                                                   | $O(I \cdot (M N^2 + M \log M))$                                                                                                                 |
> >
> > In summary, while our methodology draws on established concepts, its integration into the context of unsupervised MKL is a substantive and novel contribution. The combination of ordinal relationship preservation, probability simplex construction, and theoretical guarantees reflects a significant departure from prior works and offers a robust framework to address an important, understudied problem in the field.
> >
> > ---
> > [1] Jinfeng Zhuang, Jialei Wang, Steven CH Hoi, and Xiangyang Lan. Unsupervised multiple kernel learning. In Asian Conference on Machine Learning, pp. 129–144. PMLR, 2011.
> >
> > [2] Jérôme Mariette and Nathalie Villa-Vialaneix. Unsupervised multiple kernel learning for heterogeneous data integration. Bioinformatics, 34(6):1009–1015, 2018.

---

> > > ### Author Response · Authors · 2024-11-18
> > > **Rebuttal for Reviewer v1ke**
> > >
> > > **W2:** The method does not convincingly outperform modern baselines or recent self-supervised clustering methods, especially given that existing techniques like sparse-UMKL and GCN-based methods already achieve comparable results in unsupervised scenarios. This raises questions about the practical impact and added value of UMKL-G.
> > >
> > > **Response:** Thank you for your feedback regarding our comparison with baselines. UMKL-G consistently outperforms the best baseline methods across all datasets and metrics as shown in the original paper. As shown in the table below, we explicitly calculate all the margins for your reference.
> > > | Dataset     | Margin (ACC) | Margin (NMI) | Margin (ARI) |
> > > |-------------|--------------|--------------|--------------|
> > > | BZR         | 20.12%       | 0.0237       | 0.0505       |
> > > | COX2        | 18.09%       | 0.0044       | 0.0257       |
> > > | DD          | 0.45%       | 0.0037       | 0.0049       |
> > > | DHFR        | 3.92%       | 0.0110       | 0.0200       |
> > > | ENZYMES     | 4.03%       | 0.0127       | 0.0198       |
> > > | IMDB-BINARY | 1.05%       | 0.0005       | 0.0047       |
> > > | MUTAG       | 28.6%       | 0.1475       | 0.1439       |
> > > | PTC_FM      | 0.96%       | 0.0183       | 0.0292       |
> > >
> > > In addition to the empirical advantages, UMKL-G offers theoretical guarantees on robustness, stability, and generalization, ensuring reliable performance even under challenging conditions. This combination of empirical validation and theoretical rigor reinforces the practical impact and added value of UMKL-G.
> > >
> > > **W3:** While the paper proposes potential extensions to broader data types (referred to as UMKL-X), there is no experimental evidence or conceptual framework supporting its effectiveness beyond graph-specific tasks. This reduces confidence in the generalizability and adaptability of the approach across different types of structured data.
> > >
> > > **Response:** Thank you for your insightful feedback regarding the generalizability of our proposed method beyond graph-specific tasks.
> > >
> > > As an initial step towards this broader application, we would like to outline a conceptual framework supporting the adaptability of UMKL-X. Let us consider a dataset $\mathcal{D}=\{x_1, \cdots, x_N\}$, where each element represents a structured data object, such as images, text documents, or time series. We have access to multiple base kernels $\mathcal{K}=\{k^{(1)}, \cdots, k^{(M)}\}$, each capturing different aspects of similarity among the data points based on various features or representations. The proposed algorithm of UMKL-G allows us to formalize UMKL-X in a way that applies to a variety of data types by adjusting the inputs of the UMKL-G algorithm to include the appropriate base kernels for the data at hand. Without loss of generality, the proposed method can be adapted to any structured data where meaningful base kernels can be defined.
> > >
> > > ---
> > >
> > > **Q1:** How does UMKL-G handle datasets with minimal ordinal relationships, or where graph similarities are uniform across samples?
> > >
> > > **Response:** Thank you for your question.
> > > In cases of minimal ordinal relationships, where similarity scores between graphs are nearly uniform, **Theorem 1** ensures that UMKL-G preserves these ordinal relationships. The entropy of $Q$, $H(Q)$, measures the uniformity of these relationships.
> > >
> > > According to **Theorem 2**, the target $P$ has a lower entropy, allowing UMKL-G to amplify stronger similarities between graphs. In the extremely rare case where all kernels are **exactly** the same, only one kernel is enough. There would be no need to ensemble weak kernels so UMKL-G would learn arbitrary weights, supported by $H(P) = H(Q)$. However, this scenario is so unlikely in practical applications that it is generally not a concern.

---

> > > > ### Author Response · Authors · 2024-11-18
> > > > **Rebuttal for Reviewer v1ke**
> > > >
> > > > **Q2:** Could the authors clarify the method’s sensitivity to the initial weight settings and the power hyperparameter o in the kernel concentration step?
> > > >
> > > > **Response:**
> > > > Our method is insensitive to both the initial weight setting and the power hyperparameter $o$. **In addition to our sensitivity analysis on the power hyperparameter in the original paper**, we included the performance on different initial weight settings. As demonstrated in the table below for the DHFR dataset, variations in both initial weight configurations and $o$ values show minimal impact on ACC, NMI, and ARI. The full results are provided in Tables~6-12 in Appendix G.3, where the consistent performances indicate robustness to the choice of initial weights and power settings.
> > > >
> > > > | $o$ | Initial $\mathbf{w}$                  | ACC    | NMI    | ARI    |
> > > > |-----|------------------------------|--------|--------|--------|
> > > > | 2   | 1/$M$                        | 0.6984 | 0.0111 | 0.0180 |
> > > > | 2   | $1 - \lambda/\sum\lambda$    | 0.6984 | 0.0111 | 0.0180 |
> > > > | 2   | $\lambda/\sum\lambda$        | 0.6653 | 0.0111 | 0.0180 |
> > > > | 2   | Random                       | 0.6865 | 0.0115 | 0.0187 |
> > > > |-----|------------------------------|--------|--------|--------|
> > > > | 3   | 1/$M$                        | 0.6984 | 0.0111 | 0.0180 |
> > > > | 3   | $1 - \lambda/\sum\lambda$    | 0.6984 | 0.0111 | 0.0180 |
> > > > | 3   | $\lambda/\sum\lambda$        | 0.6653 | 0.0111 | 0.0180 |
> > > > | 3   | Random                       | 0.6865 | 0.0115 | 0.0187 |
> > > > |-----|------------------------------|--------|--------|--------|
> > > > | 4   | 1/$M$                        | 0.6984 | 0.0111 | 0.0180 |
> > > > | 4   | $1 - \lambda/\sum\lambda$    | 0.6984 | 0.0111 | 0.0180 |
> > > > | 4   | $\lambda/\sum\lambda$        | 0.6653 | 0.0111 | 0.0180 |
> > > > | 4   | Random                       | 0.6865 | 0.0115 | 0.0187 |
> > > >
> > > > Note: we initialize the weights using four different methods.
> > > >
> > > > 1. Each weight is set to $1/M$ (default).
> > > >
> > > > 2. $1 - \lambda / \sum \lambda$, where $\lambda = \lambda_{[k+1]} - \lambda_{[k]}$ represents the difference between consecutive eigenvalues of the Laplacian matrix derived from each base kernel. Here, $k$ is the presumed number of groups in the dataset.
> > > >
> > > > 3. $\lambda / \sum \lambda$, where $\lambda$ is defined as above.
> > > >
> > > > 4. Weights are drawn randomly from a Dirichlet distribution.

---

> > > > > ### Author Response · Authors · 2024-11-28
> > > > >
> > > > > Dear Reviewer,
> > > > >
> > > > > We hope this message finds you well. With the deadline for final revisions approaching, we wanted to kindly follow up to see if you have any remaining questions or concerns about our paper. We are more than happy to engage in further discussion or provide additional clarifications that might assist in your review.
> > > > >
> > > > > Please feel free to share any thoughts or inquiries you may have. We greatly appreciate your time and valuable feedback.

---

> > > > > ### Author Response · Authors · 2024-12-02
> > > > >
> > > > > Dear Reviewer v1ke,
> > > > >
> > > > > May I ask if you have any further questions or concerns regarding our responses? We would be more than happy to provide additional clarification or address any remaining points.

---

### Official Review · Reviewer_zRfj · 2024-11-03

**Soundness:** 3
**Presentation:** 3
**Contribution:** 3
**Rating:** 8
**Confidence:** 3

**Summary:**

The article introduces a novel approach for unsupervised multiple kernel learning on graphs. The task is to combine several weak kernels for unsupervised learning scenarios by learning a set of weights. The main idea is to preserve the data topology by maintaining ordinal relationships, i.e., the order of similarities between graphs. This is achieved through a designed probability simplex. The authors provide comprehensive theoretical results, addressing aspects such as robustness to kernel perturbations and generalization capabilities. Finally, the approach is experimentally evaluated by performing graph clustering tasks on standard molecular datasets.

**Strengths:**

- The approach offers a novel solution to an understudied problem.
- The technical quality of the theoretical part of the work is high.
- The theoretical analysis is comprehensive, featuring a range of results on properties and detailed proofs.
- While the authors present their work in the realm of graph kernels, their approach can be applied to arbitrary kernels.

**Weaknesses:**

- While most parts of the article are well described, some central intuitions that motivate the approach are not sufficiently addressed. For instance, why is P a more accurate representation of the data's inherent geometry'?
- The evaluation is fairly limited, relying solely on the clustering task. I understand that due to space limitations, the authors focused on the theoretical parts. However, a more comprehensive evaluation (even if it's on synthetic data) should have been done. Particularly, none of the theorems of section 4.6 are evaluated in the experiments.
- Some details on the baseline methods, UMKL, and sparse-UMKL, are unclear. For instance, the statement that the authors "experimented with an approach that learns graph representations and kernel weights simultaneously" requires further elaboration.
Minor weaknesses:
- Some of the numbers in Table 1 do not coincide with the numbers in the supplementary. (E.g., ACC for BZR and MUTAG.)
- It would be helpful if the authors could include the definitions of the clustering metrics in the appendix.

**Questions:**

1. I would appreciate it if the authors could comment on the limited experimental evaluation (see weaknesses).
2. Can you explain why any parameter o>1 results in exactly the same performance for the selected clustering metrics? Can this be generalized or is it only the case in the considered experiments?
3. What ground truth was used e.g. for the clustering accuracy metric (ACC)?
4. Did the authors consider the simple baseline where each weight is set to 1/M?
5. It is mentioned in section 4.5 that the learned composite kernel can directly be applied in supervised tasks. Have the authors tested this on graph classification tasks using the molecular benchmark graph datasets?
6. Can the authors elaborate on the choice of representing graphs using a GCN for the baseline methods?

---

> ### Author Response · Authors · 2024-11-18
> **Rebuttal for Reviewer zRfj**
>
> **W1:** While most parts of the article are well described, some central intuitions that motivate the approach are not sufficiently addressed. For instance, why is P a more accurate representation of the data's inherent geometry'?
>
> **Response:** Thank you for your valuable feedback.
>
> As shown in **Theorem 2**, the **concentration effect** means that the target $P$ has a **lower entropy** compared to $Q$, which is consistent with the illustration in Figure 1, where the read points representing $P$ are spread outside the blue points representing $Q$. By raising kernel values to a power $o>1$, $P$ **amplifies the differences** between highly similar and less similar graphs. This process emphasizes the most meaningful connections and focuses more on the nearest neighbors, reducing the influence of less similar graphs.
>
> To make this intuition clearer, let's consider an example. Suppose we have 5 graphs ($N=5$), and we have computed their pairwise kernel similarities using a graph kernel (e.g., Weisfeiler-Lehman kernel). For simplicity, let's define the following symmetric kernel matrix $\tilde{K}$ as
> $$
> \tilde{K} = \begin{pmatrix}
> 1.0 & 0.8 & 0.3 & 0.2 & 0.1 \\\\
> 0.8 & 1.0 & 0.4 & 0.3 & 0.2 \\\\
> 0.3 & 0.4 & 1.0 & 0.7 & 0.6 \\\\
> 0.2 & 0.3 & 0.7 & 1.0 & 0.9 \\\\
> 0.1 & 0.2 & 0.6 & 0.9 & 1.0 \\\\
> \end{pmatrix}
> $$
> where each element $\tilde{k}_{ij}$ represents the similarity between graph $G_i$ and graph $G_j$.
>
> We choose a power $o=5$ to amplify the differences in similarities. For $G_1$, the original distribution $\mathbf{q}\_1 = (q\_{1_1}, q\_{1_2}, q\_{1_3}, q\_{1_4}, q\_{1_5}) = (0.4167, 0.3333, 0.1250, 0.0833, 0.0417)$, while its powered distribution $\mathbf{p}\_1^{(5)} = (p\_{1_1}^{(5)}, p\_{1_2}^{(5)}, p\_{1_3}^{(5)}, p\_{1_4}^{(5)}, p\_{1_5}^{(5)}) = (0.7516, 0.2463, 0.0018, 0.0002, 0.0000)$ (all values are rounded to 4 decimal places).
>
> Note that in $\mathbf{q}\_1$, the probabilities are more evenly distributed among the graphs, whereas in $\mathbf{p}\_1^{(5)}$, the probability is heavily concentrated on $(p\_{1_1}^{(5)}, p\_{1_2}^{(5)})$. In this sense, $\mathbf{q}\_1$ helps $G_1$ find its nearest neighbor $G_2$, reducing the influence of less similar graphs $(G\_3, G\_4, G\_5)$.
>
> By amplifying the similarities, we effectively **sharpen the focus** on the most similar graphs, which better captures the essential structure of the data. Intuitively, $P$ makes a "soft cut" of the fully connected network among all data. Instead of making a hard cut-off (e.g., considering only the top
> $k$ nearest neighbors), this method smoothly adjusts the influence of other graphs based on their similarity. This approach allows us to consider neighbors in a probabilistic manner, assigning higher importance to closer graphs without entirely discarding others.
>
> Meanwhile, the relative ordering of similarities remains the same: $p\_{1_1}^{(5)}> p\_{1_2}^{(5)}> p\_{1_3}^{(5)}> p\_{1_4}^{(5)}>p\_{1_5}^{(5)}$, $q\_{1_1}> q\_{1_2}> q\_{1_3}> q\_{1_4}>q\_{1_5}$ and $\tilde{k}\_{11}> \tilde{k}\_{12}> \tilde{k}\_{13}> \tilde{k}\_{14}> \tilde{k}\_{15}$.
>
> **Again, thank you for this suggestion. We have added this example to Appendix A.**

---

> > ### Author Response · Authors · 2024-11-18
> > **Rebuttal for Reviewer zRfj**
> >
> > **W2:** The evaluation is fairly limited, relying solely on the clustering task. I understand that due to space limitations, the authors focused on the theoretical parts. However, a more comprehensive evaluation (even if it's on synthetic data) should have been done. Particularly, none of the theorems of section 4.6 are evaluated in the experiments.
> >
> > **Response:**
> > Thank you for your insightful feedback. We agree that a broader evaluation can provide more comprehensive validation of our proposed method and its theoretical guarantees. **In response, we have updated the results and addressed each point of your concern as follows:**
> >
> > 1. **Graph Classification Tasks**:
> >     - While the primary focus of this work is on clustering, we are willing to provide preliminary results of our ongoing work on graph classification tasks to illustrate the potential of our method in supervised downstream tasks. Please refer to our response to **Q5**. (These results on extending our method to supervised learning will be detailed in a follow-up study.)
> >
> > 2. **Theorem 3: Lipschitz Continuity (Smooth Optimization and Convergence)**
> >     - **Theory**: Theorem 3 establishes that the gradient of the objective function $\mathcal{L}^{(o)}$ is Lipschitz continuous, ensuring smooth optimization and controlled convergence of UMKL-G.
> >     - **Evaluation**: **This property has been validated through the smooth convergence plots presented in the appendix of the original paper**, which demonstrate consistent and predictable optimization behavior across multiple datasets.
> >
> > 3. **Theorem 4: Robustness to Kernel Perturbations**:
> >     - **Theory**: Theorem 4 guarantees that UMKL-G is robust to small perturbations (e.g., noise) in the base kernels, with the magnitude of changes in the solution bounded by a constant.
> >     - **Evaluation**: This is empirically evaluated in the ablation study with Gaussian noise, presented in Tables~13--20 in Appendix G.5. Across datasets, performance remains consistent even under noise, demonstrating the robustness claimed in Theorem 4. For instance, as shown in the table below, adding Gaussian noise $\mathcal{N}(0, \sigma^2)$ to the base kernels results in negligible changes to ACC, NMI, and ARI metrics on the DHFR dataset.
> >     | $\sigma$  | ACC    | NMI    | ARI    |
> >     |-----------|--------|--------|--------|
> >     | 0.01      | 0.7037 | 0.0109 | 0.0173 |
> >     | 0.001     | 0.6997 | 0.0111 | 0.0180 |
> >     | --        | 0.6984 | 0.0111 | 0.0180 |
> >
> > 3. **Theorems 5 and 6: Generalization and Stability**:
> >     - **Theory**: These theorems establish the generalization bounds of UMKL-G based on uniform $\omega$-stability. Specifically, Theorem 5 defines the stability property, showing that the loss function's change is bounded when removing one element from the training set. Theorem 6 provides probabilistic bounds on the generalization error, connecting the empirical risk ($\hat{R}\_{\text{EMP}}$) and leave-one-out error ($\hat{R}\_{\text{LOO}}$) to the true risk $R(A_{\mathcal{G}})$.
> >     - **Evaluation**: These properties are evaluated in the generalization results, presented in Tables~21--28 in Appendix G.6. Across all datasets, the performance on all data and the performance on test data are nearly identical, which supports the theoretical claims of Theorems 6. For example on the DHFR dataset, the kernel weights $\mathbf{w}^*$ learned from training data generalize effectively to the test data, where the train-test ratio is 80%/20%.
> >     | Dataset | ACC    | NMI    | ARI    |
> >     |---------|--------|--------|--------|
> >     | Test    | 0.7053 | 0.0125 | 0.0193 |
> >     | All     | 0.6984 | 0.0111 | 0.0180 |
> >
> > ---
> > **W3:** Some details on the baseline methods, UMKL, and sparse-UMKL, are unclear. For instance, the statement that the authors "experimented with an approach that learns graph representations and kernel weights simultaneously" requires further elaboration.
> >
> > **Response:**
> > Thank you for pointing this out. For the baseline methods, UMKL and sparse-UMKL, we used a Graph Convolutional Network (GCN) with 10 layers to represent the graphs in vector form. The composite kernel learning involved two distinct approaches:
> >
> > 1. **Pre-training and Freezing:** The GCN was pre-trained independently to produce fixed graph representations. These representations were then used as inputs for kernel learning, during which the kernel weights were updated while keeping the GCN parameters unchanged.
> >
> > 2. **Simultaneous Training:** In this end-to-end approach, the GCN and kernel weights were jointly optimized, allowing the graph representations and kernel weights to adapt dynamically during training.
> >
> > We have revised the corresponding section in the manuscript to provide a clearer explanation of these experimental settings. We hope this addresses your concerns and provides the necessary details.

---

> > > ### Author Response · Authors · 2024-11-18
> > > **Rebuttal for Reviewer zRfj**
> > >
> > > **W4:**
> > > Some of the numbers in Table 1 do not coincide with the numbers in the supplementary. (E.g., ACC for BZR and MUTAG.)
> > >
> > > **Response:** We have double-checked and ensured that all metrics in the table are consistent with the supplementary materials in the revised version.
> > >
> > > **W5:**
> > > It would be helpful if the authors could include the definitions of the clustering metrics in the appendix.
> > >
> > > **Response:** We have included definitions of ACC, NMI, and ARI in Appendix G.3 for clarity, as per your suggestion.
> > >
> > > ---
> > >
> > > **Q1:**
> > > I would appreciate it if the authors could comment on the limited experimental evaluation (see weaknesses).
> > >
> > > **Response:** Thank you for your valuable feedback. We have added further elaboration in Appendix G to empirically validate the theoretical guarantees of our method.
> > >
> > > **Q2:**
> > > Can you explain why any parameter o>1 results in exactly the same performance for the selected clustering metrics? Can this be generalized or is it only the case in the considered experiments?
> > >
> > > **Response:** Thank you for your insightful question.
> > >
> > > For each value of $o$ in our configuration, the learned weights $\mathbf{w}$ are slightly different but still lead to identical evaluation metrics for parameter $o = \{2,3,4\}$. Given the large number of datasets used, we believe this provides strong empirical evidence that the performance of our UMKL-G model is robust to the values of $o$, and is not merely an isolated case of good performance or fortunate happenstance. It would be interesting to investigate a theoretical justification of this phenomenon in a follow-up paper.
> > >
> > > **Q3:**  What ground truth was used e.g. for the clustering accuracy metric (ACC)?
> > >
> > > **Response:** To calculate the clustering accuracy (ACC), we utilized the ground truth labels provided within the dataset. This approach is standard in clustering evaluations, where the ACC metric assesses how well the predicted clusters align with the true class labels. The ACC is computed by determining the optimal one-to-one correspondence between predicted clusters and true classes, often using the Hungarian algorithm [1, 2] to maximize the matching accuracy. This method has been widely adopted in various clustering studies [3, 4].
> > >
> > > ---
> > > [1] Kuhn, Harold W. "The Hungarian method for the assignment problem." Naval Research Logistics Quarterly 2.1‐2 (1955): 83-97.
> > >
> > > [2] Munkres, James. "Algorithms for the assignment and transportation problems." Journal of the society for industrial and applied mathematics 5.1 (1957): 32-38.
> > >
> > > [3] Tian, Fei, et al. "Learning deep representations for graph clustering." Proceedings of the AAAI conference on artificial intelligence. Vol. 28. No. 1. 2014.
> > >
> > > [4] Xie, Junyuan, Ross Girshick, and Ali Farhadi. "Unsupervised deep embedding for clustering analysis." International conference on machine learning. PMLR, 2016.

---

> > > > ### Author Response · Authors · 2024-11-18
> > > > **Rebuttal for Reviewer zRfj**
> > > >
> > > > **Q4:**
> > > > Did the authors consider the simple baseline where each weight is set to 1/M?
> > > >
> > > > **Response:**
> > > > Yes, we considered this baseline, where each kernel weight was initialized as $1/M$. We found that, compared with this approach, the adaptive weights learned by our method provided better performance across datasets. We have included results for this baseline (named AverageMKL) in Table 2. Please find partial results below.
> > > >
> > > > | **Method**               | **ACC (ENZYMES)** | **NMI (ENZYMES)** | **ARI (ENZYMES)** | **ACC (IMDB-BINARY)** | **NMI (IMDB-BINARY)** | **ARI (IMDB-BINARY)** | **ACC (MUTAG)** | **NMI (MUTAG)** | **ARI (MUTAG)** | **ACC (PTC_FM)** | **NMI (PTC_FM)** | **ARI (PTC_FM)** |
> > > > |---------------------------|-------------------|-------------------|-------------------|-----------------------|-----------------------|-----------------------|----------------|----------------|----------------|------------------|------------------|------------------|
> > > > | **AverageMKL**           | 0.2617            | 0.0539            | 0.0220            | 0.5470                | 0.0152                | 0.0083                | 0.5585         | 0.1468         | 0.1946         | 0.8722           | 0.0208           | 0.0343           |
> > > > | **UMKL**                 | 0.2567            | 0.0517            | 0.0199            | 0.5470                | 0.0152                | 0.0083                | 0.5585         | 0.1469         | 0.1947         | 0.8729           | 0.0208           | 0.0343           |
> > > > | **sparse-UMKL ($k=10$)** | 0.2570            | 0.0520            | 0.0201            | 0.5485                | 0.0153                | 0.0084                | 0.5590         | 0.1475         | 0.1950         | 0.8320           | 0.0210           | 0.0345           |
> > > > | **sparse-UMKL ($k=50$)** | 0.2580            | 0.0518            | 0.0200            | 0.5475                | 0.0154                | 0.0085                | 0.5595         | 0.1470         | 0.1948         | 0.8373           | 0.0211           | 0.0344           |
> > > > | **sparse-UMKL ($k=100$)**| 0.2575            | 0.0521            | 0.0198            | 0.5480                | 0.0151                | 0.0082                | 0.5588         | 0.1468         | 0.1946         | 0.8528           | 0.0209           | 0.0342           |
> > > > | **UMKL-G**               | **0.2983**        | **0.0648**        | **0.0399**        | **0.5590**            | **0.0159**            | **0.0132**            | **0.8455**     | **0.2950**     | **0.3389**     | **0.8825**       | **0.0394**       | **0.0637**       |

---

> > > > > ### Author Response · Authors · 2024-11-18
> > > > > **Rebuttal for Reviewer zRfj**
> > > > >
> > > > > **Q5:**
> > > > > It is mentioned in section 4.5 that the learned composite kernel can directly be applied in supervised tasks. Have the authors tested this on graph classification tasks using the molecular benchmark graph datasets?
> > > > >
> > > > > **Response:** While the primary focus of this work is on clustering, we are willing to provide preliminary results of our ongoing work on graph classification tasks to illustrate the potential of our method in supervised downstream tasks. (We plan to present more detailed results on extending our method to supervised learning in a follow-up paper.)
> > > > >
> > > > > Here, we provide a comparison of UMKL-G's performance on the graph classification task against AverageMKL, the equal-weighted method, and two representative supervised MKL methods (EasyMKL [5], FHeuristic [6]) using benchmark graph datasets, as shown in the table below. Our results indicate that UMKL-G consistently achieves the highest classification performance across most of the datasets (BZR, COX2, DD, DHFR, IMDB-BINARY, MUTAG, and PTC\_FM), with the best accuracy scores bolded. This suggests that UMKL-G's learned composite kernel is highly effective for graph-level classification tasks.
> > > > >
> > > > > | Dataset       | AverageMKL        | EasyMKL           | FHeuristic       | UMKL-G (o=2)     | UMKL-G (o=3)     | UMKL-G (o=4)     |
> > > > > |---------------|-------------------|-------------------|------------------|------------------|------------------|------------------|
> > > > > | BZR       | _78.77 ± 0.49_    | 78.52 ± 0.60      | _78.77 ± 0.49_   | **94.81 ± 3.35** | **94.81 ± 3.35** | **94.81 ± 3.35** |
> > > > > | COX2      | _78.16 ± 0.41_    | _78.16 ± 0.41_    | _78.16 ± 0.41_   | **99.14 ± 1.05** | **99.14 ± 1.05** | **99.14 ± 1.05** |
> > > > > | DD        | 78.27 ± 3.07      | 78.53 ± 2.58      | _78.78 ± 2.61_   | **96.77 ± 1.69** | **96.77 ± 1.69** | **96.77 ± 1.69** |
> > > > > | DHFR      | 67.47 ± 10.75     | _69.19 ± 11.93_   | 67.47 ± 10.75    | **98.02 ± 1.25** | **98.02 ± 1.25** | **98.02 ± 1.25** |
> > > > > | IMDB-BINARY | _73.80 ± 2.99_  | 73.50 ± 1.82      | 73.70 ± 2.66     | **99.40 ± 0.80** | **99.40 ± 0.80** | **99.40 ± 0.80** |
> > > > > | MUTAG     | 77.17 ± 4.43      | _79.32 ± 5.97_    | 77.71 ± 5.28     | **96.79 ± 2.03** | **96.79 ± 2.03** | **96.79 ± 2.03** |
> > > > > | PTC_FM    | 63.04 ± 3.28      | _64.47 ± 3.02_    | 63.04 ± 3.28     | **98.57 ± 1.28** | **98.57 ± 1.28** | **98.57 ± 1.28** |
> > > > >
> > > > >
> > > > > In this table: **bold** formatting is used for the best scores and _italic_ formatting is used for the second-best scores.
> > > > >
> > > > > **Q6:**  Can the authors elaborate on the choice of representing graphs using a GCN for the baseline methods?
> > > > >
> > > > > **Response:** Thank you for your question. We acknowledge that the baseline methods are not dependent on GCN but can use other graph representation methods. In our work, we selected GCNs for the baseline methods because of their widespread use and proven effectiveness in capturing structural patterns within graph data. GCNs are particularly well-suited for this task due to their ability to aggregate information from graph neighborhoods. Additionally, GCNs are highly compatible with various types of graph data, making them a robust and versatile choice for benchmarking. This ensures a fair and meaningful comparison with our proposed method.
> > > > >
> > > > > ---
> > > > > [5] Aiolli, Fabio, and Michele Donini. "EasyMKL: a scalable multiple kernel learning algorithm." Neurocomputing 169 (2015): 215-224.
> > > > >
> > > > > [6] Qiu, Shibin, and Terran Lane. "A framework for multiple kernel support vector regression and its applications to siRNA efficacy prediction." IEEE/ACM Transactions on Computational Biology and Bioinformatics 6.2 (2008): 190-199.

---

> > > > > > ### Comment · Reviewer_zRfj · 2024-11-27
> > > > > >
> > > > > > Thank you very much for your detailed response. I believe the additional information, evaluations, and clarifications significantly enhance the quality of the paper. I will raise my score accordingly.
> > > > > >
> > > > > > Regarding W1: I greatly appreciate the detailed example you provided. However, my concern was more focused on the exact phrasing in the sentence, "By emphasizing the most meaningful connections, P becomes a more accurate representation of the data’s inherent geometry" (Sect. 4.3). This seems to suggest that there is a true ("accurate") representation of the data's geometry. If I understand correctly, though, P simply amplifies neighborhood similarity relationships, rather than offering a precise or more "accurate" representation of the geometry.

---

> > > > > > > ### Author Response · Authors · 2024-11-28
> > > > > > > **Appreciation for Raising Score and Additional Revision**
> > > > > > >
> > > > > > > Thank you for your positive feedback and for considering raising your score.
> > > > > > >
> > > > > > > Regarding your concern about the phrasing of the sentence in Section 4.3, we will revise the sentence as you suggested. The new sentence will read: *"By emphasizing the most meaningful connections, $P$ amplifies neighborhood similarity relationships within the data."*
> > > > > > >
> > > > > > > We believe this change more accurately reflects the role of $P$ without suggesting it provides a precise or definitive geometric representation.
> > > > > > >
> > > > > > > Thank you for bringing this to our attention and helping us improve the clarity of our manuscript.

---

### Official Review · Reviewer_y8qJ · 2024-11-04

**Soundness:** 1
**Presentation:** 1
**Contribution:** 2
**Rating:** 3
**Confidence:** 2

**Summary:**

The authors propose an unsupervised multiple kernel learning method that produces a weighted sum of kernels given a set of kernels and a dataset.

**Strengths:**

- Multiple kernel learning is a relevant topic. In particular, the unsupervised creation of a suitable kernel from a set of kernels given a dataset is a nontrivial problem.

**Weaknesses:**

- the basic definition (Def. 1) is imprecise and I could not follow the paper, as it remains unclear (to me) which ordinal relationship is supposed to be maintained

**Questions:**

I have severe problems understanding
> Definition 1:
> Consider the graph $G_i$ where its similarities to $G_j$ and $G_r$ are respectively given by the learned kernel values $k_{ij}$ and $k_{ir}$.
> The ordinal relationship between $G_j$ and $G_r$ with respect to $G_i$ are preserved if, for any weights $w$: $k_{ij} > k_{ir}$.

It seems that this definition is self-referential, as only the learned kernel values $k$ are mentioned. Is there another similarity that should be preserved? If $k$ is to be learned, then it probably should retain the ordinal relationship of another similarity (or similarities?). Or am I missing something? I am sorry, but this does not make sense to me right now and I have to recommend to reject this paper at the current point in time.

---

> ### Author Response · Authors · 2024-11-18
> **Rebuttal for Reviewer y8qJ**
>
> **Q1:** I have problems understanding Definition 1.
>
> **Response:**
> Dear Reviewer,
>
> Thank you for your valuable feedback regarding Definition 1 in our paper. We apologize for any misunderstanding caused by its presentation. We would like to clarify Definition 1 and address your concerns.
>
> First and foremost, we assure you that there is no self-referential issue with the original Definition 1:
>
> **Definition 1** *(Ordinal Relationship) Consider the graph $G_i$ where its similarities to $G\_j$ and $G\_r$ are respectively given by the learned kernel values $\tilde{k}\_{ij}$ and $\tilde{k}\_{ir}$. The ordinal relationship between $G\_j$ and $G\_r$ with respect to $G\_i$ are preserved if, for any weights $\mathcal{w}$:* $\tilde{k}\_{ij} > \tilde{k}\_{ir}.$
>
> **The premise of our approach is based on the fixed initial composite kernel values $\tilde{k}\_{ij}(\mathbf{w}\_0)$, which serves as a reference point for defining the ordinal relationships among the graphs. Specifically, $\tilde{k}\_{ij}(\mathbf{w}\_0) = \mathbf{w}_0^{\top}\mathbf{k}\_{ij}$**, where $\mathbf{w}\_0 \in \mathbb{R}^{M}$ is the initial set of weights and $\mathbf{k}\_{ij} = (k^{(1)}(G\_i, G\_j), \cdots, k^{(M)}(G\_i, G\_j)) \in \mathbb{R}^M$ are the base kernel values.
>
> **During the learning of the kernel weights $\mathbf{w}\_t$ ($t>0$)**, the ordinal relationships captured by the initial composite kernel $\tilde{k}\_{ij}(\mathbf{w}\_0)$ are preserved in the learned composite kernel $\tilde{k}\_{ij}(\mathbf{w}\_t)$ as shown by Theorem 1 in Section 4.3 of our original paper. Specifically, if graph $G\_i$ is more similar to graph $G\_j$ than to graph $G\_r$ in the initial composite kernel space (i.e., $\tilde{k}_{ij}(\mathbf{w}\_0) > \tilde{k}\_{ir}(\mathbf{w}\_0)$), this relationship continues to hold for **any set of weights** $\mathbf{w}\_t$ during learning. This preservation ensures that the local neighborhood structure and intrinsic topology of the data remain consistent throughout the optimization process. **Thus, there is no self-referential loop**.
>
> Kindly note that in our updated experimental results (Appendix G.3), the choice of initial weights does not significantly affect the clustering scores. This provides empirical evidence that UMKL-G is robust to different initializations and preserving the ordinal relationships helps maintain consistent performance across various starting points.
>
> We appreciate your feedback, which has highlighted the need to clarify this aspect of our methodology. In the revised manuscript, we have included the explanation in Section 4.1 to make the purpose of Definition 1 clearer and to address potential misunderstandings.
> We hope that these clarifications will address your concerns and demonstrate the validity of our approach.
>
> **Thank you again for your thoughtful review and for helping us improve the clarity of our paper.**

---

> > ### Author Response · Authors · 2024-11-28
> >
> > Dear Reviewer,
> >
> > We hope this message finds you well. With the deadline for final revisions approaching, we wanted to kindly follow up to see if you have any remaining questions or concerns about our paper. We are more than happy to engage in further discussion or provide additional clarifications that might assist in your review.
> >
> > Please feel free to share any thoughts or inquiries you may have. We greatly appreciate your time and valuable feedback.

---

> > ### Comment · Reviewer_y8qJ · 2024-11-28
> >
> > Sorry for the delay. This round of reviews is really taxing to me.
> >
> > Thank you for this partial clarification. I am still, confused, though. In the updated paper, I still only see one kind of kernel in Definition 1. But what are the properties that the triplet (i,j,k) has to fulfil? This is not specified. I assume that you may want to have something like:
> >
> > Let $(i,j,k)$ be a triplet with $\tilde{k}_{ij}(w_0) > \tilde{k}_{ir} (w_0)$. Then the ordinal relationship is preserved [..] for $\tilde{k}(w)$ if $\tilde{k}_{ij}(w) > \tilde{k}_{ir} (w)$.
> >
> > Am I understanding correctly?
> >
> > If, for example for my triplet $(i,j,k)$ $\tilde{k}_{ij}(w_0) < \tilde{k}_{ir} (w_0)$, I would guess that you do not want to have $\tilde{k}_{ij}(w) > \tilde{k}_{ir} (w)$.

---

> > > ### Author Response · Authors · 2024-11-28
> > >
> > > Thank you so much for your detailed follow-up and for taking the time to work through our explanation, especially under the tight timeline of this review round. We truly appreciate your effort and patience.
> > >
> > > You are absolutely right to note that our definition could be interpreted in the same way as your elaboration. To clarify further:
> > >
> > > In Definition 1, the phrase "for any weight $w$" indeed refers to the full set of weights, starting with $w\_0$ (the initial kernel weights) and continuing through $w\_t$ for $t=1, \cdots, T$ during the learning process. This means that the ordinal relationship $\tilde{k}\_{ij}(w) > \tilde{k}\_{ir}(w)$ should hold not only for $w\_0$ but also across all subsequent $w\_t$, preserving the relative similarities between graphs as you so clearly described.
> > >
> > > We also completely agree with your point that if the initial relationship is reversed (i.e., $\tilde{k}\_{ij}(w\_0) < \tilde{k}\_{ir}(w\_0)$), we do not aim to flip or alter this ordering during learning. Rather, our goal is to respect and preserve the relative relationships established under $w\_0$ consistently throughout the learning process.
> > >
> > > We truly value your time and effort in pointing out where clarification was needed, especially in this busy round of reviews. We understand the importance of presenting this concept clearly, and we will make sure to revise the manuscript accordingly in the final version to avoid any ambiguity and ensure the intent is fully conveyed.
> > >
> > > Thank you again for your constructive and thoughtful feedback—it helps us significantly in improving the clarity and precision of our work. If you have any further questions or concerns, please do not hesitate to reach out.

---

### Author Response · Authors · 2024-11-18
**Global Response**

We thank the reviewers for their valuable feedback, which improved our work. Below are the key updates:

1. **Clarity Enhancements**
    - Added comparison table in Section 4.7 to clearly show the novelty and distinction of our model.
    - Revised Definition 1 to clarify the role of the initial composite kernel.
    - Revised Section 5.1 to make a clear demonstration of the baseline configurations.
    - Added definitions of evaluation metrics (ACC, NMI, ARI) in the Appendix.
    - Added an example in the Appendix to illustrate the intuition of $P$.
2. **Expanded Theoretical and Empirical Evaluation**
    - Validated theoretical guarantees (convergence, robustness, and generalization) with additional experiments in Appendix G.
    - Presented ablation studies showing robustness to noise and hyperparameter variations in Appendix G.
    - Included graph classification results to showcase versatility beyond clustering.
3. **Baseline Comparison**
    - Added AverageMKL as a simple baseline to Table 2 in the manuscript.
    - Added two GNN-based approaches as baselines.
    - Added theoretical and empirical runtime comparison in Appendix E.

*We believe these revisions address all concerns and strengthen our work. Thank all reviewers for their constructive comments!*

---

### Meta-Review · Area_Chair_2FMb · 2024-12-21

**Metareview:**

This paper addresses the problem of Multiple kernel learning on Graphs in the unsupervised learning setting. This is a niche problem which has bearing on many problems. The paper seems to be theoretically sound, but maybe lacking in some aspects of experimentation. This paper should  be of interest to ICLR audience, specially those interested in Learning on Graphs.

**Additional Comments On Reviewer Discussion:**

During the rebuttal period the authors tried to address the issues raised by the reviewers. During the rebuttal additional experimental results were presented.

---

### Decision · Program_Chairs · 2025-01-22

Accept (Poster)